# Histone methyltransferase ASH1L primes metastases and metabolic reprogramming of macrophages in the bone niche

Chenling Meng[1], Kevin Lin[2], Wei Shi[1], Hongqi Teng[1], Xinhai Wan[3], Anna DeBruine[1,4], Yin Wang [1], Xin Liang [1], Javier Leo[1,4], Feiyu Chen[1], Qianlin Gu[1], Jie Zhang[1], Vivien Van[5], Kiersten L. Maldonado[5], Boyi Gan [1], Li Ma [1], Yue Lu[2,6] ✉ & Di Zhao [1,6] ✉

Bone metastasis is a major cause of cancer death; however, the epigenetic determinants driving this process remain elusive. Here, we report that histone methyltransferase ASH1L is genetically amplified and is required for bone metastasis in men with prostate cancer. ASH1L rewires histone methylations and cooperates with HIF-1α to induce pro-metastatic transcriptome in invading cancer cells, resulting in monocyte differentiation into lipid-associated macrophage (LA-TAM) and enhancing their pro-tumoral phenotype in the metastatic bone niche. We identified IGF-2 as a direct target of ASH1L/HIF-1α and mediates LA-TAMs' differentiation and phenotypic changes by reprogramming oxidative phosphorylation. Pharmacologic inhibition of the ASH1L-HIF-1α-macrophages axis elicits robust anti-metastasis responses in preclinical models. Our study demonstrates epigenetic alterations in cancer cells reprogram metabolism and features of myeloid components, facilitating metastatic outgrowth. It establishes ASH1L as an epigenetic driver priming metastasis and macrophage plasticity in the bone niche, providing a bona fide therapeutic target in metastatic malignancies.

Metastasis is the primary cause of cancer mortality. The success of cancer therapies and improvements in overall survival mainly depend on the prevention and treatment of metastatic malignancies. Although advances have been made in cancer treatment over the decades, effective therapies are urgently needed for patients with metastatic disease. During metastasis, cancer cells undergo loss of cellular adhesion, increased invasiveness, dissemination into the circulation, and eventual colonization at distant sites[1]. Recent large-scale genomic studies have uncovered many emerging genetic alterations in metastatic cancers[2-4]. Characterizing and functionally defining the roles of

these alterations in different steps of the metastatic cascade will advance our understanding of the molecular and cellular basis of metastasis and facilitate the development of improved treatment for patients with lethal diseases.

Increasing evidence suggests that most steps in the metastasis cascade are largely driven by interactions between cancer cells, extracellular matrix, stromal cells, and immune components in the tumor microenvironment and metastatic niche[5-8]. Tumor-associated macrophages (TAMs) are among the most abundant myeloid cells in primary tumors and metastasis lesions and are associated with cancer

[1]Department of Experimental Radiation Oncology, The University of Texas MD Anderson Cancer Center, Houston, TX 77030, USA. [2]Department of Epigenetics and Molecular Carcinogenesis, The University of Texas MD Anderson Cancer Center, Houston, TX 77030, USA. [3]Department of Endocrine Neoplasia & Hormonal Disorders, The University of Texas MD Anderson Cancer Center, Houston, TX 77030, USA. [4]The University of Texas MD Anderson Cancer Center UTHealth Graduate School of Biomedical Sciences, Houston, TX 77030, USA. [5]Department of Imaging Physics, The University of Texas MD Anderson Cancer Center, Houston, TX 77030, USA. [6]These authors jointly supervised this work: Yue Lu, Di Zhao. ✉e-mail: YLu4@mdanderson.org; dzhao2@mdanderson.org

progression, metastasis, and therapy resistance[9–12]. The molecular and cellular communication between invading cancer cells and TAMs results in tumor vascularization, epithelial-to-mesenchymal transition (EMT), intravasation, and colonization of distal sites, including the bone marrow[12–15]. Understanding how invading cancer cells interact with TAMs facilitates the development of effective therapeutics and combinations for metastatic diseases.

Histone methyltransferases (HMTs) catalyze the methylation of lysine and arginine residues on histone tails, which affect chromatin compaction and transcriptional programs. ASH1L (absent, small, or homeotic 1-like) is a SET domain-containing HMT that methylates Lys36 of histone H3 (H3K36me2/3) and is required for maintaining the trimethylation of Lys4 of H3 (H3K4me3) and active transcription[16–20]. By regulating *HOX* gene expression, ASH1L supports the proliferation and survival of hematopoietic stem cells and plays a vital role in leukemogenesis[20–23]. Overexpression of ASH1L was also found in solid tumors[24,25]; however, its biological function in metastatic diseases remains elusive.

Here, we identify ASH1L as an epigenetic determinant and a bona fide therapeutic target in metastatic malignancies. We uncovered that ASH1L is genetically amplified and overexpressed in diverse metastatic cancer types. Using prostate cancer (PCa) as a model system, we characterized the loss-of-function and gain-of-function effects of ASH1L on cancer invasiveness and bone metastases. Then, we performed high-throughput transcriptomic and epigenetic profiling to dissect the molecular basis of ASH1L's role in transforming invading cancer cells. Besides, we generated a PCa bone metastasis model in immunocompetent mice, which recapitulates human diseases. Combining it with single-cell transcriptomics, we determined the role of ASH1L in modulating the communication between disseminated cancer cells and host immune cells in the bone niche. Furthermore, we assessed the relevance of ASH1L amplification or overexpression with cancer progression, clinical outcomes, and immunophenotype in men with metastatic PCa. At last, we evaluated the therapeutic potential of agents targeting ASH1L and its partner hypoxia-inducible factor (HIF) in preclinical models of metastatic PCa.

## Results

### ASH1L is genetically amplified and overexpressed in metastatic cancers

In a meta-analysis of human cancers, we found that amplification and gain of the *ASH1L* gene occur in >40% of metastatic PCa, much more frequently than in localized tumors (Fig. 1a). High ASH1L mRNA levels are correlated with genetic amplification status and are strongly associated with metastatic diseases (Supplementary Fig. 1a, b). Besides, we found that genetic amplification of *ASH1L* was associated with worse overall survival in men with metastatic PCa (Supplementary Fig. 1c), and human prostate tumors with high ASH1L mRNA levels exhibited an enriched cancer metastasis profile (Supplementary Fig. 1d). Immunohistochemistry (IHC) staining of ASH1L in human PCa tumors revealed that ASH1L protein levels were positively correlated with cancer progression and aggressiveness (Fig. 1b and Supplementary Fig. 1e). In human PCa cell lines, PC-3M, a metastasis-derived variant from PC-3, showed an increased expression of ASH1L (Supplementary Fig. 1f). Our prior studies established and characterized a metastatic PCa genetically engineered mouse (GEM) model with PB-Cre-driven co-deletion of *Pten/Smad4/Trp53* (*PbPPS*)[26,27]. This model develops invasive prostate adenocarcinoma with lymph node, lung, and liver metastases (Supplementary Fig. 1g). We found that ASH1L was highly expressed in invasive cancer cells and metastatic tumors in male *PbPPS* mice (Fig. 1c and Supplementary Fig. 1h). In addition to PCa, frequent genetic amplification of ASH1L was found in other malignancies (Supplementary Fig. 1i). We observed that breast and melanoma cells with higher metastatic potential also expressed higher levels of ASH1L than their parent cells (Supplementary Fig. 1j).

Collectively, these studies in human samples, GEM models, and cancer cell lines demonstrate that histone methyltransferase ASH1L is genetically amplified and overexpressed in metastatic PCa, among other malignancies.

### Depletion of ASH1L suppresses prostate cancer invasiveness and bone metastases

Next, we knocked out *ASH1L* in PC-3M cells using the CRISPR/Cas9 system (Fig. 1d). and found that loss of ASH1L had a moderate effect on cell proliferation or tumor growth (Supplementary Fig. 2a, b), but it dramatically reduced the migration of PC-3M cells (Supplementary Fig. 2c). Besides, we performed 2D and 3D invasion assays and found that ASH1L depletion markedly suppressed PCa cell invasion through the extracellular matrix, a crucial step in metastatic progression (Fig. 1e and Supplementary Fig. 2d). Similar results were also observed in DU145, another metastatic PCa cell line (Supplementary Fig. 2e–g).

Next, we assessed the effects of ASH1L depletion on skeletal metastases, which occur in 80% of advanced PCa and confer a high level of morbidity[28]. To this end, we intracardially injected control and ASH1L-depleted PC-3M cells expressing RFP reporter and firefly luciferase into male nude mice. Bioluminescence imaging showed that depletion of ASH1L suppressed the metastasis of cancer cells to the tibia, femur, or mandible (Fig. 1f). Importantly, depletion of ASH1L significantly prolonged the overall and metastasis-free survival durations of mice (Fig. 1g and Supplementary Fig. 2h). Ex vivo fluorescence imaging showed that control PC-3M cells metastasized to the bones in 67% (8 in 12) of mice; in contrast, no skeletal metastasis was detected in *ASH1L* knockout groups (Supplementary Fig. 2i). The bone metastasis model derived from PC-3M cells often forms osteolytic tumors in the bone niche[29]. Histologic analyses showed that ASH1L depletion suppressed colonization of PC-3M cells in the bone, decreased osteoclast numbers, and attenuated osteolytic lesion formation (Fig. 1h and Supplementary Fig. 2j). These in vitro and in vivo studies indicated that ASH1L is essential in PCa invasiveness and metastases to the bone.

### ASH1L overexpression promotes PCa metastatic outgrowth in the bone

Given that the *ASH1L* gene is amplified and overexpressed in metastatic diseases, we further determined the impact of ASH1L overexpression on PCa cell migration, invasion, and metastasis. Like other high–molecular–weight histone methyltransferases[30], ectopic expression of full-length ASH1L protein in mammalian cells is challenging[31,32]. Thus, we constructed three ASH1L fragments that respectively encode 1–882 amino acids (F1), 883–1890 amino acids (F2), and 1891–2969 amino acids (F3) of human ASH1L protein (Supplementary Fig. 3a, b). Notably, fragment F3 contains all functional domains of the ASH1L protein (Supplementary Fig. 3a) and showed the activity of reversing the H3K36me3 and H3K4me3 loss in ASH1L-depleted PC-3M cells (Supplementary Fig. 3c). Besides, reintroducing the ASH1L-F3 fragment partially rescued the migration and invasion capacities of ASH1L-depleted PC-3M cells in 2D and 3D culture systems (Supplementary Fig. 3d, e), suggesting that the F3 fragment is essential and sufficient for histone methyltransferase activity of ASH1L.

Next, we introduced ASH1L-F3 into LNCaP (a PCa cell line with low metastatic potential and low expression of ASH1L) and confirmed its histone methyltransferase activity at H3K36 and H3K4 (Fig. 1i). Although overexpression of ASH1L-F3 did not affect cell growth, it significantly augmented cell migration and invasion; while F1 or F2 had little effect on cell migration (Supplementary Fig. 3f–i). To further characterize the effects of ASH1L overexpression in vivo, we injected LNCaP cells expressing vector control or ASH1L-F3 into the tibias of male nude mice, followed by weekly bioluminescence imaging. As shown in Fig. 1j, overexpression of ASH1L-F3 significantly promoted metastatic outgrowth of PCa in the bone. Mice in the ASH1L-F3 group had worse overall survival than the control group (Fig. 1k). Ex vivo

fluorescent and histological analyses also confirmed that ASH1L-F3 overexpression remarkably induced PCa metastatic outgrowth in the bone, along with new bone formation (Supplementary Fig. 3j–l).

To verify if methyltransferase activity is required for ASH1L's role in promoting cancer invasiveness, we constructed a catalytically inactive mutant of ASH1L-F3, F2260A[16,32], and then introduced it into LNCaP cells (Supplementary Fig. 3m). We confirmed that the F2260A mutant is defective to catalyze methylations of histone H3 at K4 and K36 in PCa cells (Supplementary Fig. 3n). Compared to wildtype ASH1L-F3, overexpression of F2260A mutant failed to induce cancer cell migration and invasion (Supplementary Fig. 3o), suggesting methyltransferase activity of ASH1L is required for its pro-metastatic role. Altogether, these loss-of-function and gain-of-function studies establish the crucial role of ASH1L in driving cancer invasiveness and bone metastasis via its methyltransferase activity.

## ASH1L induces pro-metastatic transcriptome via methylations at H3K4 and H3K36

To understand how ASH1L promotes metastasis, we performed transcriptome profiling using RNA sequencing (RNA-seq) in control versus ASH1L-depleted PC-3M cells. Differential gene expression analysis identified 1378 downregulated genes and 307 upregulated genes (false discovery rate FDR ≤0.05, and Fold change FC ≥2 in both sgRNAs targeting ASH1L) upon ASH1L depletion (Fig. 2a–c, Supplementary Fig. 4a, b, and Supplementary Data 1), reinforcing the transcription-activating role of ASH1L in metastatic cells. Ingenuity pathway analysis (IPA) showed a strong association between the differentially expressed genes (DEGs) and metastasis-related pathways, such as EMT, extracellular matrix remodeling, integrin signaling, angiogenesis pathways, and HIF-1α signaling (Fig. 2b).

Prior studies demonstrated that ASH1L is involved in epigenetic reprogramming by catalyzing H3K36me2/3 and maintaining H3K4me3[16–18,20], which are tightly associated with active transcription[33–35]. In metastatic PC-3M cells, we found that ASH1L depletion caused a global reduction in H3K36 and H3K4 methylations (Fig. 1d); while overexpression of ASH1L increased these histone marks (Fig. 1i). To further dissect the impact of ASH1L on the landscape of histone methylations, we performed cleavage under targets and release using nuclease sequencing (CUT&RUN-seq)[36] to capture the genomic distribution of H3K4me3 and H3K36me3 in PC-3M cells with or without ASH1L depletion. We grouped all protein-coding genes into five clusters based on changes in H3K4me3 and H3K36me3 marks upon ASH1L depletion (Fig. 2d, e and Supplementary Fig. 4c, d). Among these clusters, Cluster 1 (C1, 3% of all genes) showed decreased signals in both histone marks; Cluster 2 (C2, 5.3% of all genes) had reduced H3K4me3 only; Cluster 3 (C3, 3.8% of all genes) presented decreased H3K36me3 only; Cluster 4 (C4, 2% of all genes) included genes with increased methylations at H3K4 and/or H3K36; and genes without changes in either mark were classed into Cluster 5 (C5, 86% of all genes). As expected, the genome-wide distribution of H3K4me3 was primarily observed at gene promoters (Fig. 2d, e and Supplementary Fig. 4c, d). Upon ASH1L depletion, the intensity of the H3K4me3 signal in 1622 genes (C1 + C2, 8.3% of all genes) was significantly decreased (FDR ≤0.05; FC ≥1.5) (Fig. 2d, e and Supplementary Data 2). In contrast, the signal of H3K36me3 in PC-3M cells was distributed across gene bodies, starting from the transcription start site (TSS) and gradually increasing towards the transcript end sites (TES) (Fig. 2d, e and Supplementary Fig. 4c, d). Depletion of ASH1L significantly reduced the H3K36me3 signal (FDR ≤0.05; FC ≥1.5) at the gene bodies of 1330 genes (C1 + C3, 6.8% of all genes) (Fig. 2d, e and Supplementary Data 3).

Furthermore, we assessed the impact of ASH1L depletion on gene expression within each gene cluster. The Venn diagram analysis confirmed a significant association between the downregulated DEGs and the decreased H3K4me3 or H3K36me3 signal. Specifically, 46% and 35% of the downregulated DEGs (FDR ≤0.05; FC ≥1.5) showed

decreased H3K4me3 and H3K36me3 signals, respectively (p values of overlap <1e-300, by Fisher's exact test) (Fig. 2f), suggesting that these genes (n = 1180) are the most likely direct targets of ASH1L in metastatic cancer cells (Supplementary Data 4). Both the heatmap and box plots demonstrated a strong association between the reduction of H3K4me3 or H3K36me3 and the downregulation of mRNA levels in Clusters 1–3, with Cluster 1 exhibiting the most significant decreases in gene expression (Fig. 2d, g and Supplementary Fig. 4e). Notably, we found numerous metastasis-associated genes as ASH1L direct targets (highlighted in Supplementary Data 4). Upon ASH1L depletion, most of them showed diminished signals of both marks (PLAU); however, some displayed dramatic decreases in H3K4me3 signal only (MMP14) or H3K36me3 signal only (VEGFC) (Fig. 2h). Of note, all three types of histone methylation patterns are strongly associated with the transcriptional activation of pro-metastatic genes (Fig. 2h).

These transcriptional and epigenetic studies demonstrate that ASH1L governs the transcriptional activation of pro-metastatic genes in invading cancer cells via reprogramming histone methylations at H3K4 and H3K36. It is worth noting that, in leukemia, ASH1L plays a key role in activating HOX family genes that facilitate leukemogenesis[20–23]. Although HOX genes are involved in the tumor invasion[37], ASH1L depletion affected neither the expression nor H3K4me3/H3K36me3 marks of HOX cluster genes in metastatic PCa (Fig. 2h), suggesting the regulation of ASH1L on target genes is highly selective and context-specific.

## ASH1L co-opts with HIF-1α to induce pro-metastatic genes and enhance invasiveness

As master regulators of hypoxia signaling, transcriptional factor HIFs activate the expression of genes that mediate angiogenesis, cell motility, EMT, degradation of extracellular matrix, invasion, and metastasis[38]. HIFs are heterodimeric proteins consisting of an oxygen-sensitive HIF-α (HIF-1α, HIF-2α, or HIF-3α) subunit and a constitutively expressed HIF-1β subunit. Pathway analysis indicated that ASH1L is associated with HIF-1α signaling (Fig. 2b). Gene set enrichment analysis (GSEA) also revealed a significant downregulation of the hypoxia pathway and HIF-1α target genes upon ASH1L depletion (Fig. 3a–d), particularly those involved in cancer cell invasion (Snail, TGFB, and MET), extracellular matrix remodeling (MMPs and PLAU), and angiogenesis (VEGFs). Besides, CUT&RUN profiling verified the remarkable decrease of H3K4me3 and/or H3K36me3 marks on HIF-1α target genes upon ASH1L depletion (Fig. 3d), suggesting ASH1L contributes to the HIF-1α transcriptional program by modulating histone methylations. Then, we analyzed a published HIF-1α ChIP-sequencing dataset in PC-3 cells (GSE106305)[39] and found that 83.2% (982 in 1180) of ASH1L direct target genes (Supplementary Data 4) were bound with HIF-1α protein in their promoter regions, including those are associated with cancer invasiveness and metastasis (Supplementary Fig. 5a and Supplementary Data 5). These results indicated that HIF-1α is an important partner of ASH1L in regulating pro-metastasis genes in PCa.

Furthermore, qPCR assays revealed that ASH1L knockout dramatically down-regulates the mRNA levels of metastasis-associated HIF-1α target genes in PC-3M, whereas reintroduction of ASH1L-F3 fully rescued them (Fig. 3e and Supplementary Fig. 5b). LNCaP cells cultured under a normoxia condition have low levels of HIF-1α due to active protein degradation, but the low oxygen levels within the bone niche stabilize HIF-1α protein in metastatic outgrowth tumors (Supplementary Fig. 5c). To mimic this hypoxia condition, we treated the control and ASH1L-F3 overexpressing LNCaP cells with $CoCl_2$ and found that ASH1L-F3 significantly induced pro-metastatic gene expression (Fig. 3f), consistent with its invasive phenotype observed in vitro and in vivo (Fig. 1j, k and Supplementary Fig. 3). In contrast, the F2260A mutant showed impaired activity in inducing pro-metastatic gene transcription (Supplementary Fig. 5d), suggesting ASH1L's effects depend on its histone methyltransferase activity.

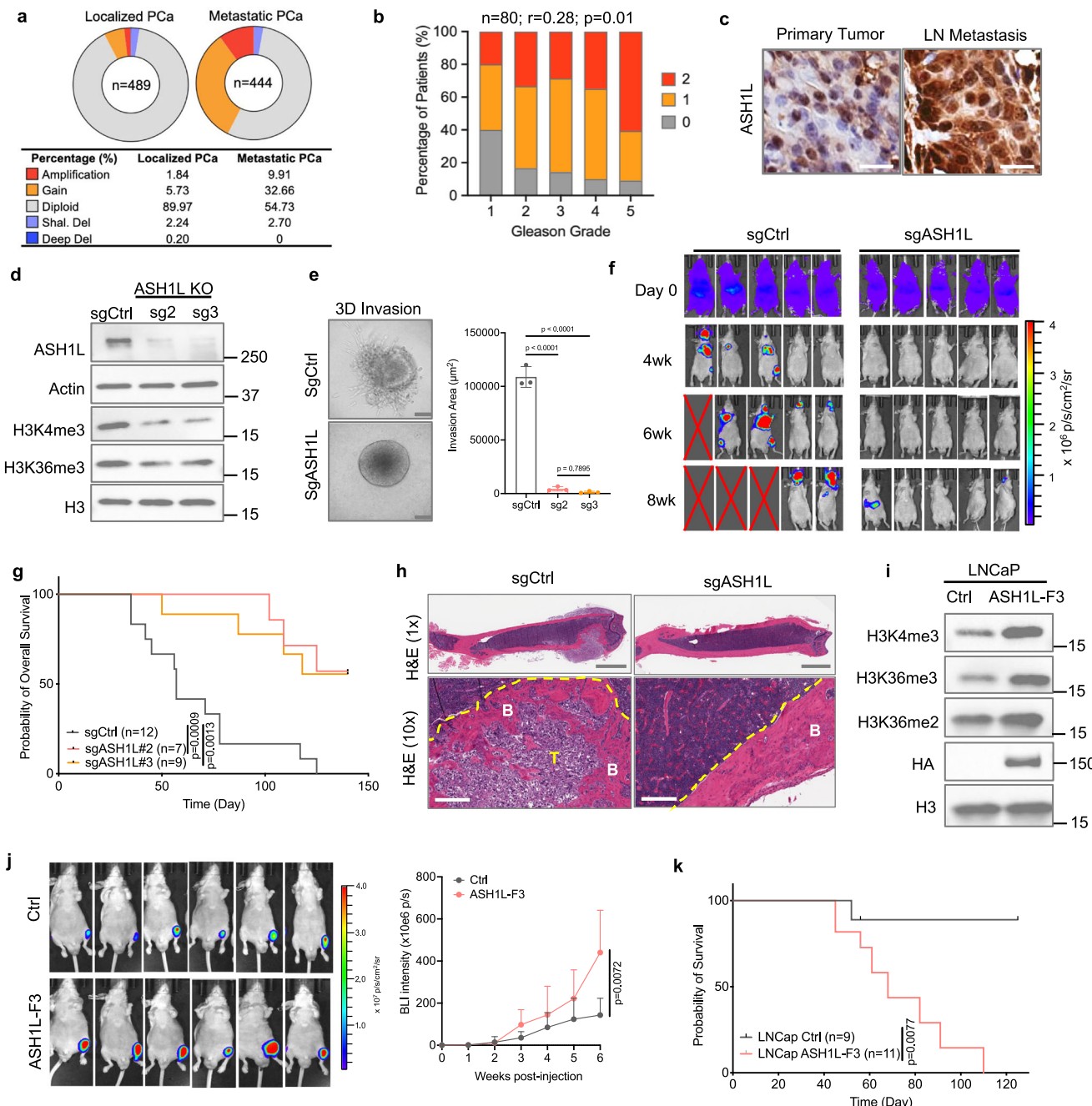

**Fig. 1 | ASH1L promotes cancer invasiveness and bone metastasis. a** The percentages of genetic alterations of *ASH1L* in human localized (TCGA) and metastatic (SU2C) PCa samples. **b** Correlation of ASH1L protein expression with Gleason grade in human prostate tumors. The staining intensity of nuclear ASH1L was scored from 0 to 2. The Pearson correlation coefficient (*r*) and two-tailed *P* value are shown. **c** IHC staining of ASH1L in primary and lymph node (LN) metastatic tumors derived from *Pb-Cre; Pten^{L/L}; Trp53^{L/L}; Smad4^{L/L}; mTmG (PbPPS)* mice. Scale bar = 20 μm. **d** Western blot analysis of the indicated proteins in control and ASH1L-depleted PC-3M cells. Two sgRNAs were used for CRISPR-Cas9-mediated *ASH1L* knockout. The samples were derived from the same experiment, but different gels for ASH1L and Actin, another for H3, another for H3K4me3, and another for H3K36me3 were processed in parallel. **e** 3D sphere invasion assays of control and ASH1L-depleted PC-3M cells (*n* = 3 biological replicates/group). Scale bar = 100 μm. **f** Control and ASH1L-depleted PC-3M cells expressing firefly luciferase and RFP were intracardially injected into male nude mice. In vivo bioluminescence images of mice over time are shown. **g** Kaplan–Meier analysis of the overall survival of mice. The log-

rank (Mantel-Cox) test was used for statistical analysis. *n* = 12, 7, and 9 mice per group, as indicated. **h** H&E staining of bone tissues from mice transplanted with control and ASH1L-depletion PC-3M cells. Scale bar = 2 mm (upper), and 300 μm (bottom). T Tumor, B Bone. **i** Western blot analysis of indicated histone marks in LNCaP cells overexpressing HA-tagged ASH1L-F3. The samples were derived from the same experiment, but different gels for HA, another for H3, another for H3K4me3, another for H3K36me2, and another for H3K36me3 were processed in parallel. **j** Control and ASH1L-F3-overexpressed LNCaP cells (Luc-RFP+) were injected into the tibias of one leg of nude male mice at 6 weeks of age. Bioluminescence images and BLI intensity quantification of bone tumors over time are shown (*n* = 6 mice per group). **k** Kaplan–Meier analysis of overall survival of mice. The log-rank (Mantel-Cox) test was used for statistical analysis. Statistical significance was determined by unpaired two-tailed *T*-test (**j**) or one-way ANOVA with Tukey's post hoc test (**e**). Data in **e**, **j** represent the mean ± standard deviation. The experiments in **d**, **i** were repeated independently three times, yielding similar results. Source data are provided as a Source Data file.

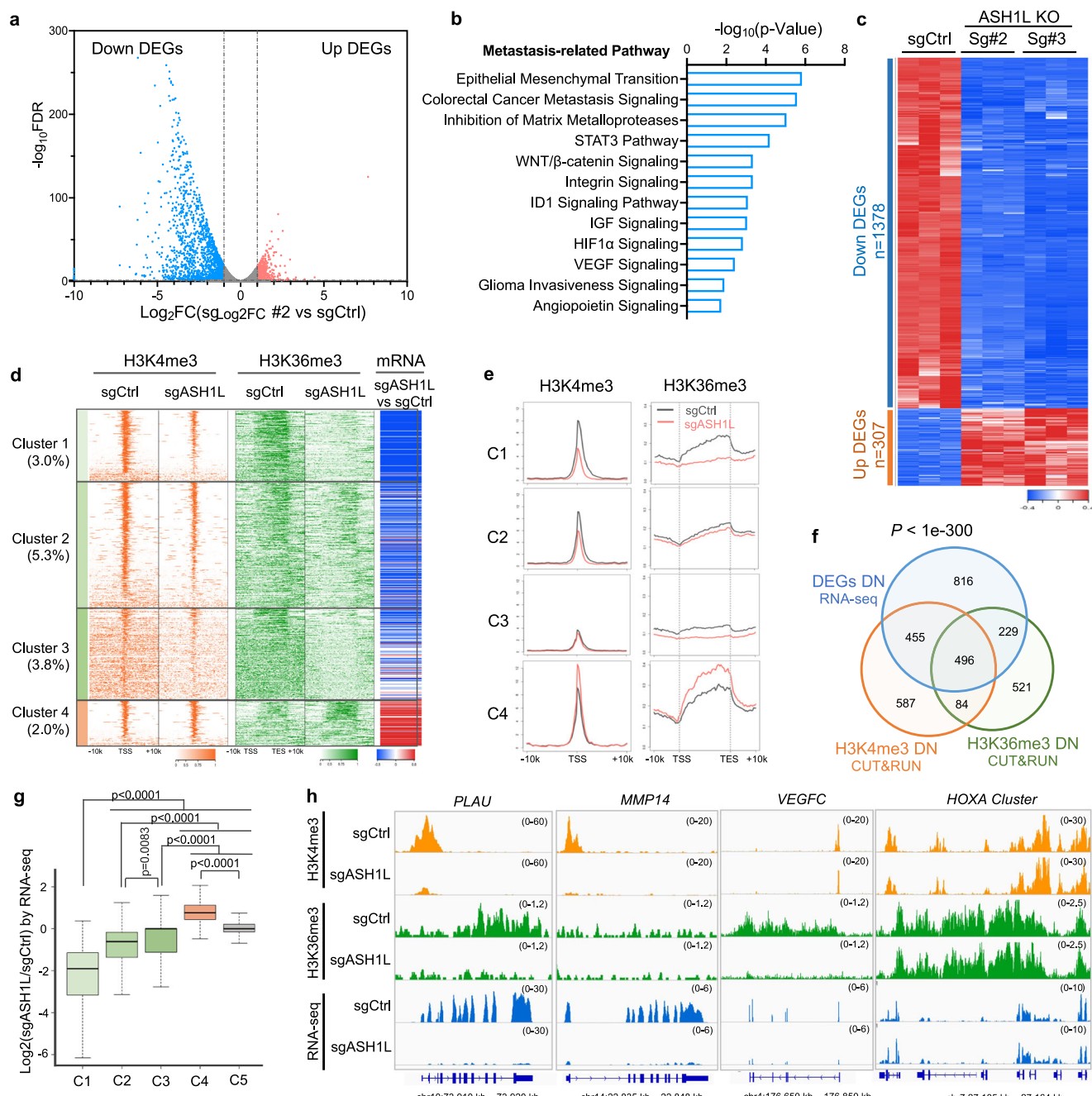

**Fig. 2 | ASH1L reprograms pro-metastatic transcriptome via modulating methylations at H3K4 and H3K36. a** Volcano plot illustrating differentially expressed genes (DEGs) in ASH1L-depleted (sg#2) versus control PC-3M. Blue: downregulated genes upon ASH1L-depletion; Red: upregulated genes. Horizontal line: FDR = 0.05; Vertical line: fold change (FC) = 2. **b** Pathway analysis of DEGs using IPA. *P* values were determined by one-sided Fisher's exact test. **c** Heatmap displaying the relative expression of DEGs in control (sgCtrl) versus ASH1L-knockout PC-3M cells mediated by two sgRNAs (sg#2 and sg#3). **d** Heatmaps illustrating the signal of H3K4me3 (orange) and H3K36me3 (green) marks over Transcription start sites (TSS) or body of genes in control and ASH1L-depleted PC-3M cells, determined by CUT&RUN. Genes were grouped into five clusters (C1–C5) by changes in H3K4me3 and H3K36me3 signals upon ASH1L depletion. % of genes in each cluster is shown. The right column, color-coded in red (upregulated) or blue (down-regulated), represents the log2FC of gene expression in sgASH1L#2 versus sgCtrl cells (RNA-seq). **e** Metaplots illustrating the average signal distribution of H3K4me3

(left) and H3K36me3 (right) over genes in C1-C4. The signals in control (gray) and ASH1L-depleted (red) PC-3M cells are compared. TSS and transcript end sites (TES) are indicated. **f** Venn diagram displaying the overlap of genes with decreased H3K4me3 (FDR ≤0.05; FC ≥1.5), genes with decreased H3K36me3 (FDR ≤0.05; FC ≥1.5), and expression downregulated genes (both sgRNAs; FDR ≤0.05; FC ≥1.5) upon ASH1L depletion. *P* values were calculated using two-sided Fisher's exact test. **g** Box plots illustrating the log2FC of gene expression in sgASH1L#2 versus sgCtrl cells for the five clusters. Each box represents the interquartile range (IQR; 25th to 75th percentile). Centerline indicates median; Whiskers extend to the most extreme data points within 1.5×IQR. *P* values were calculated using two-sided unpaired Welch's *t*-tests. *n* = 580 genes (C1); *n* = 1042 genes (C2); *n* = 750 genes (C3); *n* = 384 genes; *n* = 16875 genes (C5). **h** Representative CUT&RUN tracks illustrating the enrichments of H3K4me3 (orange) and H3K36me3 (green) at the indicated gene loci in control and ASH1L-depleted PC-3M cells, with corresponding RNA-Seq tracks (blue) for the indicated genes. Source data are provided as a Source Data file.

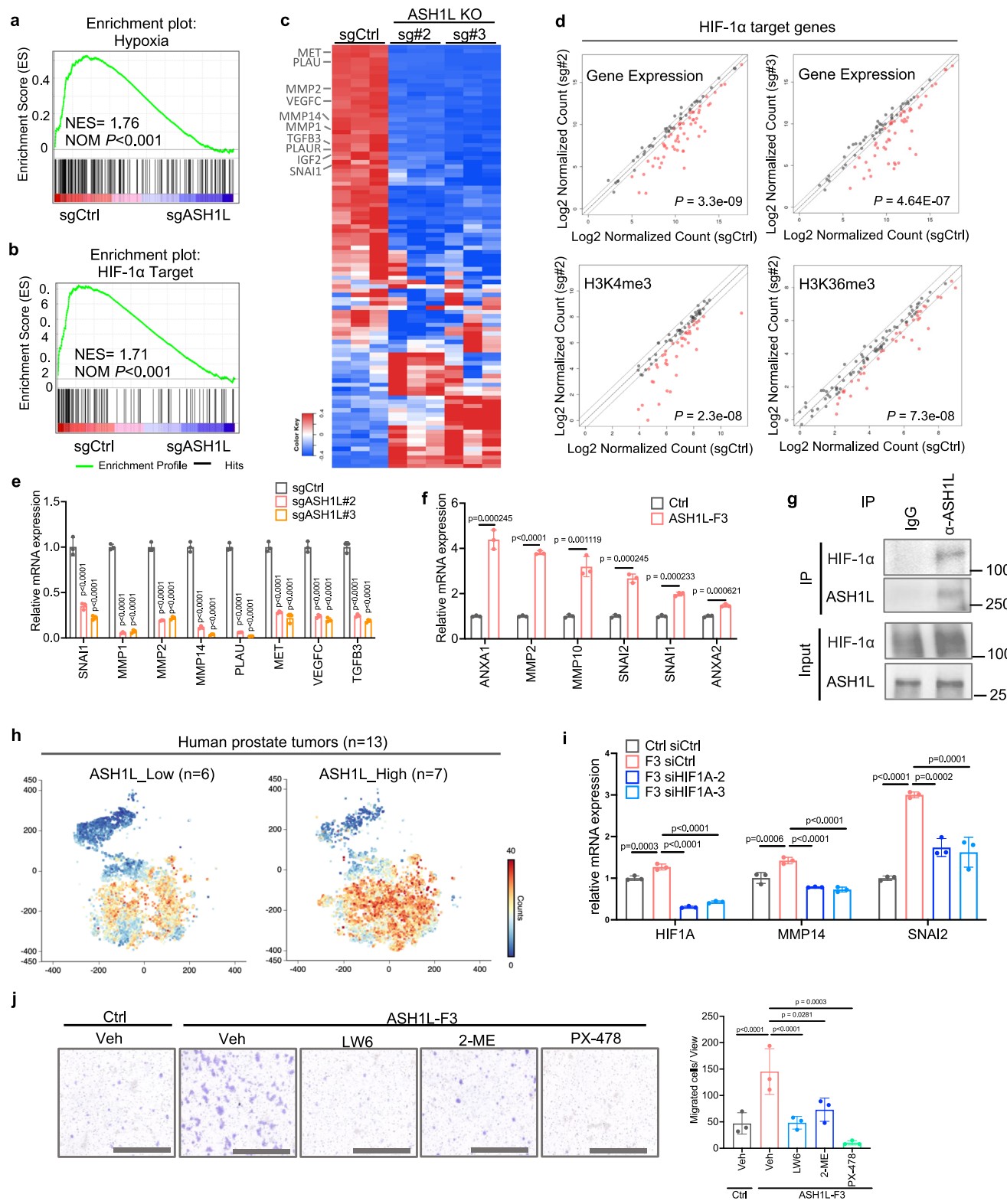

Importantly, endogenous co-immunoprecipitation (co-IP) and cell-free protein pulldown assays revealed that ASH1L-F3 directly interacted with the HIF-1α protein (Fig. 3g and Supplementary Fig. 5e), reinforcing the notion that ASH1L and HIF-1α function as a complex in metastatic PCa cells. Then, we analyzed the published bulk RNA-seq and single-cell transcriptomic datasets from human primary and metastatic prostate tumor samples[40,41]. The results showed that, among HIF family members, HIF-1α most significantly co-expressed with ASH1L in epithelial components (Supplementary Fig. 5f–h).

Compared to the ASH1L low-expressing group, PCa tumors expressing high ASH1L levels exhibited elevated HIF-1α transcriptome in epithelial components (Fig. 3h and Supplementary Fig. 5i).

To determine the role of HIF-1α in pro-metastatic gene transcription and invasiveness driven by ASH1L, we inhibited HIF-1α using siRNAs or small molecular compounds in LNCaP cells expressing ASH1L-F3. The results showed that, compared to control cells, ASH1L-F3 overexpression caused pro-metastatic gene upregulation under the hypoxia condition, but this effect was abolished by *HIF1A* knocking

**Fig. 3 | ASH1L co-opts with HIF-1α to induce pro-metastatic genes and enhance invasiveness. a, b** GSEA demonstrating the downregulation of hypoxia-associated genes and HIF-1α target genes upon ASH1L depletion. The normalized enrichment score (NES) and nominal *p* value (NOM *P*) are indicated. **c** Heatmap displaying the relative expression of HIF-1α target genes in control versus ASH1L-depleted PC-3M. **d** Scatter plots demonstrating the changes in gene expression, H3K4me3, and H3K36me3 marks of HIF-1α target genes upon ASH1L depletion. Genes with downregulated signals (FDR ≤0.05) are highlighted in red. Dotted lines: FC = 1.5. **e** Expression of metastasis-associated HIF-1α targets in control and ASH1L-depleted PC-3M, determined by qPCR. *n* = 3 biological replicates per group. **f** LNCaP cells overexpressing ASH1L-F3 were treated with CoCl2 (200uM) for 24 h to mimic hypoxia conditions. mRNA expression of the indicated genes was determined by qPCR. *n* = 3 biological replicates per group. **g** Endogenous co-IP of ASH1L and HIF-1α in PC-3M cells treated with 20 μM MG132 for 8 h, followed by Western blot analysis. The samples were derived from the same experiment, but different gels for ASH1L and another for HIF1A were processed in parallel. **h** tSNE views of 23,651 epithelial cells from human PCa samples (GSE141445; *n* = 13 patients), color-coded by the count of HIF-1α target genes. 13 PCa patients were classified into two groups (ASH1L_Low vs ASH1L_High) by ASH1L expression levels in epithelial cells. **i** Control or ASH1L-F3-overexpressing LNCaP cells were transfected with siRNA targeting HIF1A in the presence of CoCl2 (200uM). mRNA expression of the indicated genes was determined. *n* = 3 biological replicates per group. **j** Control and ASH1L-F3-overexpressing LNCaP cells were treated with Vehicle (Veh) or HIF-1α inhibitors LW6 (15 μM), 2-MeOE2 (2-ME, 25 μM), or PX-478 (20 μM) for 48 h in the presence of CoCl2 (200uM), followed by migration assays. *n* = 3 biological replicates per group. Scale bar = 1000 μm. Statistical significance was determined by unpaired two-tailed *T*-test (**f**) or one-way ANOVA with Tukey's post hoc test (**e, i, j**). Data in **e, i, j** represent the mean ± standard deviation. The experiments in (**g**) were repeated independently three times, yielding similar results. Source data are provided as a Source Data file.

down (Fig. 3i). Along the same line, targeting HIF-1α with small-molecule inhibitors (LW6, 2-MeOE2, and PX-478) significantly impaired LNCaP cell migration driven by ASH1L-F3 overexpression (Fig. 3j). In contrast, the HIF-2α inhibitor (PT2399) showed minimal effects on cell migration (Supplementary Fig. 5j).

Collectively, our mechanistic studies demonstrate that histone methyltransferase ASH1L interacts with transcriptional factor HIF-1α to activate the transcription of pro-metastatic genes and enhance the invasiveness of metastatic cells. It is worth noting that ASH1L also co-localized with HIF-1α on the promoter regions of genes in leukemia cells (Supplementary Fig. 5k, l and Supplementary Data 5); however, only 32.5% (1132 in 3482) of ASH1L target genes were shared with HIF-1α, which are distinct from those in metastatic PCa (Supplementary Fig. 5k, l and Supplementary Data 5). It suggests that ASH1L/HIF-1α interaction is a universal mechanism in regulating gene expression, but their target genes may vary in different contexts.

**Characterize ASH1L's role in the immunocompetent context**

Prior studies showed that HIF signaling plays a vital role in modulating stromal and immune components in primary and metastatic tumors[38,42,43]. Given that ASH1L promotes bone metastasis and induces HIF-1α transcriptome, we hypothesize that, in addition to reprogramming pro-metastatic genes in invading cancer cells, ASH1L has a cell-extrinsic role of remodeling the bone niche. Previously, we reported a syngeneic PCa cell line (e.g., DX1)[44,45] that was derived from the *PbPPS* metastatic PCa GEM model (Supplementary Fig. 1g). To mimic PCa metastatic outgrowth in bone, we generated a syngeneic mouse model by intratibial injection of DX1 cells into male C57BL/6 J mice (Fig. 4a). Two to three weeks after inoculation, DX1 cells formed outgrown tumors in the bone with a mixed osteolytic/osteoblastic feature (Fig. 4b, c and Supplementary Fig. 6a), which recapitulated bone metastasis in patients with PCa[43]. Histopathological analysis revealed significant absorption of cortical bone and remarkable new woven bone production in the outgrown tumors, along with enriched osteoclasts and osteoblasts (Fig. 4c–g and Supplementary Fig. 6a).

To determine ASH1L's impact on metastatic outgrowth and the bone niche in an immunocompetent setting, we genetically knocked out ASH1L in DX1 cells using the sgRNA-CRISPR-Cas9 system (Supplementary Fig. 6b) and then injected control and ASH1L-depleted cells expressing firefly luciferase into one tibia of male C57BL/6 J mice (Fig. 4a). The IVIS and ex vivo GFP-fluorescence imaging showed that genetic knockout of *ASH1L* in cancer cells significantly suppressed tumor outgrowth in the bone (Fig. 4b and Supplementary Fig. 6c). X-ray imaging and histopathology analysis revealed that ASH1L depletion in invading cancer cells protected cortical bone from absorption and reduced new woven bone production (Fig. 4c–e and Supplementary Fig. 6d). Both osteoclasts and osteoblasts were decreased in bone-

outgrown tumors upon ASH1L deletion (Fig. 4c, f, g). Similar effects were observed when knocking down ASH1L using shRNA (Supplementary Fig. 6e–k). Together with our findings in metastasis xenograft models, these results in the syngeneic model establish ASH1L as an essential epigenetic determinant in metastatic tumor outgrowth in the bone.

**Single-cell transcriptome profiling reveals ASH1L induces macrophage plasticity in metastatic bone niche**

To deconvolute the impact of ASH1L on invading cancer cells and metastatic bone niches, we performed single-cell RNA sequencing (scRNA-seq) in control (*n* = 3) and ASH1L-depleted (*n* = 4) syngeneic bone tumors. After standard data processing and quality control procedures (Methods), we obtained transcriptomic profiles for 30,850 single cells. Clustering analysis identified ten major clusters of 760 to 6999 cells each (AC1-AC10; Supplementary Fig. 6l). Cells from control and ASH1L-depleted bone tumors were distributed in all ten clusters, and each cluster contained cells from seven samples (Supplementary Fig. 6m). We characterized the cell identity of each cluster by integrating differential gene expression analysis and well-known cell lineage-specific markers (Supplementary Fig. S6l, n). Among those, fibroblast cluster AC3 highly expressed Col1a2, while endothelial cell cluster AC4 exhibited Pecam1. Clusters AC5-AC10 expressed high levels of immune cell lineage makers, recapitulating the immune landscape in human samples. Both AC1 and AC2 showed high levels of luminal epithelial cell marker Krt8, indicating they are cancer cell clusters. However, only the AC1 cluster expressed a high level of HIF-1α transcriptome, and this feature was remarkably reduced in ASH1L-depleted tumors (Supplementary Fig. 6o), strengthening the crucial role of ASH1L in regulating HIF-1α transcriptome in invading cancer cells.

To visualize the effects of ASH1L depletion on immune components of the bone niche, we subclustered 17,423 immune cells. As a result, we identified nine distinct cell subclusters that represent major immune cell types (Fig. 4h–l and Supplementary Data 6): T/NK (C1: Cd3e$^{high}$/Cd3d$^{high}$/Nkg7$^{high}$), B/Plasma (C2: Cd79a$^{high}$/Igkc$^{high}$), Monocyte (C3: Itgam$^+$/F13a1$^{high}$/Ly6c2$^{high}$), Tumor-associated Macrophage (TAM) (C4: Itgam$^+$/Csf1r$^{high}$/Cd68$^{high}$/Mrc1$^{high}$/Trem2$^{high}$), Dendritic Cell (C5: H2-Aa$^{high}$/Tcf4$^{high}$/Runx3$^{high}$), Pre-neutrophil (C6: Itgam$^+$/Mpo$^{high}$/Elane$^{high}$), MDSC_1 (C7: Itgam$^+$/S100a8$^{high}$/Cxcr2$^{high}$/Ly6c2$^{low}$), and MDSC_2 (C8: Itgam$^+$/S100a8$^{high}$/Ly6c2$^{high}$/Cxcr2$^{mid}$). Interestingly, we found that ASH1L depletion augmented the infiltration of B cells (C2) and monocytes (C3), but caused a remarkable reduction of TAM (C4) (Fig. 4j–l), suggesting ASH1L is involved in modulating monocytic myeloid cells in the metastatic bone niche.

Prior studies have demonstrated that tumor-infiltrating monocytes and TAMs are composed of diverse subtypes in the tumor microenvironment and exhibit distinct functions during cancer

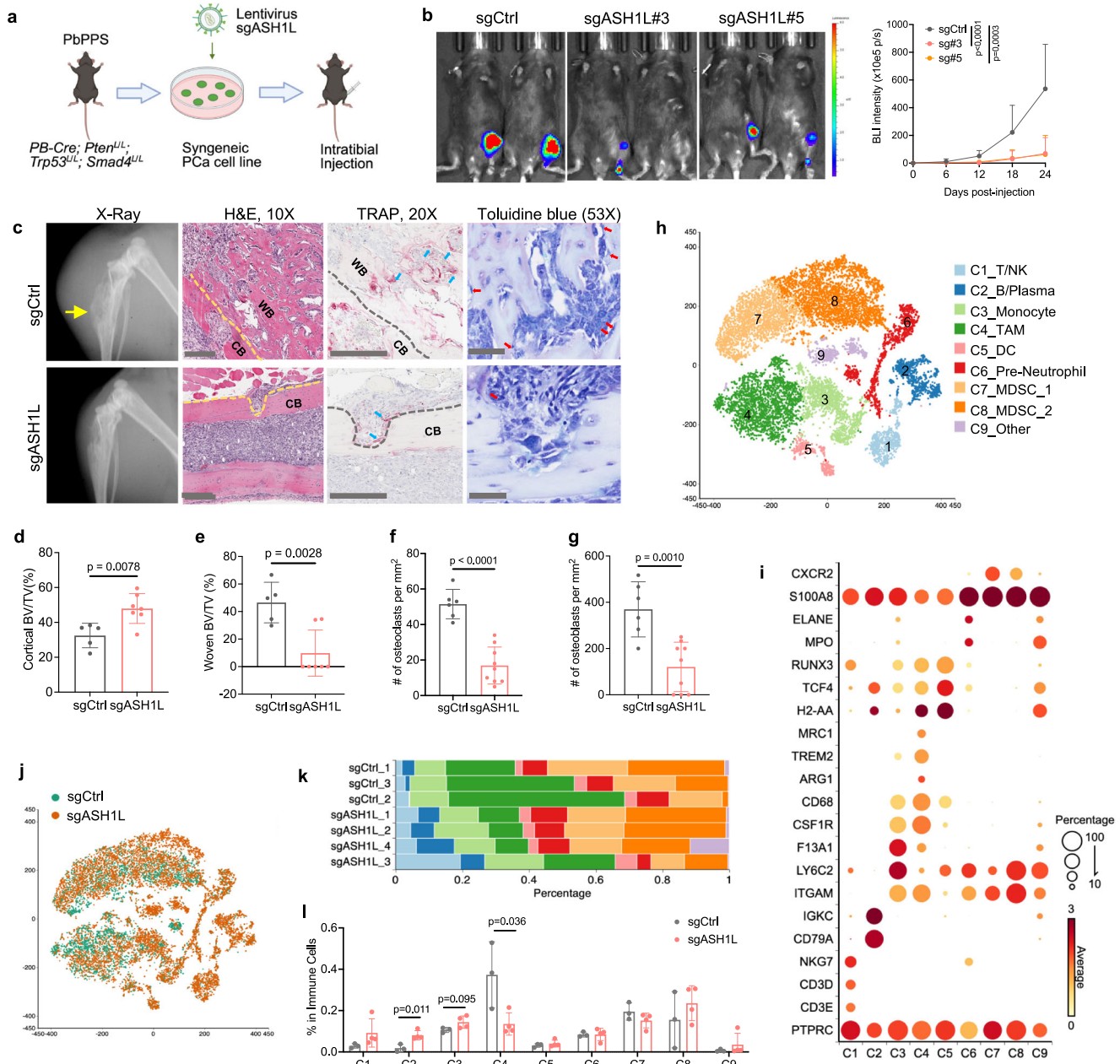

**Fig. 4 | Loss of ASH1L impairs bone tumor outgrowth and bone niche remodeling. a** Schematics of DX1-derived bone metastasis model. Control or ASH1L-depleted DX1 cells expressing firefly luciferase were injected into the tibia of one leg of C57BL/6 J male mice. Created in BioRender. Meng, C. (2025) https://BioRender. com/wzb1hcr. **b** Bioluminescence images and BLI intensity quantification of bone tumors over time are shown. $n = 15$ mice (sgCtrl), 10 mice (sg#3) or 7 mice (sg#5). **c** Representative images of X-Ray imaging, H&E, Tartrate-resistant acid phosphatase (TRAP), and Toluidine blue staining of bone tumors. Scale bar = 300 μm for H&E; Scale bar = 200 μm for TRAP; Scale bar = 50 μm for Toluidine blue. CB cortical bone, WB woven bone. **d, e** Quantification of cortical bone (**d**) and new woven bone (**e**) volume fraction from control ($n = 5$ mice) and ASH1L-depleted ($n = 7$ mice) bone tumors. Cortical BV/TV (%): percentage of cortical bone volume in the entire tibia tissue area; Woven BV/TV (%): percentage of new woven bone volume in bone formation tissue area (see more details in Fig. S6a). **f, g** Quantification of osteoclasts (**f**) and osteoblasts (**g**) in new woven bone tissues from control ($n = 6$ mice) and ASH1L-depleted ($n = 9$ mice) bone tumors. Blue and red arrows indicate osteoclasts and osteoblasts, respectively. **h–l** ScRNA-seq was performed in control ($n = 3$ mice) and ASH1L-depleted ($n = 4$ mice) bone tumors. PTPRC+ immune cells were subclustered and analyzed. tSNE views of 17,423 immune cells color-coded by nine subclusters (C1-C9, **h**) or two groups (**j**) are presented. The bubble plot presents marker gene expression for each immune cell subcluster (**i**), where dot size and color represent the percentage of marker gene expression (Percentage) and the averaged scaled expression (Average) value, respectively. Immune cell composition distribution (**k**) and percentage comparison (**l**) of nine subclusters among seven samples. TAM tumor-associated macrophage, DC dendritic cell. Statistical significance was determined by unpaired two-tailed *T*-test (**d–g, l**) or one-way ANOVA with Tukey's post hoc test (**b**). Data in **b, d–g, l** represent the mean ± standard deviation. Source data are provided as a Source Data file.

progression[7,8,10,12,46,47]. To characterize their subtypes and activation states in metastatic bone niches and determine ASH1L's impact, we performed clustering and differential gene expression analyses of monocytes (C3) and TAMs (C4) in control and ASH1L-depleted bone tumors. As a result, we identified seven distinct subclusters and

nominated them as classical monocyte (MC1), nonclassical monocyte (MC2), monocyte/TAM Intermediate (MC3), angiogenic TAMs (MC4), lipid-associated TAM (MC5), inflammatory TAM (MC6), and proliferating TAMs (MC7) (Fig. 5a, b and Supplementary Data 7). Cells from control and ASH1L-depleted tumors were distributed in all seven

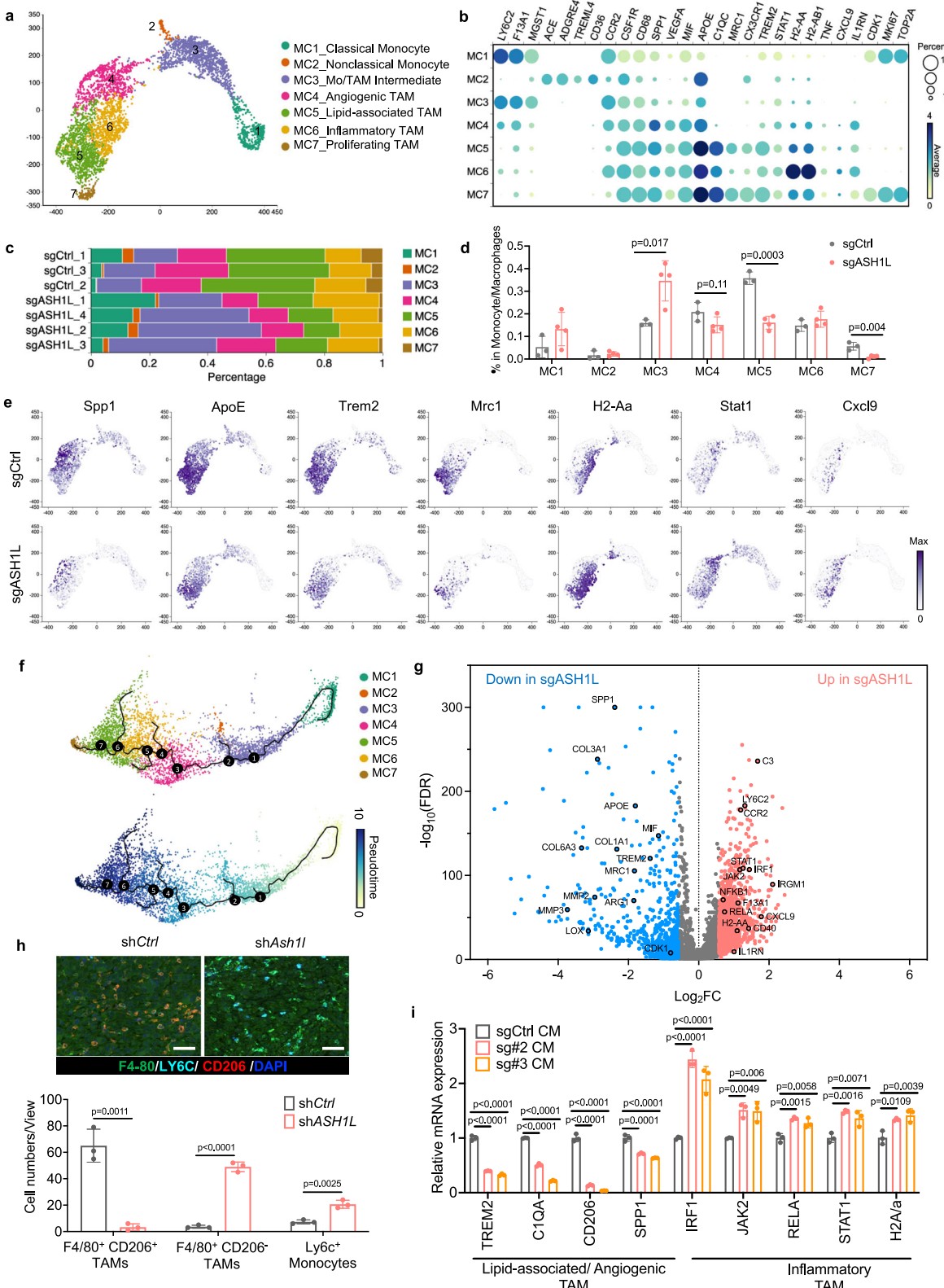

clusters with different compositions (Fig. 5c, d, and Supplementary Fig. 7a).

Among those subclusters, both MC1 and MC2 are monocytes. Compared to MC2, the MC1 cluster is relatively dominant in the bone niche and expresses classical monocyte markers, such as *Ly6c2, F13a1, Mgst1*, and *Ccr2*. MC2 only comprises 1–2% of total monocyte/TAMs and exhibits nonclassical monocyte marker genes, such as *Ace, Adgre, Treml4*, and *Cd36* (Fig. 5a–d). Subclusters MC4-MC7 express

macrophage lineage markers (*Csf1r* and *Cd68*) but represent distinct transcriptomic features and activation states (Fig. 5a, b, e, Supplementary Fig. 7b–d, and Supplementary Data 7). The MC4 subcluster highly expresses pro-angiogenesis genes, including *Spp1, Vegfa, Arg1*, and *Mif*, and thus was nominated as Angiogenic TAMs. Cells in MC5 express high-level lipid-associated TAM markers (*Apoe, C1qc*, and *Trem2*) and pro-tumoral marker *Mrc1*. MC7 shares most markers with lipid-associated TAM (MC5) but presents a high proliferating rate,

**Fig. 5 | Single-cell transcriptome profiling reveals ASH1L induces macrophage plasticity in metastatic bone niche. a** The UMAP view of 3046 Monocytes and TAMs color-coded by seven subclusters (MC1-MC7) is presented with the nomination of each subcluster. **b** The bubble plot presents marker gene expression for each subcluster, where dot size and color represent the percentage of marker gene expression (Percentage) and the averaged scaled expression (Average) value, respectively. **c, d** Composition distribution (**c**) and percentage comparison (**d**) of seven monocyte/TAM subclusters among samples. *n* = 3 mice for control group, *n* = 4 mice for ASH1L-depleted group. **e** UMAP views of TAMs from control versus ASH1L-depleted bone tumors, color-coded by expression of indicated genes. **f** Single-cell trajectory and pseudotime analysis of monocytes and TAMs. The UMAP views are color-coded by seven subclusters (left) and pseudotime (right). **g** Volcano plot of the differentially expressed genes (DEGs, FDR <0.05) in monocytes and TAMs in ASH1L-depleted versus control tumors. Blue dots indicate the

downregulated genes in TAMs upon ASH1L depletion in DX-derived bone tumors, while red dots indicate the upregulated genes (fold change [FC] >1.5). Gray dots present the genes FC <1.5. **h** Representative images (enlarged) and quantifications of pro-tumoral TAMs (F4-80⁺CD206⁺), pro-inflammatory TAMs (F4-80⁺CD206⁻), and monocytes (LY6C⁺) in control and ASH1L-knockdown metastatic bone tumors, determined by multiplex IHC staining. Scale bar = 50 μm. *n* = 3 mice per group. **i** Human monocytes were isolated from blood, induced by M-CSF (50 ng/ml) for 5 days, and then co-cultured with conditional medium (CM) derived from control and ASH1L-depleted PC-3M cells for 48 h. Marker gene expression was analyzed using qPCR. GAPDH was used as a loading control. *n* = 3 biological replicates per group. Statistical significance was determined by unpaired two-tailed *T*-test (**d**, **h**) or one-way ANOVA with Tukey's post hoc test (**i**). Data in **d**, **h**, **i** represent the mean ± standard deviation. Source data are provided as a Source Data file.

characterized by *Mki67, Cdk1*, and *Top2a*. The subcluster MC6 exhibits enriched antigen presentation molecules (*H2-aa, H2-ab1*, and *Cd40*), interferon signaling (*IL1rn, Jak2, Stat1, Irf1*, and *Nfkb1*), and cytokines involved in T cell recruitment and activation (*Cxcl9* and *Tnf*). Notably, we also identified a subcluster, MC3, with mixed expression of macrophage markers (*Csf1r* and *Cd68*) and monocyte markers (*F13a1* and *Ly6c2*). Trajectory and pseudotime analyses revealed that classical monocytes in the MC1 serve as the lineage progenitor of the other six subclusters, and cells in the MC3 are in an intermediate status during monocyte differentiation to TAMs (Fig. 5f).

Compared to the control group, the ASH1L-depleted bone tumors showed a remarkable reduction in lipid-associated TAM (MC5), proliferating TAMs (MC7), and angiogenic TAMs (M4), along with a dramatic increase in monocyte/TAM intermediate cells (MC3). Besides, we observed a slight increase in classical monocytes (MC1). Differential gene expression analysis revealed that, in addition to TAM markers, numerous pro-tumoral/metastatic genes and immunosuppressive molecules were remarkably downregulated in monocytes and TAMs upon ASH1L depletion in metastatic cells (Fig. 5e, g, Supplementary Fig. 7b, d, and Supplementary Data 8). Even though the percentage of inflammatory TAM (MC6) only had a modest change (Fig. 5d), genes that are involved in antigen presentation, interferon signaling, and inflammation were significantly upregulated in TAMs upon ASH1L depletion (Fig. 5e, g, Supplementary Fig. 7c, and Supplementary Data 8). These results indicated that ASH1L in invading cancer cells contributes to the lipid-associated, angiogenic, and anti-inflammatory states of TAMs in the bone niche.

To further visualize the spatial arrangement of monocytes and TAMs in outgrown bone tumors, we performed multiplex IHC staining in control and ASH1L-depleted tumors using monocyte marker Ly6c, total TAM marker F4/80, and pro-tumoral TAM marker CD206 (encoded by the *Mrc1* gene). Consistent with single-cell transcriptome analysis, inhibition of ASH1L in cancer cells dramatically decreased TAMs with pro-tumoral phenotype (F4/80⁺CD206⁺) and led to an increase in tumor-infiltrating monocytes (Ly6c⁺) in the bone niche (Fig. 5h and Supplementary Fig. 7e). Reversely, overexpression of ASH1L in cancer cells suppressed monocytes but promoted pro-tumoral TAMs in bone tumors (Supplementary Fig. 7f). To verify the direct interaction between metastatic cells and monocytes, we isolated monocytes from healthy human blood samples and co-cultured them in conditioned media derived from PC-3M cells with or without *ASH1L* knockout (Fig. 5i). Compared to the control group, conditioned medium from ASH1L-depleted cells failed to induce monocytes to express lipid-associated/angiogenic TAM markers, but facilitated them to express higher levels of inflammatory TAM markers (Fig. 5i). Similar effects were also observed in the human monocyte cell line, THP-1 (Supplementary Fig. 7g).

Collectively, our comprehensive single-cell transcriptomic analyses and functional validation suggest that ASH1L in invading cancer cells reshapes metastatic bone niches by promoting monocyte

differentiation into lipid-associated and angiogenic TAMs and reprogramming them toward pro-tumoral and anti-inflammatory states.

## ASH1L induces lipid-associated TAMs by promoting IGF-2-mediated oxidative phosphorylation

To understand the mechanisms of how ASH1L modulates macrophage differentiation and plasticity, we performed pathway analysis of differentially expressed genes in TAM subclusters (MC4-MC7). Compared to that of control tumors, TAMs in ASH1L-depleted bone tumors showed significant elevation in inflammation pathways, including IFN signaling, TNF signaling, IL-1/2 signaling, cGAS-STING, and antigen processing and presentation pathway (Fig. 6a). Besides, genes involved in senescence and pyroptosis pathways were significantly upregulated in TAMs upon ASH1L depletion in bone tumors, whereas cell cycle and mitosis-related genes were downregulated (Fig. 6a). It indicates that ASH1L in invading cancer cells contributes to the proliferation, survival, and anti-inflammatory phenotype of TAMs.

Interestingly, we found that depletion of ASH1L in metastatic cells led to decreased oxidative phosphorylation (OXPHOS), tricarboxylic acid (TCA) cycle, fatty acid β-oxidation (FAO), and ATP synthesis (Fig. 6a). The signature genes in OXPHOS are highly enriched in TAMs in metastatic bone tumors (Fig. 6b), consistent with prior studies in primary tumors and other metastatic sites[48,49]. However, we found that loss of ASH1L in metastatic cells led to a remarkable reduction of OXPHOS genes in lipid-associated TAMs (Fig. 6b). These results indicated that ASH1L plays a key role in rewiring macrophage metabolism in the metastatic bone niche, facilitating lipid-associated TAM induction and their pro-tumoral phenotype.

As shown in Fig. 5i and Supplementary Fig. 7g, conditioned media derived from ASH1L-depleted metastatic cells showed impaired capacity to induce human monocyte differentiation into lipid-associated/angiogenic TAMs, suggesting secretory factor(s) mediate ASH1L regulation on TAMs. To identify this factor, we performed an integrating analysis using our RNA-seq and CUT&RUN-seq datasets (Fig. 3 and Supplementary Data 1–4) and identified five TAM-associated secretory protein candidates that are co-target genes of ASH1L/HIF-1α and significantly downregulated in PC-3M cells upon ASH1L depletion (Fig. 6c). Among them, only insulin-like growth factor 2 (IGF-2) is linked to macrophage metabolism[50]. Pathway analysis in Fig. 6a also indicates that TAMs from ASH1L-depleted bone tumors exhibited decreased IGF downstream signaling, i.e., AKT and MAPK pathways, suggesting IGF signaling may be involved in the TAM plasticity and metabolic programming driven by ASH1L.

By binding to the receptors, IGF family members (IGF-1 and IGF-2) play an essential role in cell proliferation, survival, protein translation, and metabolism[51]. In metastatic PC-3M cells, the *IGF-2* promoter was strongly marked by H3K4me3, but the signal was completely abolished upon *ASH1L* deletion (Fig. 6d). However, no H3K36me3 signal across the gene body of *IGF-2* was observed in either control or ASH1L-depleted cells (Fig. 6d), suggesting that ASH1L regulates IGF-2

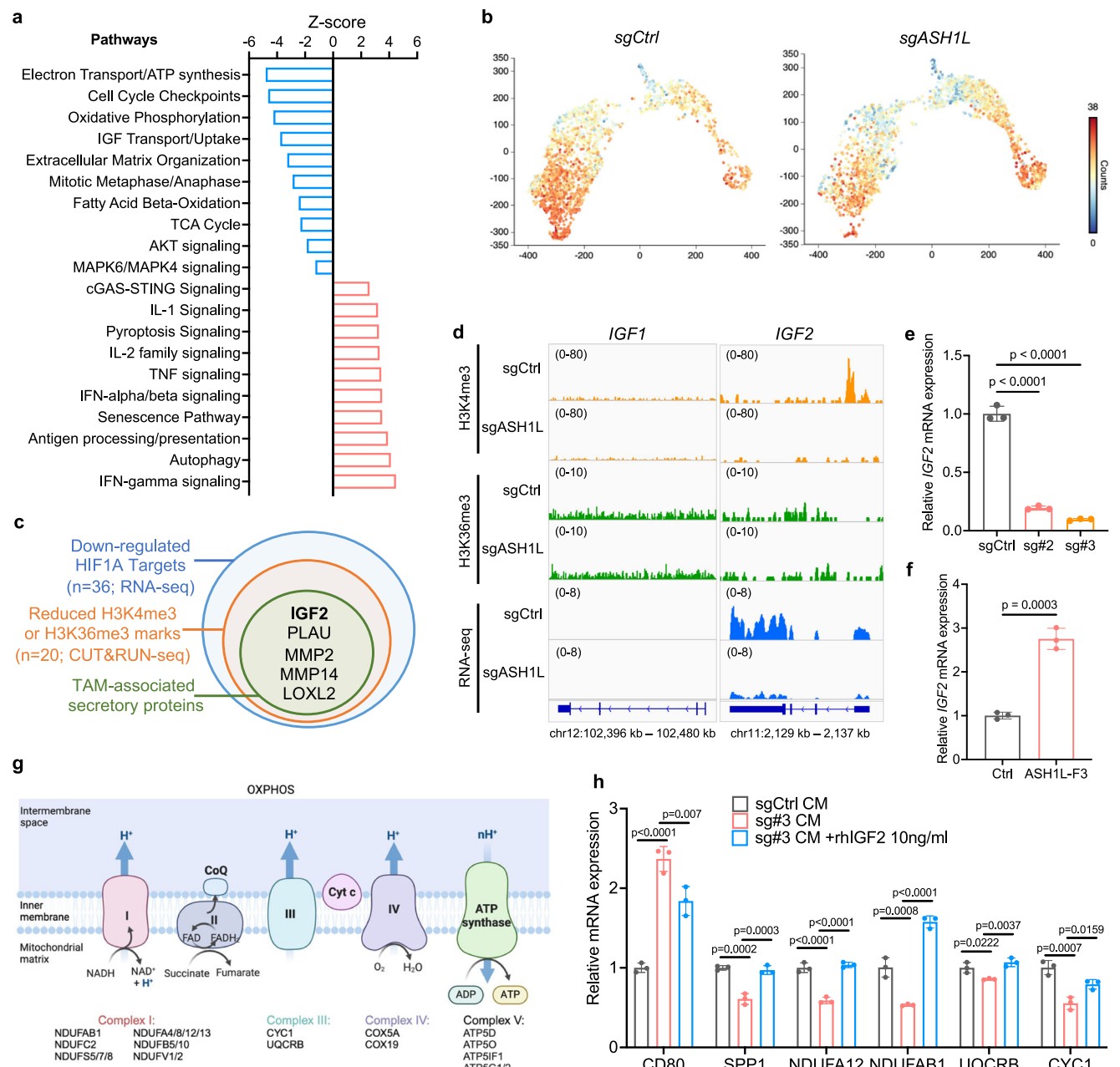

**Fig. 6 | ASH1L induces lipid-associated TAMs by promoting IGF-2-mediated oxidative phosphorylation. a** Pathway analysis of DEGs (FDR <0.05; FC >1.5) in TAMs in ASH1L-depleted versus control tumors using IPA. The activation Z-scores indicate the activation states of biological functions of each pathway. Blue bars indicate the downregulated pathways upon ASH1L depletion; Red bars indicate the upregulated pathways. **b** UMAP views of monocytes and TAMs from control versus ASH1L-depleted bone tumors, color-coded by the count of signature genes in oxidative phosphorylation (OXPHOS) pathways. **c** Venn diagram displaying the HIF-1α direct target genes that are significantly downregulated in PC-3M upon ASH1L depletion (Blue circle, RNA-seq FDR <0.05; FC >2), exhibit decreased H3K4me3 and/or H3K36me3 marks upon ASH1L depletion (Orange circle, CUT&RUN FDR ≤0.05; FC ≥1.5), and associated with TAM function modulation (Green circle). **d** Representative CUT&RUN tracks illustrating the enrichments of H3K4me3 (orange) and H3K36me3 (green) at IGF-1 and IGF-2 gene loci in control and ASH1L-depleted PC-3M cells, with corresponding RNA-Seq tracks (blue). **e, f** mRNA expression of IGF-2 in ASH1L-depleted PC-3M cells (**e**) or ASH1L-overexpressing LNCaP cells in the presence of CoCl2 (200uM) (**f**), determined by qPCR. $n = 3$ biological replicates per group. **g** Schematics of OXPHOS pathway genes downregulated in TAMs upon ASH1L depletion in cancer cells (FDR <0.05; FC >1.5). Created in BioRender. Meng, C. (2025) https://BioRender.com/5we2ig4. **h** Human monocytes were isolated from blood and co-cultured with conditional medium from control and ASH1L-depleted PC-3M cells for 48 h, with or without recombinant protein of human IGF-2 (10 ng/ml). The expression of indicated genes was determined by qPCR. $n = 3$ biological replicates per group. Data in **e, f, h** represents the mean ± standard deviation. Statistical significance was determined by unpaired two-tailed $T$-test (**f**) or one-way ANOVA with Tukey's post hoc test (**e, h**). Source data are provided as a Source Data file.

expression via modulating trimethylations at H3K4. Besides, *ASH1L* deletion dramatically reduced the mRNA expression of IGF-2 in PC-3M cells (Fig. 6d, e). Reversely, overexpression of ASH1L-F3 led to the elevation of IGF-2 expression in LNCaP cells (Fig. 6f). By contrast, the *IGF-1* gene was expressed at an extremely low level in PC-3M cells; neither H3K4me3 nor H3K36me3 was marked on the *IGF-1* gene (Fig. 6d). In addition, ChIP-seq assay showed that HIF-1α directly bound to the *IGF-2* gene promoter in PC-3 cells (Supplementary Fig. 8a), and knocking down HIF-1α using siRNAs abolished the IGF-2 induction driven by ASH1L-F3 in LNCaP cells (Supplementary Fig. 8b). These

results indicated that IGF-2 is a direct target of ASH1L and HIF-1α in metastatic cells.

Our single-cell transcriptomic profiling analysis revealed that loss of ASH1L in metastatic cells caused a transcriptional reduction of core subunits in Complexes I, III, IV, and V of TAMs, which are key components that catalyze the OXPHOS and ATP synthesis (Fig. 6g). These effects were also verified in human monocytes cultured in conditioned media from ASH1L-depleted PC-3M cells (Fig. 6h and Supplementary Fig. 8c). In autoimmune diseases, IGF-2 was found to preprogram maturing macrophages to acquire anti-inflammatory phenotype by elevating OXPHOS[50]. When treating human monocytes with IGF-2 recombinant protein, we also observed AKT-GSK3b-mTOR signaling activation, OXPHOS gene upregulation, and increased expression of lipid-associated TAM markers (Supplementary Fig. 8d, e). Importantly, the supplement of IGF-2 recombinant protein can largely rescue OXPHOS gene expression and anti-inflammatory/pro-tumoral phenotype of TAMs in the absence of ASH1L (Fig. 6h and Supplementary Fig. 8c). In metastatic PCa patients, we also found that IGF-2 expression correlated with lipid-associated TAM markers and pro-tumoral phenotype markers (Supplementary Fig. 8f, g). These mechanistic studies demonstrated that, as a direct target of the ASH1L/HIF-1α complex in metastatic cells, IGF-2 mediates monocyte differentiation toward lipid-associated TAMs and maintains their pro-tumoral and anti-inflammatory phenotype by enhancing OXPHOS.

## Inhibition of the ASH1L-HIF-1α-TAM axis suppresses bone metastases of PCa

To determine if TAMs are essential to promote metastatic tumor outgrowth in bone driven by ASH1L, we performed macrophage depletion assays in vivo (Fig. 7a). C57BL/6J male mice received intratibial injection of control and ASH1L-knockdown DX1 cells expressing firefly luciferase, followed by macrophage depletion using monoclonal antibodies against CSF1R or IgG control (i.p., 300 μg per injection) and bioluminescence imaging weekly. Phenocopying ASH1L depletion, the blockade of CSF1R significantly suppressed the outgrowth of prostate tumors in the bone (Fig. 7b, c and Supplementary Fig. 9a). Of note, inhibiting ASH1L only impaired bone tumor outgrowth in the presence of macrophages, while depletion of macrophages abolished these effects (Fig. 7b, c and Supplementary Fig. 9a), suggesting TAMs are required for ASH1L's pro-metastatic role. Immunofluorescence staining not only verified the depletion of total TAMs (both F4/80+CD206+ and F4/80+CD206-) in the bone tumors after CSF1R blockade but confirmed the suppressing effects of ASH1L inhibition on pro-tumoral TAMs (F4/80+CD206+) in the bone niche (Fig. 7d and Supplementary Fig. 9b).

A recent study reported that AS-99 was a small-molecule inhibitor of ASH1L and showed anti-leukemic activities[52]. In metastatic PCa cells, we found that inhibiting ASH1L with AS-99 decreased histone methylations at H3K4 and H3K36 in a dose-dependent manner (Supplementary Fig. 9c) and impaired cell migration abilities in vitro (Supplementary Fig. 9d). Similar effects were also observed in breast cancer cells (Supplementary Fig. 9e). Next, we evaluated its efficacy in preclinical models of metastatic PCa. Fourteen days after PC-3M cells were injected into the left ventricle of male nude mice, vehicle control and AS-99 were administrated five times a week (i.p., 25 mg/kg) for three weeks (Fig. 7e). The results showed that AS-99 treatment significantly suppressed metastases to bone and prolonged the overall survival of mice (Fig. 7e, f). Compared to the vehicle control group, pro-tumoral TAMs (F4/80+CD206+) were reduced in metastatic tumors treated with AS-99 (Fig. 7g and Supplementary Fig. 9f). These preclinical studies demonstrated the therapeutic potential of targeting ASH1L in metastatic diseases.

Considering that HIF-1α is a crucial partner of ASH1L in metastatic cells and agents targeting HIF-1α are currently under clinical investigation[38,53], we also evaluated the effects of HIF-1α inhibitor PX-478 in our preclinical model of bone metastasis. We injected control and ASH1L-depleted DX1 cells into one tibia of C57BL/6J male mice and then treated the mice with vehicle or PX-478 thrice weekly (i.p.,40 mg/kg) for three weeks. IVIS imaging and ex vivo fluorescence imaging revealed that ASH1L depletion significantly suppressed tumor outgrowth in bone, but this effect was abolished by PX-478 treatment (Fig. 7h and Supplementary Fig. 9g). Besides, HIF-1α inhibition showed anti-tumor effects in the control group but not in the ASH1L depletion group (Fig. 7h and Supplementary Fig. 9g). Furthermore, we performed multiplex IHC staining in those bone tumors and found that pro-tumoral TAMs (F4/80+CD206+) were significantly reduced in ASH1L-expressing tumors upon HIF-1α inhibitor treatment (Fig. 7i and Supplementary Fig. 9h). These results demonstrate that HIF-1α plays a vital role in metastatic tumor outgrowth and TAM plasticity in the bone driven by ASH1L, suggesting the potential of HIF-1α targeted therapy in PCa patients with metastasis.

Last but not least, we assessed the association between ASH1L and TAMs in human metastatic PCa. To this end, we analyzed 24,433 non-epithelial cells in the scRNA-seq dataset of 13 men with metastatic PCa[41] and identified 18 clusters representing diverse subtypes of myeloid cells, lymphocytes, cancer-associated fibroblasts, and endothelial cells in metastatic sites (Fig. 7j and Supplementary Fig. 10a, b). We found that myeloid components in human metastatic tumors were mainly composed of monocytes and TAMs, and all three TAM clusters highly expressed lipid-associated TAM markers (APOE, C1QA, and TREM2) and exhibited a pro-tumoral phenotype featured by CD206 (MRC1) expression (Fig. 7j and Supplementary Fig. 10b, c). Based on ASH1L expression in epithelial cells, we classified 13 metastatic tumors into two groups, ASH1L-high (n = 5) and ASH1L-low (n = 8) (Supplementary Fig. 10d). Compared to the ASH1L-low group, lipid-associated TAMs are much more abundant in metastatic tumors expressing high levels of ASH1L (Fig. 7k, l), along with enriched HIF-1α transcriptome in cancer cells (Supplementary Fig. 10e). Furthermore, we performed immune cell proportion analyses of bulk RNA-seq data from 208 metastatic PCa patients[3] using the computational approach CIBERSORT. The results indicated that metastatic tumors containing ASH1L gene gain or amplification had a higher abundance of TAMs (Supplementary Fig. 10f). These human-relevance studies provide crucial evidence to support ASH1L's role in promoting TAMs, particularly lipid-associated TAMs, in the metastatic niche.

## Discussion

Bone metastases are common in cancer patients and are strongly associated with high mortality. Identifying genetic and epigenetic determinants promoting bone metastases and the crosstalk between invading cancer cells and metastatic bone niches facilitates the development of effective treatments. In this study, we identified an understudied histone methyltransferase, ASH1L, as a bona fide epigenetic driver in metastatic PCa. We found that ASH1L was genetically amplified and overexpressed in metastatic tumors and was associated with disease progression and worse prognosis. By characterizing the loss-of-function and gain-of-function effects, we report that ASH1L primes metastasis and reshapes the immune landscape in the bone niche (Fig. 8). Mechanistically, ASH1L co-opts with HIF-1α to induce a pro-metastatic transcriptome in cancer cells via catalyzing histone methylations at H3K4 and H3K36. Furthermore, we identified an unrecognized regulatory axis of ASH1L-IGF-2 in invading cancer cells, which induces lipid-associated and angiogenic TAMs and maintains their pro-tumoral and anti-inflammatory phenotypes by enhancing OXPHOS (Fig. 8). Importantly, our preclinical and human-relevance studies demonstrate the therapeutic potential of targeting ASH1L in metastatic bone tumors, providing insights into targeted therapies in patients with lethal malignancies.

Dysregulation of HMTs is common in leukemia and impinges on oncogenic and tumor suppressor functions[54,55]. Cancer genome and transcriptome studies revealed that HMTs are frequently altered in metastatic diseases[56], but much less is known about their biological

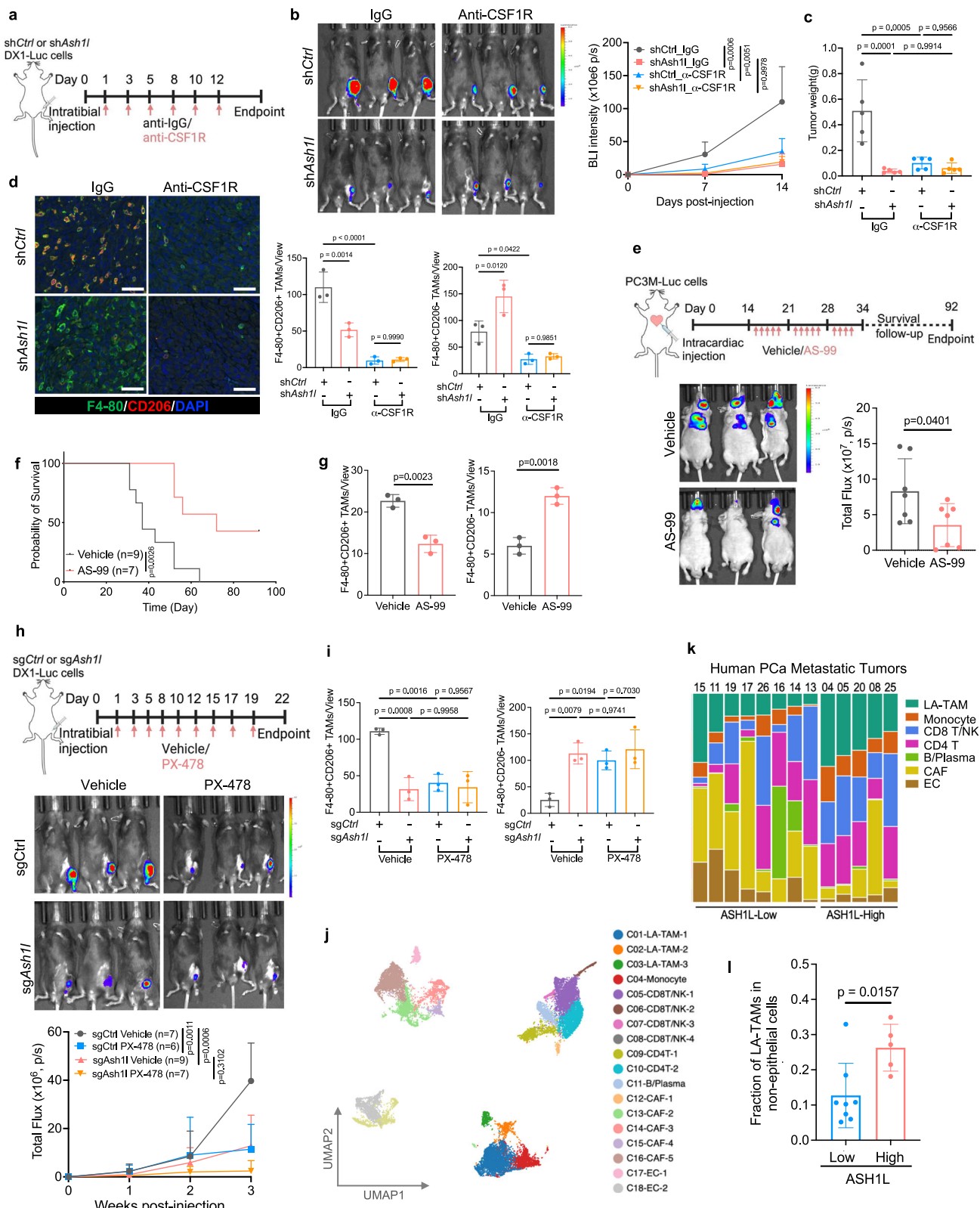

functions and mechanism of action during metastatic progression. Our studies establish ASH1L as an essential HMT that drives cancer invasiveness and bone metastasis. Distinct from its role in leukemia[21–23], ASH1L has little impact on the proliferation of metastatic cells but profoundly enhances their migration/invasion capacities, reprograms the myeloid cells in the metastatic niche, and promotes metastatic outgrowth in the bone. Prior studies reported that ASH1L

transcriptionally activates HOX family genes by modulating H3K36me2/3 and H3K4me3 in hematopoietic stem cells and leukemia[16–23]. Despite exhibiting similar methyltransferase activities in metastatic cells, ASH1L mainly regulates genes that are involved in EMT, extracellular matrix remodeling, and immunomodulation, partially explaining its unique biological function in this context. Besides, ASH1L was found to form complexes with other epigenetic factors or

**Fig. 7 | Inhibition of the ASH1L-HIF-1α-TAM axis suppresses bone metastases of PCa. a** Schematics of experimental design. $5 \times 10^5$ control and ASH1L-knockdown DX1 cells were injected into the tibia of C57BL/6 J male mice, followed by macrophage blockade using CSF1R or IgG antibodies (i.p., 300 µg/injection). Created in BioRender. Meng, C. (2025) https://BioRender.com/xsz0qzx. **b** Bioluminescence images and intensity quantification of bone tumors over time are shown. $n = 5$ mice/group. **c** Quantification of bone tumor weight at the endpoint. **d** Representative immunofluorescent images (enlarged, left) and quantifications of TAMs (right) in bone tumors. $n = 3$ mice/group. Scale bar = 50 µm. **e** $2 \times 10^6$ PC-3M cells were intracardiac injected into male nude mice, followed by treatment of vehicle or ASH1L inhibitor AS-99 (i.p., 25 mg/kg). Bioluminescent imaging and intensity quantification of metastatic tumors on Day 34 are presented ($n = 7$ mice/group). Created in BioRender. Meng, C. (2025) https://BioRender.com/2ngdt7t. **f** Overall survival of tumor-bearing mice. **g** Multiplex IHC staining of TAMs in bone tumors. $n = 3$ mice/group. **h** $5 \times 10^5$ control and ASH1L-depleted DX1 cells were injected into the tibia of C57BL/6J male mice, followed by treatment of vehicle or HIF-1α inhibitor PX-478 (i.p.,40 mg/kg). Created in BioRender. Meng, C. (2025) https://BioRender.com/bwpla4s. Bioluminescence images and intensity quantification of bone tumors over time are shown. $n = 7, 6, 9$, and 7 mice/group as indicated. **i** Multiplex IHC staining of pro-tumoral and pro-inflammatory TAMs in bone tumors. $n = 3$ mice/group. **j–l** Single-cell transcriptomic analysis of human PCa metastatic tumors (GSE210358; $n = 13$ patients). **j** The UMAP view of 24,433 non-epithelial cells, color-coded by 18 subclusters with cell-type assignment. LA-TAM lipid-associated TAM, CAF cancer-associated fibroblast, EC endothelial cells. **k** Composition distribution of major cell types among 13 samples grouped by ASH1L expression in epithelial cells. **l** Fraction of LA-TAMs in total non-epithelial cells from metastatic tumors with low versus high ASH1L expression (Low $n = 8$; High $n = 5$). Statistical significance was determined by unpaired two-tailed $T$-test (**e, g, l**), one-way ANOVA with Tukey's post hoc test (**b–d, h, i**), or log-rank (Mantel-Cox) test (**f**). Data in **c–e, g–i, l** represents the mean ± standard deviation. Source data are provided as a Source Data file.

transcriptional coactivators during leukemogenesis or DNA damage response, such as MRG15, MLL, and LEDGF[21–23,32,57,58]. Here, we reported an underappreciated partner of ASH1L, transcriptional factor HIF-1α, which directly interacts and co-opts with ASH1L in regulating pro-metastatic genes in invading cancer cells. Although the co-localization of ASH1L and HIF-1α found in both metastatic tumors and leukemia suggests a universal mechanism of gene regulation, their co-target genes in metastatic cells are distinct from those in leukemia. Future proteomic studies are needed to decipher ASH1L's epigenetic machinery and regulatory specificity in different contexts.

Nearly 80% of men with advanced PCa develop bone metastases, leading to bone pain, skeletal fracture, and a high mortality rate[28,59]. Metastases from prostate adenocarcinomas nearly always form osteoblastic lesions in bone, however, commonly used translational models of PCa bone metastasis only present osteolytic features[60,61]. Besides, these models are derived from human cancer cell lines in immunodeficient mice, and thus cannot meet the increasing needs of investigating the interaction between host immune cells and invading cancer cells. In this study, we generated a syngeneic model of PCa metastatic outgrowth in bone, which had mixed osteolytic/osteoblastic features, lacked T cell infiltration, and abundant immunosuppressive myeloid cells, recapitulating bone metastases in PCa patients[62]. Even though intratibial injections couldn't model the metastatic process of invasion, intravasation, or extravasation, our rapid syngeneic bone metastatic model provides a valuable tool for investigating metastatic colonization and outgrowth in the bone, understanding the immunosuppressive metastatic bone niche, and preclinical evaluation of immunomodulating therapies in metastatic diseases. Prior studies reported the invasive bone drilling procedure of intratibial injections caused local inflammation[63,64], which is known to enhance monocyte recruitment and alter the phenotypes and function of tissue-resident macrophages[65,66]. Thus, caution should be taken when investigating the bone niche and TAMs using intratibial injection models. Like what we did throughout this study, appropriate control groups and functional validation are required for immunophenotyping and data interpretation.

Although increasing evidence supports TAMs' essential role in metastasis[7,8,10,12,49,67,68], the spectrum of TAMs in the metastatic bone niche remains elusive. Combining single-cell transcriptomics with our unique bone metastasis model, we characterized five diverse subtypes and activation states of TAMs in metastatic bone tumors, expanding our knowledge beyond traditional M1/M2 TAM theory. Among them, lipid-associated and angiogenic subtypes comprise ~70% TAMs in the metastatic bone niche and highly express M2 markers, suggesting their dominance and pro-tumoral functions in the metastatic niche. Our findings in preclinical models and metastatic PCa patients reinforce the notion that lipid-associated macrophages contribute to cancer progression, metastasis, and therapy resistance. More importantly, we identified ASH1L as a key epigenetic regulator mediating the crosstalk between invading cancer cells and TAMs in the metastatic bone niche. By epigenetically reprogramming invading cancer cells, ASH1L not only induces tumor-infiltrating monocyte differentiation into lipid-associated and angiogenic TAMs, but also suppresses antigen presentation, interferon signaling, and immunostimulatory molecule expression in inflammatory TAMs. In myeloid cells, chromatin remodeling or histone modifications were reported to affect lineage differentiation, phenotype switch, and activation states during hematogenesis and tumor progression[69–71]. Here, we propose a conceptual framework, i.e., epigenetic alterations in invading cancer cells equally contribute to TAM plasticity and function in metastatic niches, which, in turn, facilitates the fitness, colonization, and outgrowth of invading cancer cells during distant metastasis.

During cancer progression, TAMs undergo profound metabolic reprogramming and present the selective utilization of glucose or fatty acids as an energy source[46,72,73]. In contrast to pro-inflammatory (M1-like) TAM utilizing glycolysis, pro-tumoral (M2-like) TAM have higher basal mitochondrial oxygen consumption rates and are heavily skewed towards using FAO and OXPHOS to support rapid cell proliferation and pro-tumoral functions[73,74]. Recent studies also revealed that lipid-associated TAMs have a distinctive enrichment of lipid metabolism and OXPHOS pathways[48,49]. In this study, we reported that the ASH1L/HIF-1α complex in invading cancer cells promotes IGF-2 expression, which triggers AKT-GSK3b-mTOR signaling and activates OXPHOS pathways in monocytes in the bone niche. This leads to monocyte differentiation into lipid-associated and angiogenic TAMs and maintains their pro-tumoral function during metastatic tumor outgrowth. In autoimmune diseases, IGF-2 was also found to prime macrophages to acquire an OXPHOS-dependent anti-inflammatory phenotype[50], highlighting that IGF-2's regulation on TAMs is a broadly applicable mechanism. IGF-1R, the receptor of IGF-2, is recognized as a promising target of cancer therapy, and monoclonal antibodies targeting human IGF-1R have been studied in clinical trials[75,76]. Future studies are warranted to evaluate IGF-1R blockade's therapeutic potential and its effects on TAM plasticity in preclinical models of bone metastasis.

Given ASH1L's crucial role in leukemogenesis, small-molecule inhibitors have been developed and tested in preclinical leukemia models[52]. Our studies support ASH1L as a promising therapeutic target in metastatic malignancies, particularly in patients with bone metastases. In cellular experiments, we showed that AS-99, a small-molecule inhibitor against ASH1L, suppressed cancer cell migration and invasion. In preclinical mouse models, AS-99 also exhibited considerable effects in suppressing bone metastases and reversing the pro-tumoral TAM-enriched metastatic bone niche, providing proof-of-concept evidence for targeting ASH1L in metastatic PCa and other malignancies. In summary, we identified ASH1L as a bona fide epigenetic oncoprotein and a promising therapeutic target in metastatic cancers, uncovered its cancer cell-intrinsic and extrinsic roles in metastatic progression and bone

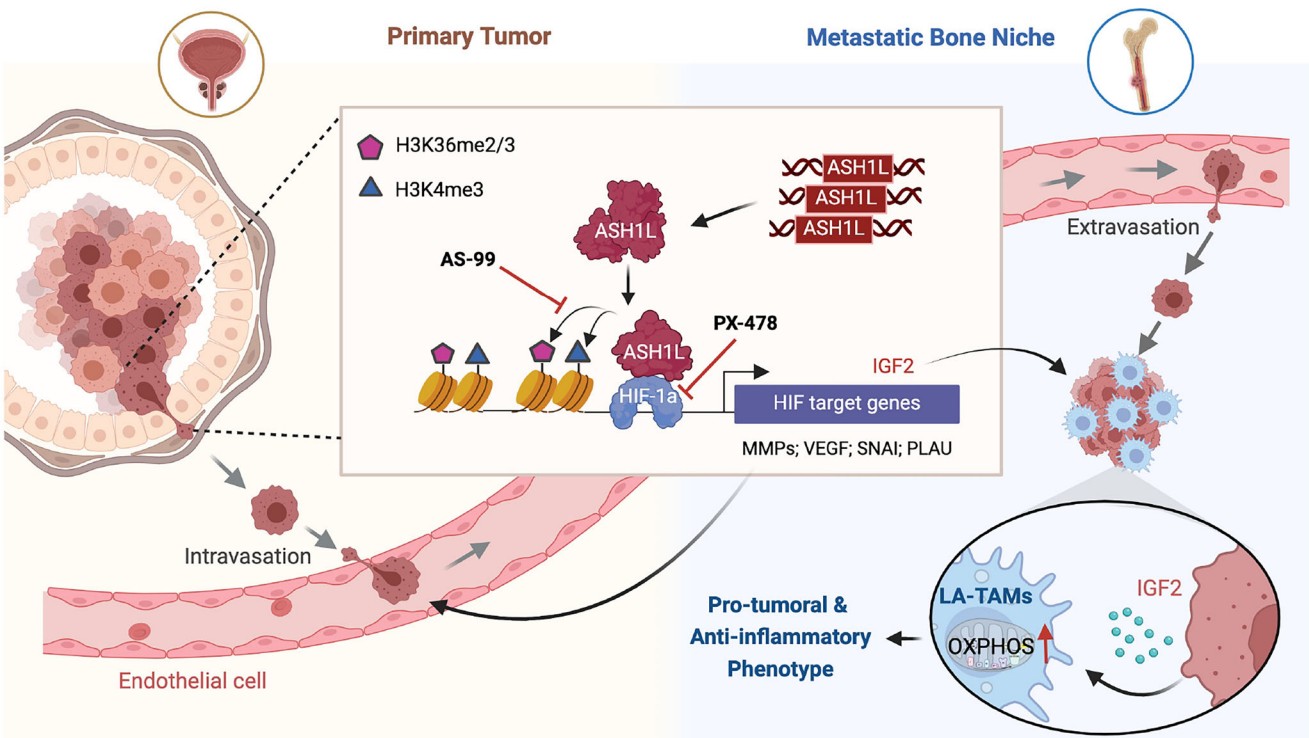

**Fig. 8 | Schematic of working model.** Histone methyltransferase ASH1L is genetically amplified and overexpressed in metastatic cells and promotes H3K4me3 and H3K36me2/3 on HIF-1α target genes. ASH1L cooperates with HIF-1α to induce pro-metastatic genes, including MMPs, VEGF, SNAI, and PLAU, facilitating cancer cell invasiveness. Besides, the regulatory axis of ASH1L-IGF-2 in metastatic cells induces lipid-associated TAMs (LA-TAMs) and maintains their pro-tumoral and anti-inflammatory phenotypes by enhancing oxidative phosphorylation (OXPHOS). As an epigenetic determinant in metastatic cancers, ASH1L plays vital cell-intrinsic and -extrinsic roles in priming metastasis and reshaping the immune landscape in the bone niche. Pharmacologic inhibition of ASH1L using small molecular compound AS-99 elicits anti-metastasis responses. Schematic of the working model created in BioRender. Meng, C. (2025) https://BioRender.com/el9a9q1.

niche remodeling, and shed light on ASH1L-targeted therapy in PCa and other malignancies with high metastasis rates.

## Methods

All mouse experimental procedures followed the Institutional Animal Care and Use Committee (IACUC) protocol (#00001955). The mice were housed under a 12-h light/dark cycle, at 68–78 °F with 30–70% humidity. MD Anderson IACUC's guidelines for the proper and humane use of animals in biomedical research were followed. Due to the nature of the PCa study, only male mice were included throughout the study. Euthanasia was carried out at the endpoint by $CO_2$ inhalation followed by cervical dislocation, which is an approved method of euthanasia for mice.

### Cell culture and reagents

293T (CRL-3216), DU145 (HTB-81), PC-3 (CRL-1435), LNCaP (CRL-1740), and THP-1 (TIB-202) cell lines were obtained from the American Type Culture Collection (ATCC). B16F1, B16F10, MDA-MB-231, MDA-231-LM2, and MDA-231-BoM-1833 melanoma and breast cancer cell lines were gifts from Dr. Li Ma's laboratory (MD Anderson Cancer Center, Houston, TX). PC-3M cell lines were gifts from Dr. Ronald DePinho's laboratory (MD Anderson Cancer Center, Houston, TX). All cell lines were cultured in media supplemented with 10% FBS (Gibco, 10082147) and 1% penicillin/streptomycin (Gibco, 15140163) and incubated at 37 °C under 5% $CO_2$. 293T, PC-3M, B16F1, B16F10, MDA-MB-231, MDA-231-LM2, and MDA-231-BoM-1833 cells were cultured in Dulbecco's modified Eagle's medium (DMEM) (Corning, 10-017-CV). DU145 cells were cultured in the EMEM medium (ATCC, 30-2003). PC-3 cells were cultured in F-12K medium (ATCC, 30-2004). LNCaP and THP-1 cells were cultured in RPMI 1640 medium (Corning, 10-040-CV). All of the cell lines were confirmed by The University of Texas MD Anderson Cancer Center's Cytogenetic and Cell Authentication Core and were regularly tested for mycoplasma using MycoAlert PLUS detection kit (Lonza, LT07-710), according to the manufacturer's instructions. The key reagents used in the cell culture and treatment included ASH1L inhibitor AS-99 (MedChemExpress, HY-141429A), human IGF-2 protein (Sino Biological, 13032-HNAY-100), Cobalt chloride (Sigma-Aldrich, 15862), HIF-2α inhibitors PT1299 (Selleck, S8351), HIF-1α inhibitors LW6 (Selleck, S8441), 2-MeOE2 (Selleck, S1233), and PX-478 (MedChemExpress, HY-10231).

### Transient transfection and lentiviral transduction

Negative control siRNA and siRNAs targeting *ASH1L* were purchased from Sigma-Aldrich. All transient transfections of siRNA into cells were performed according to the manufacturer's instructions. In brief, siRNA and Lipofectamine 2000 reagent (Thermo Fisher, #11668019) were mixed and incubated in serum-free Opti-MEM (Gibco, 31985-062) at ambient temperature and then added to cells with 60% confluence. Cells were collected for further analyses 48–72 h after transfection. Lentivirus was used to establish stable cell lines for CRISPR knockout or gene overexpression. Empty lentiviral vectors were used as the control. For lentiviral transduction, the lentiviral constructs, psPAX2, and pMD2.G were transfected into 293 T cells using Lipofectamine 2000 reagent (Thermo Fisher, #11668019). After 48 h, the viral supernatants were harvested and filtered. Transduced cells were incubated with viral supernatants in the presence of 10 mg/mL polybrene (EMD Millipore, TR-1003-G), followed by selection using 10 μg/mL puromycin (Gibco, A1113803) or sorting with a FACS Aria II Cell Sorter. The human ASH1L sgRNA-CRISPR/Cas9 All-in-One Lentivector set (cat# 125351110595) was purchased from ABM. The human ASH1L shRNAs were purchased from Sigma-Aldrich. To knock out Ash1l in murine cancer cells, sgRNAs targeting the mouse Ash1l gene were

designed using CHOPCHOP and cloned into pLentiCRISPRv2 (Addgene, 52961). The siRNA, lentiviral construct, and plasmid information are shown in Supplementary Data 9.

### Site-directed mutagenesis

The HA-tagged ASH1L-F3 plasmid was used as a template to construct HA-tagged F3-F2260A mutation plasmids using the Q5 Site-Directed Mutagenesis Kit (New England BioLabs, E0554S), following the manufacturer's instructions. Plasmids were amplified using the QIAprep Spin Miniprep Kit (Qiagen, 27106) and verified by sequencing at MD Anderson's Sequencing and Microarray Facility. Mutagenesis primers (Supplementary Data 9) were designed using the NEBaseChanger website (http://nebasechanger.neb.com/).

### Maintenance of GEM Models

The *Pb-Cre; Pten^{L/L}; TrpS3^{L/L}; Smad4^{L/L}; mTmG (PbPPS)* mouse strain was a gift from Dr. Ronald DePinho's research group at MD Anderson Cancer Center (MDACC, Houston, TX)[26,27,77]. Mice were interbred, regularly genotyped, and maintained at the MDACC animal facility. Mice were monitored for signs of ill health every day and sacrificed and necropsied when moribund. The number of micro-metastases in the lymph nodes in *PbPPS* mice was measured by ex vivo fluorescence imaging using a Leica M165 FC microscope. All animal-related methods here and below were reviewed and approved by the MDACC Institutional Animal Care and Use Committee (protocol #00001955). Due to the nature of the PCa study, only male GEM mice were used in this study.

### Tumor Growth

$2 \times 10^6$ control and ASH1L-depleted PC-3M cells in 100 μL PBS supplemented with 50% growth factor-reduced Matrigel (Corning, 354230) were injected subcutaneously into the flanks of male nude mice (6 weeks old, Taconic). Tumor growth was monitored by calipers (volume = length × width × width/2) twice a week. 1200 mm³ was the pre-determined endpoint. The maximal IACUC-approved tumor volume was not exceeded.

### Xenograft models of bone metastasis

For ultrasound-guided intracardiac injection, control and ASH1L-depleted PC-3M cells expressing firefly luciferase and RFP (Luc/RFP) were generated. Cells ($3 \times 10^6$) in 100 μL of PBS were intracardially injected into the left cardiac ventricle of anesthetized male nude mice (6-week-old), with the guidance of Vevo 2100/LAZR ultrasound. Tumor metastasis was monitored weekly by IVIS 200 (In Vivo Imaging System, PerkinElmer). D-luciferin (3 mg/injection; PerkinElmer, 122799) was intraperitoneally injected into the mice for IVIS imaging. To examine the effect of ASH1L inhibitor AS-99 on tumor metastasis, $2 \times 10^6$ PC-3M-Luc cells were intracardially injected into the left cardiac ventricle of male nude mice. After 2 weeks of cell injection, the mice were randomly assigned to 2 groups for administration of AS-99 (i.p. 25 mg/kg) or vehicle control five times per week (Monday-Friday, three consecutive weeks). Tumor metastasis was monitored weekly by IVIS 200 and the mice survival was documented. For intra-tibia injection, $10^6$ LNCaP-Luc/RFP cells expressing ASH1L-F3 or vector control were injected into the left tibia of 6-week-old male nude mice using insulin syringes (30Gx1/2″, 3/10 cc; BD, 328431). Tumor metastasis was monitored weekly by IVIS Lumina XR (PerkinElmer). Bioluminescence signals were analyzed using Living Image software (Caliper Life Sciences). All nude mice used in this study are the strain of CrTac:NCr-Foxn1nu (Sp/Sp) purchased from Taconic Biosciences (Cat # NCRNU-M). Due to the nature of the PCa study, only male mice were used in this study.

### Generation of a syngeneic model of PCa metastatic outgrowth in bone

The primary murine PCa cell line DX1 was generated from male mice of *PbPPS* GEMM in our previous study[44] and cultured in DMEM (Corning, 10-013-CV) with 10% FBS and 1% penicillin/streptomycin. $5 \times 10^5$ DX1 cells expressing firefly luciferase (DX1-Luc) were injected into the tibia of 6-week-old male C57BL/6 mice (Taconic) using insulin syringes (30Gx1/2″, 3/10 cc; BD, 328431). The outgrowth of metastatic tumors and bone formation was monitored by IVIS 200 weekly. The bone tumors were imaged by ex vivo fluorescence imaging using a Leica M165 FC microscope. To determine the impact of ASH1L on PCa metastatic outgrowth in bone immunocompetent mice, $5 \times 10^5$ DX1-Luc cells, with or without Ash1l knockout/knockdown, were injected into the tibia of 6-week-old male C57BL/6 mice (Taconic), followed by weekly monitoring using IVIS 200 and MX-20 Cabinet X-ray System (Faxitron). For HIF1A inhibitor treatment, $5 \times 10^5$ control and ASH1L-knockout DX1 cells were injected into the tibia of one leg of C57BL/6J male mice (Taconic), followed by treatment of vehicle or PX-478 from Day 1 post-implantation, three times per week (i.p.,40 mg/kg) for 3 weeks. The outgrowth of metastatic tumors and bone formation was monitored by IVIS spectrum weekly. Due to the nature of the PCa study, only male mice were used in this study. Euthanasia criteria approved by the IACUC were not exceeded, including (1) a maximum cumulative tumor burden of 1.5 cm in diameter; (2) tumor-induced impairment of eating, urination, defecation, or ambulation; and (3) very poor body condition.

### Histomorphometric analysis of the bone samples

The tumor-bearing bones were fixed in 10% neutral-buffered formalin, decalcified in 12.5% EDTA for 2 weeks, and then embedded in paraffin. Haematoxylin and eosin (H&E), tartrate-resistant acid phosphatase (TRAP), or Toluidine blue staining were performed on serial sections. To assess cortical bone volume fraction, we used Qupath software[78] version 0.5.1-x64 on H&E-stained images to define an ROI encompassing the entire tibia, excluding regions of new woven bone formation. The measured ROI area was considered the tissue volume. We then trained the software to specifically classify the cortical bone area as cortical bone volume. For new woven bone volume fraction analysis, we used Qupath software on H&E-stained images to define a tissue area, including the new woven bone and tumor region, and trained the software to specifically classify the new woven bone area as woven bone volume. For clarity and consistency in our analytic approach, we report the results as bone volume/tissue volume (BV/TV), and we batch processed all images using a saved script to ensure uniformity. For TRAP staining analysis, the osteoclast number per area (mm²) in the new woven bone area was assessed as multi-nucleated TRAP+ as well as the adjacent bone surface cells. Toluidine blue staining was administered to quantify the number of cuboid osteoblasts per area (mm²) in the new woven bone area.

### Macrophage depletion assay

$5 \times 10^5$ control or Ash1l- knockdown DX1-Luc cells were intratibial injected into male C57BL/6 mice. On the next day, mice were randomly assigned to two groups for administration of anti-CSF1R antibodies (i.p., 300 μg per injection; Bio X Cell BE0213) or Isotype IgG control (Bio X Cell BE0089) 3 times per week for 2 weeks. The outgrowth of metastatic tumors was monitored by IVIS 200 weekly.

### Human primary monocytes and THP-1 culture in conditioned media

PC-3M cells with or without ASH1L knockout were cultured in DMEM with 10% FBS and incubated at 37 °C under 5% $CO_2$. When the cell confluency reached 60–70%, the culture medium was replaced with 1% FBS for 48 h. Cell-free supernatants were harvested and used directly for the experiment.

Peripheral blood was collected from healthy donors (MD Anderson Blood Donor Center) in EDTA-coated blood tubes and diluted 1:1 using serum-free PBS. About 30 mL of the diluted blood was then stratified on top of 15 mL of Ficoll, and samples were centrifuged at

400×*g* for 30 min at room temperature (RT). The peripheral mononuclear cell (PBMC) fraction was collected and washed with PBS. CD14+ monocytes were isolated from PBMCs using the Human CD14+ Monocytes Isolation Kit (Biolegend, 480048) per the manufacturer's instructions. Primary monocytes were then induced by 50 ng/ml macrophage colony-stimulating factor (M-CSF, Sino Biological, 11792-HNAH-20) for 5 days and then co-cultured with conditioned medium derived from control or ASH1L-depleted PC-3M cells. The cells were collected after 48 h for quantitative real-time (qPCR) analysis. For the IGF-2 rescue assay, primary monocytes were cultured in M-CSF supplemented conditioned medium with or without 10 ng/ml human IGF-2 recombinant protein. The MD Anderson Blood Donor Center has a Blood Donor Consent signed by the blood donors, which allows MDACC permission to test and use donated blood as deemed appropriate, including use for research protocols for MDACC investigators (IRB protocol: 2020-0529).

The human monocyte cell line THP-1 supplemented with 150 nM phorbol-12-myristate-13-acetate (PMA, Sigma-Aldrich, P8139-1MG) was co-cultured with control or ASH1L-depleted PC-3M cells using transwell system for 48 h. For the IGF-2 rescue assay, THP-1 cells were cultured in the PMA-supplemented conditioned medium with or without 10 ng/ml human IGF-2 recombinant protein. The cells were collected after 48 h for qPCR analysis. To examine the effect of IGF-2 in regulating OXPHOS and macrophage phenotype, THP-1 cells were stimulated with PMA in RPMI 1640 medium without FBS for 12 h, and then the adherent THP-1 cells were cultured in the FBS-supplemented RPMI 1640 medium with different doses of IGF-2 for 24 h. The next day, maturing THP-1 cells were treated with IGF-2 for another 24 h in fresh FBS-supplemented RPMI 1640 medium. These cells were collected for qPCR and western blotting analysis.

### Immunoprecipitation (IP) and in vitro pulldown assay

Cells were lysed in the NP-40 buffer containing 20 mM Tris-HCl (pH 7.5), 0.4 M NaCl, 1 mM EDTA, 0.3% Nonidet P-40, and protease inhibitor cocktails (Roche, 11836153001) for 1 h. Cell lysates (800 μL) were incubated with anti-ASH1L antibody (2 ug, Bethyl, A301-749A) or control rabbit IgG (2 ug, Millipore Sigma, 12-370) overnight at 4 °C. Protein G Dynabeads (Thermo Fisher Scientific, 10009D) were added to each sample. After 4 h, the beads were eluted three times with NP-40 buffer and resuspended in 1x SDS sample buffer (Bio-Rad Laboratories, 1610747), followed by Western blotting analysis. For pulldown assay, Flag-tagged HIF1A proteins were overexpressed in LNCaP cells treated with MG132 and then purified using Flag beads IP. HIF1A proteins were then eluted using Flag peptides and then incubated with HA-tagged ASH1L-F3 proteins and HA beads at 4 °C overnight. After being washed with the lysis buffer (with 0.1% Nonidet P-40), HA beads-bound proteins were collected for Western blot.

### Western blotting analysis

Cells were lysed in 1x SDS sample buffer (Bio-Rad Laboratories, 1610747). Proteins were separated with 4–15% Mini-PROTEAN TGX Precast Protein Gels (Bio-Rad, 4561086) and transferred to a 0.2-μm nitrocellulose membrane (Bio-Rad, 1620168). After being blocked in 5% nonfat milk or BSA, membranes were incubated with the following primary antibodies overnight: anti-ASH1L (1:1000, Bethyl, A301-749A), anti-H3K4me3 (1:2000, Cell Signaling Technology, 9751S), anti-H3K36me3 (1:2000, Cell Signaling Technology, 4909S), anti-H3K36me2 (1:2000, Abcam, ab9049), anti-histone H3 (1:5000, Cell Signaling Technology,4499S), anti-HA (1:1000, Cell Signaling Technology, 3724S), anti-HIF1A (1:1000, Cell Signaling Technology, 36169), anti-p-AKT (1:1000, Cell Signaling Technology, 4060S), anti-AKT (1:1000, Cell Signaling Technology, 4691S), anti-p-ERK (1:1000, Cell Signaling Technology, 9101S), anti-ERK (1:1000, Cell Signaling Technology, 9102S), and anti-β-actin (1:5000, Sigma, A5441). Following three washes in TBST, the blots were incubated with secondary antibodies anti-rabbit IgG (HRP) (1:5000, Cell Signaling Technology, 7074 V) or anti-mouse IgG (HRP) (1:5000, Cell Signaling Technology, 7076 V). Proteins were developed by Western ECL substrates (Bio-Rad, 1705060 or Thermo Scientific, 34095) and imaged by the ChemiDoc Imaging System (Bio-Rad).

### Cell proliferation assay

$10^4$ control and ASH1L-depleted PC-3M cells or $5 \times 10^3$ control and ASH1L-F3 LNCaP cells were seeded in 24-well plates (Corning, 3526). Cells were digested with 0.25% Trypsin and resuspended in 1XPBS. A flow cytometer (Attune NxT, Thermo Fisher) was used to count cell numbers in 50 ul cell suspension at days 0, 2, 4, and 6. All events were included for cell-counting purposes; thus, no gating or sorting strategy was used for this assay.

### Migration and invasion assays

PC-3M ($5 \times 10^5$), MDA-231-BoM-1833 ($8 \times 10^4$), DU145 ($2 \times 10^4$), and LNCaP ($2 \times 10^5$) cells in 200 μL of serum-free medium were seeded into the upper chamber of 8-μm pore size transwell inserts (Falcon, 353097), and 750 μL of 10% FBS-containing medium was added into the lower chamber. After incubation at 37 °C for 24 h (PC-3M and MDA-231-BoM-1833) or 48 h (DU145 and LNCaP), the migrated cells were stained with crystal violet and visualized by Cytation 5. The migrated cells were analyzed and quantified using Image J software. Similar procedures were applied for invasion assays, in which the inside membrane of transwell inserts was coated with 100 μL of 2% Matrigel (Corning, 354230). For migration and invasion assays with drug treatment, PC-3M or MDA-231-BoM-1833 cells were treated with AS-99 or HIF-1α inhibitor 2-MeOE2 in serum-free media, and then seeded in the upper chamber for 24 h incubation. LNCaP cells were treated with HIF-1α inhibitors LW6, 2-MeOE2, or PX-478 in the presence of CoCl₂ or HIF-2α inhibitor PT2399 for 48 h. 3D spheroid invasion assays were performed using the hanging drop method. In brief, $10^3$ PC-3M cells were resuspended in DMEM medium and seeded with 20 μL of droplets for 48 h in a Petri dish to form tumor spheroids; the cell spheroids were then plated in Ultra-Low–attachment 24-well plates (Corning, 3473), coated with Matrigel, and cultured in DMEM medium supplied with 10% FBS. After incubation at 37 °C for 72 h, 3D spheroids were imaged by the microscope, and the length and area of the invading cells were analyzed and quantified using Image J software.

### Immunohistochemistry (IHC) and immunofluorescence (IF)

Mouse tissues were paraffin-embedded, and H&E and IHC staining were performed as previously described[79]. For IHC, rehydrated sections underwent antigen retrieval by incubation in antigen unmasking solution (Vector Laboratories, H-3300-250) at 95 °C for 30 min, followed by 110 °C for 10 s. Sections were treated with 0.3% hydrogen peroxide solution and blocked with Rodent Block M buffer (Biocare Medical, RBM961L) for 30 min. Sections were then incubated with primary antibodies at 4 °C overnight. The secondary antibody was Rabbit-on-Rodent HRP-Polymer (Biocare Medical, RMR622L). The signals were detected using the DAB Quanto chromogen and substrate kit (Fisher Scientific, TA-125-QHDX) and scanned by an Aperio CS2 scanner (Leica Biosystems). The clinically annotated tissue microarray of human PCa was purchased from BioCoreUSA (B-110Pro-1). Eighty human PCa samples were used for data analysis. The human tissue sections were reviewed and scored in a blinded manner for staining intensity (0–2). High expression of nuclear ASH1L corresponded to a score of 2, whereas low or negative expression corresponded to staining scores of 1 and 0, respectively. GraphPad Prism version 9.2.0 software was used for statistical analyses.

For IF staining, slides were incubated with primary antibodies at 4 °C overnight. The antibodies were diluted in 1XPBS (Corning, 21-030-CV), containing 1% bovine serum albumin (BSA) Fisher Scientific, BP9703100) and 0.3% Triton X-100 (Sigma-Aldrich, X100-500ML). The

sections were incubated with secondary antibodies (1:200) for 1 h, and then mounted with ProLong Gold Antifade Mountant with DNA Stain DAPI buffer (Invitrogen, P36931). The slides were scanned by Vectra Polaris (Akoya Biosciences). The primary antibodies for IHC and IF included anti-ASH1L (1:50, Bethyl, A301-748A), anti-HA (1:800, Cell Signaling Technology, 3724S), anti-MMR/CD206 (1:100, R&D Systems, AF2535), anti-F4/80 (1:100, Cell Signaling Technology, 70076), and anti-HIF1A (1:200, Abcam, ab51608).

## Multiplex IHC

Multiplex IHC staining was carried out using Opal Polaris 7-Color Manual IHC Detection Kit (Akoya Biosciences, NEL861001KT) according to the manufacturer's instructions. Paraffin-embedded mouse tissues were dewaxed, rehydrated, fixed in 10% neutral-buffered formalin for 20 min, and subjected to antigen retrieval using the microwave. After cooling, slides were blocked in blocking buffer for 10 min at RT and incubated with primary antibodies at 4 °C overnight. Primary antibodies information is listed here: anti-MMR/CD206 (1:100, R&D Systems, AF2535), anti-F4/80 (1:100, Cell Signaling Technology, 70076), anti-Ly-6C (1:100, Biolegend, 128001), and anti-Ly-6C (1:100, Abcam, ab314120). Slides were incubated in Opal polymer HRP Ms+Rb for 10 min at RT. Primary antibodies derived from goat or rat, Goat-on-Rodent HRP-Polymer (Biocare Medical, GHP516H), or Rat-on-Mouse HRP-Polymer (Biocare Medical, RT517H) kits were used to generate HRP signals. Opal Working Solutions were then applied to the tissue sections, followed by a 10-min incubation at RT to generate Opal signals. After completing all staining cycles, slides were counterstained with the DAPI working solution (provided in the kit) and scanned using the Vectra Polaris imaging system (Akoya Biosciences). Multiplex IHC was performed in three tumors per group throughout the study. Three to five views per tumor were captured in a blinded manner using Phenochart 1.1.0. The average cell numbers of these individual views were used to represent each tumor sample. At least three biological replicates (three tumor samples) were used for quantification. Statistical analyses were performed using GraphPad Prism version 9.2.0.

## qPCR analysis

Total RNAs were isolated from the indicated cells using the RNeasy Mini Kit (Qiagen, 74104). cDNA was synthesized using the High-Capacity cDNA Reverse Transcription Kit (Applied Biosystems, 4368813). Quantitative PCR was performed using SYBR Green Master Quantitative PCR Mix (Applied Biosystems, A25779) and QuantStudio 3 Real-Time PCR Systems (Thermo Fisher Scientific). The primers for real-time PCR are listed in Supplementary Data 9. Data were normalized against ACTB. All experiments were performed in triplicate. Data analysis was performed using GraphPad Prism version 9.2.0.

## Bulk RNA sequencing and data analysis

Control and ASH1L-knockout PC-3M cells were lysed with TRIzol (Invitrogen, 15596026), and total RNAs were extracted using the RNeasy Mini Kit (Qiagen, 74104). Library preparation and sequencing were performed by the ATGC next-generation sequencing core facility at MD Anderson, using a $2 \times 75$ bases paired-end protocol on a Next-Seq500 instrument. Nine libraries (three biological replicates per condition) were sequenced, generating 36–53 million pairs of reads per sample. Each pair of reads represents a cDNA fragment from the library. The reads were aligned to the human genome (hg38) using TopHat (version 2.0.10)[80] and the overall mapping rate for the reads was 84–89%. For differential expression analysis, the number of fragments for each gene from GENCODE[81] was quantified using htseq-count from the HTSeq package (version 0.6.0)[82]. Genes with fewer than ten fragments in all samples were excluded before the differential expression analysis.

The statistical assessment of differential expression between conditions was performed using the R/Bioconductor package DESeq

(version 1.18.0)[83]. Differentially expressed protein-coding genes (FDR ≤0.05; FC ≥2) were identified in both sgRNAs targeting ASH1L (Supplementary Data 1), followed by centralization of each gene across all samples, hierarchical clustering using Ward.D2 method, and heatmap generation using R or GenePattern. DESeq expression values for all genes were utilized for gene enrichment analysis using GSEA 4.1.0. In addition to hallmark gene sets, a gene set consisting of 110 HIF-1α target genes (Supplementary Data 10) was also used for GSEA analysis. To generate the signal landscape for bulk RNA-Seq, the first reads from the fragments with both ends mapped were retained. Additionally, reads from fragments with only one end mapped were included. Each read was extended to its 3' end by 200 bp within exon regions. For each read, a weight of 1/n was assigned, where n represents the number of positions the read was mapped to. The sum of weights for all reads covering each genomic position was rescaled to normalize the total number of fragments mapped to exons to 1 million, and then averaged over a 10 bp resolution. The resulting averaged values were displayed using the Integrative Genomics Viewer (IGV)[84].

## CUT&RUN and data analysis

CUT&RUN was carried out using the CUT&RUN Assay Kit (Cell Signaling Technology, 86652). In brief, $10^5$ cells were harvested and bound to Concanavalin A magnetic beads and then incubated with IgG control (5ul per reaction, Cell Signaling Technology, 66362), anti-H3K4me3 (2ul per reaction, Cell Signaling Technology, 9751), and anti-H3K36me3 (2ul per reaction, Abcam, ab9050) antibodies overnight at 4 °C. The cell-bead suspension was washed with digitonin buffer and incubated with pAG-MNase enzyme for 1 h at 4 °C, followed by incubation with digitonin buffer containing $CaCl_2$ to activate pAG-MNase digestion for 30 min at 4 °C. The reaction was stopped by the addition of 1×STOP buffer with digitonin, RNase A, and heterologous spike-in DNA. After 10 min incubation at 37 °C, DNA fragments were released from cells and extracted using the QIAquick PCR purification kit (QIAGEN, 28106). Sequencing libraries were prepared using the DNA Library Prep Kit (Cell Signaling Technology, 56795) for Illumina Platforms. The multiple bead-based clean-ups were used to remove oligonucleotides and small fragments. The libraries were then sequenced using an Illumina NovaSeq 6000 (pair-end 150, Illumina) by Novogene (Sacramento, CA, USA). 12 CUT&RUN libraries (two biological replicates per histone mark and condition), as well as the corresponding IgG and input libraries, were sequenced. This generated 13–44 million pairs of reads/DNA fragments per sample.

**Mapping.** After adapter trimming using Trim Galore! (Version 0.6.5), the reads were mapped to the spike-in genome (sacCer3) and the human genome (hg38) using Bowtie (version 1.1.2)[85] with the following parameters: "--allow-contain --maxins 2000 -v 2 -m 1 --best --strata". To avoid PCR bias, only one copy of multiple fragments mapped to the same genomic position was retained for further analysis.

**Peak calling.** For each H3K4me3 sample, two sets of peaks were identified using MACS (version 1.4.2, window size 500 bp, p value cutoff 1e⁻⁵)[86] by comparing against the corresponding IgG sample or input sample, respectively. The H3K4me3 peaks that overlapped in both sets but were not included in ENCODE blacklisted regions[87] were used for further analysis.

**Differential analysis.** The peaks from all H3K4me3 samples were merged, and the number of fragments within these merged peaks was counted for each H3K4me3 sample. The resulting count table was used to identify differential H3K4me3 peaks between ASH1L-knockout and control conditions using the R/Bioconductor package edgeR[88]. The numbers of fragments uniquely mapped to the spike-in genome were used as the library size for the four H3K4me3 samples in edgeR. Protein-coding genes (from GENCODE Release 37) that exhibited

differential H3K4me3 peaks within their promoter regions (defined as −1000 bp to +500 bp of the transcription start site) were considered to be associated with differential H3K4me3, using a threshold of FDR ≤0.05 and FC ≥1.5 (Supplementary Data 2). For H3K36me3, the genomic locations of protein-coding gene bodies (from GENCODE Release 37) were identified, and the number of fragments within these gene bodies was counted for each H3K36me3 sample. The resulting count table was used to identify genes with differential H3K36me3 between ASH1L-knockout and control conditions using edgeR. As the percentage of uniquely mapped fragments to the spike-in genome is extremely low (-0.1%) for H3K36me3 samples, spike-in fragments were not utilized for the normalization in differential analysis. Instead, a list of common significant gene bodies in all H3K36me3 samples was determined by comparing against IgG and input samples using a binomial test with FDR ≤0.05. Then the numbers of fragments uniquely mapped to these common significant gene bodies were used as the library sizes in edgeR. Protein-coding genes that exhibited differential H3K36me3 peaks within their gene body were considered to be associated with differential H3K36me3, using a threshold of FDR ≤0.05 and FC ≥1.5 (Supplementary Data 3).

**Signal landscape.** To generate the signal landscape for CUT&RUN, each fragment was resized to a length of 151 bp, spanning from −75 bp to +75 bp around its midpoint. The count of fragments covering each genomic position was multiplied by $1 \times 10^6$ divided by the library size used in edgeR. These values were then averaged over a 10 bp resolution and displayed using the Integrative Genomics Viewer (IGV).

**Classification of genes.** Genes were grouped into five clusters (C1–C5) based on changes in H3K4me3 and H3K36me3 signals upon ASH1L depletion: C1 (both marks decreased), C2 (only H3K4me3 decreased), C3 (only H3K36me3 decreased), C4 (H3K4me3 and/or H3K36me3 increased), and C5 (no changes in either H3K4me3 or H3K36me3).

**Heatmap and metaplot.** The longest transcript was selected to represent each gene. For each CUT&RUN sample, the coverage of fragments was multiplied by $1 \times 10^6$, divided by the library size used in edgeR, and then averaged across replicate samples. For the metaplot, the resulting values for each histone mark and condition were averaged over genes within each gene group (C1–C5) and plotted accordingly. For the heatmap, the resulting values for H3K4me3 and H3K36me3, along with the log2 ratio values from RNA-Seq, were plotted in a heatmap using the R function heatmap.2. The genes in the heatmap were first ordered by groups (C1–C5), and then by the control H3K4me3 signal within a 2000 bp region around the TSS (C1, C2, C3, and C5), or the signal of control H3K36me3 within the gene body (C5).

**Analysis of public ChIP-Seq datasets (ASH1L and HIF-1α).** The ChIP-Seq peak data for ASH1L in K562 cells and HIF-1α in PC-3 and K562 cells were obtained directly from ReMap2022[89]. The corresponding accession numbers in ReMap2022 and their original sources are: ASH1L in K562 cells (ENCODE: ENCSR115BBC), HIF-1α in K562 cells (GEO: GSE123461) and in PC-3 cells (GEO: GSE106305). Protein-coding genes (from GENCODE Release 37) with ASH1L or HIF-1α peaks located within −5 kb to +1 kb of their transcription start sites were identified as target genes of ASH1L or HIF-1α and used in the Venn diagrams presented in Supplementary Fig. 5. Additionally, raw fastq files of these datasets were downloaded from their original data sources and processed similarly as described above to generate the signal landscape.

**Single-cell RNA-seq and data analysis**
Control and ASH1L-knockout metastatic bone tumors were dissected, minced, and digested into a single-cell suspension using Collagenase A (Sigma, 11088793001) at 37 °C for 45 min with continuous gentle shaking. The dissociated tissues were centrifuged at 500×g for 5 min,

followed by further digestion with 0.25% Trypsin (Corning, 25053CI) at 37 °C for 5 min. The samples were filtered using a 70-uM strainer (Falcon, 352350) and centrifuged at 500×g at 4 °C for 5 min, followed by incubation in 1x RBC lysis buffer (Biolegend, 420302) at RT for 5 min. After washing with cold DMEM, the cells were collected and resuspended in cold PBS containing 0.04% BSA (Ambion, AM2616), followed by measurement of cell number and viability. Three control samples and four ASH1L-knockout samples (two samples per sgRNA) were submitted to Novogene (sgCtrl_1 and sgASH1L_4) or the Single Cell Genomics Core facility at MD Anderson (the rest of the samples) for library preparation and sequencing. The scRNA-seq libraries were generated using the 10X Genomics Chromium Controller Instrument and Chromium Single Cell 30 V3 Reagent Kits (10X Genomics) according to the manufacturer's recommendations. Libraries were subsequently sequenced on NovaSeq 6000 (Illumina) with a paired-end 50-base pair (50PE) reading strategy.

The raw data of scRNA-seq was processed by CellRanger 6.1.2 (10X Genomics) and then analyzed using Bioturing (https://academic.bioturing.com/login). Briefly, the Feature_matrix data of seven samples was uploaded to Bioturing with Mito filter 0.25, followed by batch harmony under the "Input Embedding" function. A total of 30,850 single cells with mitochondrial gene percentage <0.08 were subclustered for downstream analyses. UMAP (# of Neighbors = 25) or tSNE (Perplexity = 30) were generated and visualized in Vinci, followed by Louvain Clustering with a resolution of 0.3–0.8. Cell composition distribution of clusters among seven samples was generated under the "composition" function and visualized in Vinci, and a comparison of each cluster in control versus ASH1L-depleted samples was performed using GraphPad Prism version 9.2.0. The differential expression analyses of cells in each cluster versus remaining cells were performed using T-tests (reasoning that some clusters contain much fewer cells than others) (Supplementary Data 6, 7). Marker genes in each cluster were identified based on top DEGs and well-known cell-type markers in the literature, and then used for cluster nomination as well as violin plot and bubble heatmap plot generation. The DEGs in monocytes and TAMs from control versus ASH1L-depleted bone tumors were analyzed using the Wilcoxon method, followed by volcano plot generation using GraphPad Prism version 9.2.0. The DEGs (FDR <0.05; FC >1.5) in TAMs were used for pathway analysis using IPA. UMAP views color-coded by expression of genes of interest were generated in Vinci. OXPHOS pathway geneset (n = 43) was listed in Supplementary Data 10. Single-cell trajectory and pseudotime analysis of monocytes and TAMs were performed under the "Pseudotime" function.

**Clinical relevance analyses**
ASH1L putative copy number alterations and mRNA expression (FPKM capture) data on 444 metastatic PCa tumors (SU2C)[3] and 489 primary PCa tumors (TCGA)[90] were downloaded from cbioportal (www.cbioportal.org). The putative copy number alterations in the ASH1L gene include shallow deletion, diploid, gain, and amplification. The copy number alterations (CNA) of ASH1L in primary (TCGA) versus metastatic (SU2C) prostate tumors are presented. The expression levels of ASH1L in metastatic tumors (SU2C) containing ASH1L shallow deletion, diploid, gain, and amplification were compared using the unpaired Student's t-test. The survival data of metastatic PCa patients (SU2C datasets downloaded from cbioportal) with amplified or diploid ASH1L was compared using the log-rank (Mantel-Cox) test. The P value and hazard ratio (log-rank) were calculated using GraphPad Prism version 9.2.0. The correlations of high ASH1L expression with metastasis (Grasso dataset) in PCa were analyzed and visualized using the Oncomine™ Platform (Thermo Fisher, Ann Arbor, MI).

To determine the association between ASH1L high expression and metastatic signatures in human PCa, we downloaded the bulk RNA-seq RSEM dataset from the PCa TCGA project (493 cases). Samples were ranked by ASH1L expression and classified into three groups: ASH1L-

high (150 cases), ASH1L-medium (193 cases), and ASH1L-low (150 cases). The CHANDRAN METASTASIS gene set was analyzed in ASH1L-high versus ASH1L-low samples using GSEA 4.1.0. HIF-1α target gene enrichment analysis was performed using GSEA 4.1.0. Pearson correlation of mRNA levels between the two indicated genes were analyzed in PCa (TCGA dataset) or metastatic PCa (SU2C dataset) and downloaded from cbioportal. HIF-1α target geneset ($n = 110$) was listed in Supplementary Data 10.

The published scRNA-seq datasets of primary PCa ($n = 13$ patients; GSE141445) and metastatic PCa ($n = 13$ patients; GSE210358)[40,41] were analyzed using Bioturing (https://academic.bioturing.com/login). For 13 metastatic PCa samples, the samples were first processed by batch harmony under the 'Input Embedding' function. A total of 24,433 non-epithelial cells were used for UMAP generation and Louvain Clustering (resolution = 1). Cell composition distribution of clusters among 13 samples was generated under the "composition" function, and the comparison of lipid-associated-TAMs in ASH1L-high versus ASH1L-low groups was performed using GraphPad Prism version 9.2.0. Marker genes in each cluster were identified based on top DEGs and well-known cell-type markers and then used for cluster nomination as well as bubble heatmap plot generation. UMAP views, color-coded by gene expression and co-expression plots between ASH1L and HIF family genes were generated in Vinci. A similar method was used for analyzing 13 primary PCa samples. HIF-1α target geneset ($n = 110$) was listed in Supplementary Data 10.

CIBERSORT (Cell-type identification by estimating relative subsets of RNA transcripts) created by Newman et al., is an analytical tool that allows the abundance of member cell types in a mixed cell population to be estimated using gene expression data[91]. CIBERSORT gene signature matrix (LM22) contains 547 genes and distinguishes 22 human hematopoietic cell phenotypes, including 7 T cell types, naïve and memory B cells, plasma cells, NK cells, and myeloid subsets. The bulk RNA-seq and CNA datasets of 208 metastatic PCa samples[3] (SU2C dataset) were downloaded from cbioportal and uploaded to CIBERSORTx (https://cibersortx.stanford.edu) to generate the proportions of immune cells in metastatic PCa tumors. For statistical analysis, 22 immune cell populations were combined into nine major immune cell subtypes. Metastatic tumors containing ASH1L gene gain or amplification were classified as the "Amp/Gain" group, and tumors with diploid ASH1L were classified as the "Diploid" group. The immune cell proportion in the "Amp/Gain" versus "Diploid" groups was compared using the unpaired Student's t-test in GraphPad Prism version 9.2.0. Due to the nature of the PCa study, only male patient data were included for the clinical relevance study.

## Statistics and reproducibility

All data were presented as the mean ± standard deviation of at least three biological replicates. Statistical analyses were performed using GraphPad Prism version 9.2.0. No statistical method was used to predetermine the sample size. For animal experiments, each mouse was counted as a biologically independent sample. The mice were randomized before commencing an in vivo experiment. The investigators were not blinded to allocation during the experimental procedures and during outcome assessments. Mice's survival was determined by Kaplan–Meier analysis. Unpaired two-tailed Student's t-test (two groups) and one-way ANOVA with Tukey's post hoc test (multiple groups) were used for statistical analysis. Correlation analyses of gene expression were performed using Pearson correlation. Differentially expressed gene analysis in the scRNA-seq dataset was performed using two-tailed Wilcoxon tests or two-tailed T-tests. $P < 0.05$ was considered statistically significant. Exact P values are reported in each figure unless $P < 0.0001$. Each experiment was independently repeated at least three times and showed similar results when results from representative experiments were shown.

## Reporting summary

Further information on research design is available in the Nature Portfolio Reporting Summary linked to this article.

## Data availability

The raw data and processed data for scRNA-seq (GSE269895), bulk RNA-seq (GSE269830), and CUT&RUN-seq (GSE269829) generated in this study have been deposited in the GEO repository and released to the public. Public scRNA-seq data used in this study are available from GEO under the following accession codes: GSE141445 (scRNA-seq data of 13 primary PCa patients) and GSE210358 (scRNA-seq data of 13 metastatic PCa patients). The bulk RNA-seq dataset was downloaded from cBioPortal (https://www.cbioportal.org/). All data are available in the main text or the Supplementary Information. Source data are provided with this paper.

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

## Acknowledgements

The authors thank Dr. Ronald A. DePinho (MD Anderson) for sharing the PCa GEM models of PbPPS. The authors thank Leah Guerra for the histopathology analysis of bone metastasis in PC-3M bone metastatic samples. We also thank the MDACC Bone Histomorphometry Core Laboratory for TRAP and Toluidine blue staining in tumor-bearing bones. We acknowledge the Small Animal Imaging Facility, the Cytogenetic and Cell Authentication Core, and the ATGC NGS Core Facility at MD Anderson, which are supported by NIH/NCI R01 CA166051 (to ATGC). This research was also performed through the Single Cell Genomics Core Facility, which is supported in part by CPRIT Single Core grant RP180684 and NIH 1S10OD024977-01 (to ATGC). L.M. is supported by NIH/NCI R01 CA166051 and CA269140, American Cancer Society Grant DBG-22-161-01-MM, and the Nylene Eckles Distinguished Professorship of MD Anderson Cancer Center. D. Zhao is a CPRIT Scholar in Cancer Research and has been supported by the CPRIT Recruitment of First-Time Tenure-Track Faculty Award RR190021, NIH/NCI R01 CA275990 and CA278889, Prostate Cancer Foundation Challenge Award FP00016492, and DoD CDMRP IDA award PC230358.

## Author contributions

C.M. and D.Z. designed the research, analyzed the results, and wrote the manuscript. C.M. performed most of the experiments. D.Z. performed scRNA-seq analyses of bone metastasis samples, clinical relevance analyses, and pathway analyses. K.L. and Y.L. performed the bioinformatics analysis for RNA sequencing, CUT&RUN-sequencing, and public ChIP-sequencing. W.S. performed scRNA-seq analyses in human PCa datasets and stable cell line generation. H.T. helped with part of the intracardiac injection. X.W. assisted in analyzing bone volume fraction and osteoblast quantification of the murine bone metastasis models. Y.W. and X.L. provided mouse husbandry for GEM models. A.D., J.L., F.C., and Q.G. provided technical support. J.Z. helped with the preparation of pathological samples. V.V. and K.M. provided training in Bioluminescence and X-ray imaging. B.G., L.M., and Y.L. provided intellectual contributions throughout the project. D.Z. and Y.L. jointly supervised this work.

## Competing interests

The authors declare no competing interests.
