## [Transparent Peer Review file · Nature Communications]

Histone Methyltransferase ASH1L Primes Metastases and Metabolic Reprogramming of Macrophages in the Bone Niche

Corresponding Author: Dr Di Zhao

Version 1:

Reviewer comments:

Reviewer #1

(Remarks to the Author)

This interesting manuscript by Meng et al describes extensive studies into the role of ASH1L in prostate cancer metastasis and its ability to cause metabolic reprogramming of macrophages in bone. Loss and gain of function experiments in pre-clinical models showed that ASH1L played a role in invading cancer cells by increasing lipid-associated macrophages (LA-TAMs) and maintained their pro-tumoral and anti-inflammatory phenotype in the bone niche. The activation of LA-TAMs appeared to be due to the reprogramming of oxidative phosphorylation via increased ASH1L-dependent expression of IGF-2 in the tumor cells. Given that the role of ASH1L in metastasis is unknown, the findings are novel and important. The pre-clinical studies are generally convincing, well-designed and the manuscript is mostly clear, however further clarification is required, particularly regarding the analysis and interpretation of data with clinical specimens, and more data are necessary to draw specific conclusions about the effect of tumors on bone parameters in vivo. Specific concerns are listed below:

Major points:

1. The definition of ASH1L “high and low” human PCa samples is subjective and the two groups are not clearly defined in Fig. S1D,E and Fig. 3H. The expression level of ASH1L from both the bulk RNAseq and scRNAseq data of each patient should be included to clearly distinguish the ASH1L-high and ASH1L-low groups, with statistical analysis of differences in expression.
2. Is ASH1L expression in epithelial cells really different between the “high and low” groups of metastatic tumors shown in Figure S9I? Without statistical analysis to support the definition of these two groups, their distinction may not be meaningful.
3. The authors claim that growth of DX1 tumors in bone displayed mixed osteolytic/osteoblastic features, but Figure 4C,D is insufficient to show this. Similarly, ASH1L KO in the DX1 model is claimed to lead to a decrease in bone formation (Page 11 line 263-266), but the images shown in Figure 4B-D (of one representative bone from each group) are not sufficient without proper quantification and statistical analysis. The authors should consider using standard histomorphometric approaches to measure bone formation and resorption (including assessment of bone volume, osteoblast and osteoclast number, bone formation rate) ideally combined with micro-CT analysis. Similarly, it is not clear what Figure S6E-F actually shows without appropriate labelling and quantification.
4. In Fig. 7B-E the authors claim that ASH1L only affected bone metastasis in the presence of TAMs. However, the anti-CSF1R treatment also affected metastatic outgrowth in the shCtrl mice. Hence, the lack of effect in shASH1L vs shCtrl in the anti-CSF1R group may be due to the already very low tumour burden resulting from the depletion of CSF1R-positive cells.

Minor points:

1. Histological data of 1C and 1H should be quantified, or at the very least more examples should be shown.
2. The intraosseous model used in Fig. 1J and Fig. 4 is a model of tumor outgrowth in bone, but it does not represent bone metastasis, since it does not involve any of the processes involved in systemic dissemination and homing to the skeleton. The authors should clarify that this model represents the outgrowth of tumors in bone, rather than referring to it as a model of bone metastasis. Did the authors attempt to use intracardiac injection for the ASH1L-F3 cells and the syngeneic DX1 model? If not what are the reasons for direct intratibial injection?

3. The quantification of the immunofluorescence data in Fig. 5H and Fig. 7E were not clearly explained. Why were there more than 3 data points if the sample size was 3 mice? Are the data points technical replicates of individual views? If so, the mean of those measurements should be used for each mouse.
4. The markers for all clusters in Fig. 4E should be included to further support the annotation of immune cells. The MDSC clusters (C7 and C8) appeared to be very similar to what would conventionally be annotated as neutrophils, unless there are specific markers distinguishing them as MDSCs? C6 are actually pre-neutrophils due to expression of early markers e.g Mpo and Elane. Similarly, annotation of the high-resolution clustering of the monocytes and macrophages in Fig. 5 needs further clarification
6. The markers used for qPCR in Fig. 5I are not the same as those identified in scRNAseq. For example, for lipid-associated TAMs CD163 was not differentially expressed in MC5_Lipid-associated TAM and IL10 was moderately upregulated. Wouldn't Trem2, Mrc1 and Apoe be better markers to quantify? The same argument applies to CD80 for inflammatory TAMs, which was not differentially expressed in MC6.
7. The signature genes of oxphos pathway used in Fig. 6B should be disclosed.
8. More detail should be included in the Methods section to explain how the immune cell composition analysis was performed using CIBERSOFT for Fig. S9J. The data of the 208 metastatic PCa samples should also be more clearly described – which study was this data derived from?
9. The discussion section is rather long and could be more concise.

Reviewer #2

(Remarks to the Author)

The manuscript by Chenling Meng et al. presents different pieces of evidence in support of a role of the methyltransferase ASH1L in the formation of bone metastasis. They principally used a genetically engineered mouse model of prostate cancer, a xenograft metastasis model of prostate cancer in immunosuppressed (nude) mice, as well as a syngeneic mouse model of prostate cancer metastases in bones.

First, the authors show that the ASH1L is frequently amplified and overexpressed in metastatic cancer cells. Second, this ASH1L gain of function results in increased histone methylation. Third, ASH1L interacts with HIF-1alpha to induce a gene expression profile that is favorable to metastasis. Forth, cancer cells overexpressing ASH1L induce monocyte differentiation into tumor-associated macrophages, which are known to stimulate tumor progression. This effect is mediated by IGF-2. Fifth, in human cancer, ASH1L overexpression correlates with the presence of tumor-associated macrophages. Finally, in preclinical models, inhibition of IGF-2 or inhibition of ASH1L suppresses the ability of cancer cells to form metastases.

1. The possibility that ASH1L is a driver of metastasis is an appealing observation. However, gene amplifications are very common events during the progression from primary cancer to metastases. Therefore, it is very important to unequivocally show that the described effects are caused by ASH1L itself and not by accompanying genetic changes. It is also possible that ASH1L overexpression generally increases the cell fitness required to occupy metastatic niches through overall stimulation of gene expression without any specific mechanistic pathway. For that reason and considering the potential of off-target effects in gene deletions, I propose to confirm the data of Figure 1 with a true "rescue" experiment, where the endogenous ASH1L gene deletion in PC-3M cells is reversed by expression of the full-length protein (or fragment F3) from an integrated vector construct. The gain-of-function effects observed in LNCaP cells provides additional circumstantial evidence but is not a substitute for rescue experiments.

2. For the same reasons as above, across the manuscript, experiments should be carried out with a catalytically inactive mutant of ASH1L (or of fragment F3) to confirm the hypothesis that the effects of ASH1L occur through protein methylation. This concern also applies to the DX1 cells used in the syngeneic model of bone metastases.

Further comments:

3. How do the profiles of CUT&RUNseq compare with those of canonical histone-ChIPseq? For the interpretation of findings, it would be useful to confirm that the two methods are equivalent. This would be important particularly for the H3K36me2/3 marks that are more broadly distributed across gene bodies. In this context, I would urge to be more cautious as to the causative link between histone methylation changes and transcription. Perhaps, histone methylation effects are a consequence rather than the cause of active transcription.

4. How do we know if ASH1L methylates only histones or whether the pro-metastatic effects are triggered by direct methylation of transcription factors or other proteins?

5. For the dislocation of a self-inhibitory loop, ASH1L requires the regulatory co-factor MRG15. What is the stoichiometry of ASH1L and MRG15 in metastatic cancer compared to the primary cancer counterparts? Is the ASH1L overexpression accompanied by an MRG15 overexpression?

6. Figure 3G shows that HIF-1alpha and ASH1L co-exist in the same protein complex but does not prove that ASH1L

“interacts” with HIF-1alpha. Biochemical experiments with recombinant proteins or protein fragments are necessary to demonstrate a direct interaction. Accordingly, the model of Figure 8 is a working model (or hypothesis), not a “summary”.

7. The terms co-opt and cooperate are very vague. What does it mean that ASH1L and HIF-1alpha “cooperate”? Do ASH1L and HIF-1alpha co-localize across chromatin in ChIP-seq experiments? Does ASH1L locate preferentially in the promoter of HIF-dependent genes? Is there a methylation of HIF-1alpha by ASH1L?

8. Please clarify if IGF-2 is also induced by HIF and if the pro-metastatic macrophages are also enriched in response to HIF activation.

9. The quantification of metastatic tumors in Figure 7G shows only 5 control (vehicle) animals out of a total of 10 animals. What happened with the analysis of the missing 5 animals? Why is their tumor cell load not shown?

Minor comments:

10. In the Western blot of Figure 1D, there is some residual ASH1L in the knockout cells. Please explain the reason. Are these cells able to reamplify the gene during repeated passages?

11. Please describe the genotype of the “nude” mouse used in the xenograft metastasis model.

12. Line 129: should inhibition be replaced by deletion?

Reviewer #3

(Remarks to the Author)

Reviewer #4

(Remarks to the Author)

Bone metastasis is associated with high rates of mortality in cancers including prostate, breast, lung, kidney, ovarian, and melanoma, with 80% of prostate cancer patients developing painful osteoblastic bone lesions. Despite a known association between alterations in histone methyltransferases and metastatic disease across cancer types, epigenetic determinants of bone metastases, and the mechanisms governing this, are not well understood. Meng and colleagues identify prostate cancer cell expressed ASH1L as an epigenetic driver of tumor cell invasion and outgrowth. Furthermore, they demonstrate that ASH1L leads to upregulation of HIF1 target genes, including IGF2, and that deletion of ASH1L in tumor cells is associated with a reduction in tumor-associated macrophages (TAMs) within the tumor microenvironment. They go on to conclude that secreted IGF2 signals to macrophages, promoting a pro-tumoral lipid-associated TAM phenotype through OXPHOS. Using small molecule inhibitors, they demonstrate that ASH1L can be targeted to reduce prostate tumor burden in the bone and improve overall outcome in a pre-clinical model. While the authors combine several omics strategies with mouse models and in vitro studies, there are some issues that need to be addressed prior to publication:

- Throughout the text, the authors ascribe their findings that ASH1L is an epigenetic regulator of prostate tumorigenesis to generic “metastatic malignancies” and “cancers”. It is recommended that specific text is used instead to refer to their data-supported conclusions within prostate cancer
- Figure 1H requires quantification and statistics to support claims that “ASH1L-depleted PC-3M cells had impaired capacities to colonize the bone and form osteolytic lesions”
- Figures 4C, and 4D require quantification and statistics to support claims that intratibial injected DX1 cells “produce osteolytic/osteoblastic features”
- The representative images in Figure S9E do not reflect the data in the matched main figure 7I
- It is recommended to include percentages in Fig 1A either in the panel or within the legend
- Please include brightfield images for Figs S1G, S2I, S6B, S6E
- Please include information on how the in vitro proliferation assays were performed in the Methods section
- Intratibial injections do not model the metastatic process of invasion, intravasation, or extravasation but instead model outgrowth in the metastatic target tissue. It is recommended that careful, clear language be used in describing this model, reaching conclusions from the data generated using this model, and in the design of Figure 8. It should also be noted that intratibial injections cause massive inflammation which activates and enhances macrophage populations, a consideration for data interpretation
- If IVIS images were available during tumor growth time course corresponding to Figures 1J, 4B, 7B, a tumor growth curve would be a nice addition
- For figure 3d, why not look at cut/run seq for these genes as done in figure 2h
- Figure 3d, only one panel is sg#3 the rest are sg#2 – why? In addition, please report R2 values
- Figure S5 – the y axis is cropped
- When discussing the validation efforts in figure 5H with CD206 staining – it was confusing why this marker was chosen. It would be helpful to the reader if introduced on line 317 that the pro-tumoral gene Mrc1 identified in cluster MC5 is transcribed into the protein called CD206
- Representative images in Figure S7E do not reflect the changes shown in the corresponding bar graphs
- Please use clear language when describing how the loss of ASH1L in tumor cells effects the macrophage populations. An

example is line 370 when the authors write "..., indicating ASH1L promoting the proliferation and survival of TAMs". This becomes confusing when conceptualizing the mechanism

- On line 387 the authors refer to Figure 6A to conclude that "...TAMs from ASH1L-depleted bone tumors exhibited decreased IGF uptake...", however Figure 6A shows pathway analysis and therefore can only support an inference via IGF downstream signaling
- Throughout the text, the authors refer to their fragment 3 expression of ASH1L (see Fig S3A) as "ASH1L overexpression". The full length protein is not being expressed in this model, however. Please clarify in text
- On line 410 the authors write "we also found that IGF-2 expression was strongly associated with lipid-associated TAM markers and pro-tumoral phenotype markers" in reference to Figures S8B,C. However, the Pearson's correlations range between 0.29 and 0.4. Please revise language to reflect the weaker correlation that is shown
- In reference to the data presented in Figure 7K,L, a correlation plot would be a nice addition to evaluate ASH1L expression and lipid-associated TAMs, agnostic of binning
- For all stains where images were used for quantification, please include secondary only controls in the supplement, split by channels

Reviewer #5

(Remarks to the Author)

ASH1L is a histone methyltransferase. In this manuscript, the authors report that ASH1L drives the metastatic progression of prostate cancers via co-op with hypoxia/HIF1A to induce pro-metastatic transcriptome and to promote monocyte differentiation to pro-tumor and anti-inflammatory macrophages via IGF2. Although it is known that hypoxia upregulate IGFs, which may induce immune repressive macrophages, this study provides more mechanistic details. Overall, this is an excellent study, and will lead to new understanding in how tumor cells, especially prostate cancer, develop immune repressive metastasis via ASH1L and HIF1A. The strength of this manuscript includes 1) the combinational use of single-cell and bulk sequencing data to develop new insights of ASH1L interaction with hypoxia/HIF1A and the consequences on metastatic niche and monocyte/macrophage differentiations, 2) strong gain/loss-function studies of ASH1L in vitro and in metastatic mouse models (intracardiac and intratibial). A major weakness is that, in the 2nd half of the study, the role of HIF1A in ASH1L-driven macrophage plasticity is less established. It is unclear whether the ASH1L/IGF2/pro-tumor and anti-inflammatory macrophage axis requires HIF1A or hypoxic microenvironment.

Major concerns:

1. Hypoxia is known to upregulate IGFs. However, the role of hypoxia/HIF1A in facilitating the ASH1L/IGF2/TAM axis is unclear. In the first half of the manuscript, the authors provide strong data in establishing that ASH1L promotes metastatic gene expression in the presence of HIF1A. However, it is less clear whether the communication between tumor cells and macrophage cells still requires hypoxia/HIF1A. The role of HIF1A in ASH1L-driven IGF2 mRNA and TAM differentiation is not specifically tested in Figures 4-7.
2. The involvement of HIF1A makes sense in PC3M cells, since HIF1A protein is present in non-hypoxic condition in this cell line. However, it may also cast uncertainty of the involvement of this gene/protein in other prostate cancer cells or cell lines in non-hypoxic conditions. Most of the prostate cancer cell lines e.g., LNCaP, have very low or none-detectable levels of HIF1A proteins in non-hypoxic conditions. In vivo, at least in primary prostate cancers, the impact of hypoxia was reported as low based on hypoxia-signature genes via bulk RNA-Seq (Bhandari et al, Nat Genet, 2019). Thus, it will be highly informative to use the single-cell sequencing data from the 13 patients and the mouse models to evaluate the involvements (requirements) of hypoxia and/or HIF1A in subpopulations of the tumor and macrophage cells in Figures 4, 5, and 7.

Minor concerns:

1. Figure 3 shows the interaction between ASH and HIF1 to regulate gene expression. LNCaP is used for Fig 3F, however, HIF1A protein is expressed at very low level in non-hypoxic condition in this cell line, thus the question is whether ASH1L overexpression bypasses the requirement of HIF1A to upregulate these genes (unlikely based on results in Fig 3I). The level of HIF1A protein in 3I is needed.
2. Most metastatic prostate cancer cells express androgen receptor (AR). However, PC3 cells, which do not express AR, are used in many cell/xenograft models. In TCGA and SU2C analysis (Fig 1A), what are the association between ASH1L and AR and AR activity? Further, AR-expressing LNCaP has a metastatic origin, however, the ASH1L mRNA level is low, it is unclear how this gene (expression) is regulated.
3. Some of the single-cell analyses require quantification, e.g., Figures 3H and 6B. Also, the exact identities of genes used for these analyses are required.
4. IGF2 was identified via bulk sequencing data (Fig 6C). What are the levels of IGF2 in single-cell sequencing of the human and mouse tumors?
5. For the intratibial models in figure 1J and 4B, it would be interesting to see the bone tumor signals over several time points.
6. The exact molecular mechanism, by which IGF2 promote immune repressive phenotype via OXPHOS remains unclear.

Version 2:

Reviewer comments:

Reviewer #1

(Remarks to the Author)

In this revised version of the manuscript, the authors have answered each of our comments. The additional data provided by the authors has addressed the major concerns and has substantially improved the manuscript. However, further clarification is needed regarding the methods used for bone histomorphometric analysis and data presentation:

1. The Y axes of Suppl. Fig. 5i and Suppl. Fig. 10d should be labelled clearly (ASH1L expression?). We strongly suggest that both graphs should be plotted, and statistics performed, using normalised count/expression level of ASH1L on a per patient basis, instead of per cell basis. The latter can underestimate variability and overstate statistical significance.
2. The addition of quantitative data is an improvement in confirming the effects of tumor on bone parameters in a ASH1L-dependent manner. However, the methods used for the quantitative analysis are not entirely clear. In Fig. 4 and Suppl.Fig. 6, the regions of interest (ROI) for cortical and woven bone BV/TV analysis are not defined. What were the criteria for selecting these ROIs? Also, the meaning of "# osteoclasts/osteoblasts per view" is unclear. How many "views" were analyzed, and what is the ROI? It may be more appropriate to quantify osteoclasts/osteoblasts per unit of bone surface (number/mm).
3. The histological images in Fig. 4C are shown at different magnifications. It would be helpful to define which areas were magnified, to guide readers in focusing on the relevant areas.
4. In Figs. 5h and 7e the authors clarified that data points and statistical analyses were based on technical replicates that were pooled from all 3 mice (therefore >10 data points instead of just 3). It is debateable whether the use of pooled data is appropriate. While the authors stated that this method of analysis was included in both the methods section and figure legend, it was not found in the latter. Please include this information in the figure legend to make it clear that the data shown are technical replicates pooled from 3 mice.

Reviewer #2

(Remarks to the Author)

The reviewer acknowledges that major issues are now eliminated by additional experimental data that strengthen the manuscript.

Reviewer #3

(Remarks to the Author)

Reviewer #4

(Remarks to the Author)

The revised manuscript contains new rescue experiments and loss/gain of function experiments, which strengthen the conclusions that ASH1L directly contributes to tumor cell invasion and migration. Furthermore, the addition of experiments using the HIF1a inhibitor PX-478 demonstrate the effects of ASH1L-driven tumor outgrowth and pro-tumoral TAM accumulation are dependent on HIF1a activity. This supports the mechanism described in the second half of the manuscript. There are a few minor points to address however.

1. Line 157 – the authors write "Functionally, overexpression of ASH1L-F3, not F1 or F2, can significantly augment cell migration and invasion (Supplementary Fig. 3f-h)". However, Supplementary Figure 3 does not show invasion data from cells expressing F1 or F2 (as for migration in supp fig 3g). It is recommended to change the language in text or include the data indicating no changes to cell invasion in cells with F1/F2.
2. The changes on the western blot in supplementary figure 8e are extremely slight. Please provide appropriate densitometric & statistical analysis supporting the claim in lines 455-456 that AKT-GSK3b-mTOR signaling is activated.
3. We thank the authors for including the consideration that intratibial injection-induced inflammation may influence macrophage behavior into lines 572-579 in the discussion section. However, rather than copying/pasting our suggestion into the discussion, the authors should instead consider how inflammation resulting from their model may activate/recruit macrophages with appropriate citations.

Reviewer #5

(Remarks to the Author)

The revision has adequately addressed all my concerns.

Version 3:

Reviewer comments:

Reviewer #1

(Remarks to the Author)

The authors have further improved the manuscript by providing additional methodological details and reanalysis/alterd presentation of some of the data. Previous concerns have now been addressed and there are no additional comments.

Reviewer #3

(Remarks to the Author)

Reviewer #4

(Remarks to the Author)

REVIEWER COMMENTS

Reviewer #1 and #3 (Remarks to the Author): expert in TAMs, immunometabolism

This interesting manuscript by Meng et al describes extensive studies into the role of ASH1L in prostate cancer metastasis and its ability to cause metabolic reprogramming of macrophages in bone. Loss and gain of function experiments in pre-clinical models showed that ASH1L played a role in invading cancer cells by increasing lipid-associated macrophages (LA-TAMs) and maintained their pro-tumoral and anti-inflammatory phenotype in the bone niche. The activation of LA-TAMs appeared to be due to the reprogramming of oxidative phosphorylation via increased ASH1L-dependent expression of IGF-2 in the tumor cells. Given that the role of ASH1L in metastasis is unknown, the findings are novel and important. The pre-clinical studies are generally convincing, well-designed and the manuscript is mostly clear, however further clarification is required, particularly regarding the analysis and interpretation of data with clinical specimens, and more data are necessary to draw specific conclusions about the effect of tumors on bone parameters in vivo. Specific concerns are listed below:

Response: We thank reviewers #1 and #3 for their nice summary and positive feedback! We also appreciate their constructive comments, which significantly helped us improve the manuscript. Please find our point-by-point responses below:

Major points:

1. The definition of ASH1L “high and low” human PCa samples is subjective and the two groups are not clearly defined in Fig. S1D,E and Fig. 3H. The expression level of ASH1L from both the bulk RNAseq and scRNAseq data of each patient should be included to clearly distinguish the ASH1L-high and ASH1L-low groups, with statistical analysis of differences in expression.

Response: To distinguish the ASH1L-high and ASH1L-low groups clearly, we analyzed the ASH1L expression levels of the bulk RNA-seq dataset of TCGA prostate cancer (**Supplementary Fig. 1d; Line 101**) and scRNA-seq dataset of epithelial cells from 13 patients with primary prostate cancer (**Supplementary Fig. 5i; Line 265**). Statistical analysis showed significant differences in ASH1L expression in “ASH1L-high” versus “ASH1L-low” groups. The new data are presented below:

Supplementary Fig. 1d. Based on ASH1L mRNA expression, 493 PCa samples in the TCGA dataset were classified into ASH1L_high (n=150), ASH1L_mid (n=193), and ASH1L_low (n=150) groups. The enrichment of metastasis profile in ASH1L_high versus ASH1L_low samples was determined by GSEA. The normalized Enrichment Score (NES) and Nominal p-value (NOM p) are shown.

Supplementary Fig. 5i. 13 patients with primary PCa (GSE141445) were classified into two groups (ASH1L_Low vs ASH1L_High) by ASH1L expression levels in epithelial cells.

2. Is ASH1L expression in epithelial cells really different between the “high and low” groups of metastatic tumors shown in Figure S9I? Without statistical analysis to support the definition of these two groups, their distinction may not be meaningful.

Response: We analyzed the expression level of ASH1L from the scRNA-seq dataset of epithelial cells from 13 patients with metastatic prostate cancer (**Supplementary Fig. 10d; Line 516**). Statistical analysis showed significant differences in ASH1L expression in “ASH1L-high” versus “ASH1L-low” groups. The new data are presented below:

3. The authors claim that growth of DX1 tumors in bone displayed mixed osteolytic/osteoblastic features, but Figure 4C,D is insufficient to show this. Similarly, ASH1L KO in the DX1 model is claimed to lead to a decrease in bone formation (Page 11 line 263-266), but the images shown in Figure 4B-D (of one representative bone from each group) are not sufficient without proper quantification and statistical analysis. The authors should consider using standard histomorphometric approaches to measure bone formation and resorption (including assessment of bone volume, osteoblast and osteoclast number, bone formation rate) ideally combined with micro-CT analysis. Similarly, it is not clear what Figure S6E-F actually shows without appropriate labelling and quantification.

Response: We typically use Bioquant Osteo II software for bone histomorphometry analysis within defined regions of interest (ROIs) in bone marrow cavities. We quantify osteoblasts and osteoclasts around the trabecular bone. However, as observed with most mouse-derived tumor lines, the DX1 tumor cells are highly aggressive and have destroyed all trabecular bone in the bone marrow cavities in some tumor-bearing bone samples. This damage made it challenging to analyze the images using Bioquant software in the traditional way. Interestingly, we observed that the tumor-induced the formation of new woven bone along the outer surfaces of the cortical bone. This finding suggests that DX1 tumors simultaneously produce both osteolytic and osteoblastic lesions in different locations. Then, we performed TRAP and Toluidine blue staining on DX1 bone-outgrown tumors, as shown in **Fig. 4 and Supplementary Fig. 6**. For the analyses of bone volume, osteoblast, and osteoclast, we found that QuPath, an open-source AI software (PMID: 38625969), is better suited for quantification and segmentation in this context after comparison. Even so, we applied the same criteria in BioQuant to identify osteoclasts, including TRAP staining positivity, multiple nuclei, and proximity to bone surfaces. For osteoblasts, we only counted active cuboidal-shaped cells, excluding spindle-shaped ones.

As shown in the revised figures (**Fig. 4c-g and Supplementary Fig. 6a, i-k; Line 294-299**), histopathological analysis revealed significant absorption of cortical bone and remarkable new woven bone production in the outgrown tumors, along with enriched osteoclasts and osteoblasts,

confirming that DX1 tumors in bone displayed mixed osteolytic/osteoblastic features. Besides, ASH1L depletion in invading cancer cells protected cortical bone from absorption and reduced new woven bone production (Fig. 4c-e and Supplementary Fig. 6d). Both osteoclasts and osteoblasts were decreased in bone-outgrown tumors upon ASH1L deletion (Fig. 4c,f,g). Similar effects were observed when knocking down ASH1L using shRNA (Supplementary Fig. 6h-k). Together with our findings in metastasis xenograft models, these results in the newly developed syngeneic model establish ASH1L as an essential epigenetic determinant in metastatic tumor outgrowth in the bone (Line 305-310).

4. In Fig. 7B-E the authors claim that ASH1L only affected bone metastasis in the presence of TAMs. However, the anti-CSF1R treatment also affected metastatic outgrowth in the shCtrl mice. Hence, the lack of effect in shASH1L vs shCtrl in the anti-CSF1R group may be due to the already very low tumour burden resulting from the depletion of CSF1R-positive cells.

Response: Our mechanistic studies in **Fig. 5 and 6** demonstrate that ASH1L-IGF2 induces TAM plasticity in metastatic bone niches. In **Fig. 7b-e**, we aimed to determine if TAMs are required for the pro-metastatic outgrowth role of ASH1L by depleting TAMs using anti-CSF1R antibodies. Consistent with our hypothesis, ASH1L knockdown or TAM depletion alone inhibited metastatic outgrowth of prostate tumors; however, ASH1L knockdown has little impact on metastatic outgrowth in TAM-depleted tumors, suggesting TAMs play a key role in mediating the effects of ASH1L. We understand the concerns about the small size of tumors after TAM depletion, but this strong phenotype underscores TAMs' essential role in ASH1L-induced metastatic outgrowth.

Minor points:

1. Histological data of 1C and 1H should be quantified, or at the very least more examples should be shown.

Response: IHC staining of ASH1L in **Fig. 1c** was performed in murine primary prostate tumors versus lymph node metastases three years ago. Unfortunately, this IHC-qualified ASH1L antibody was discontinued. To verify this result, we performed Western blot assays of ASH1L in additional samples, and the data was shown in **Supplementary Fig. 1h**.

Regarding Fig. 1h, in the revised manuscript, we clearly labeled the bone and tumor regions in the representative images (**Fig. 1h**) and analyzed the bone metastasis rates in control versus ASH1L-knockout groups in the main text (**Line 132**). In addition, we quantified the osteoclast and osteoblast in metastatic tumors and found that ASH1L depletion significantly suppressed the abundance of osteoclast but had little impact on osteoblast (**Supplementary Fig. 2j; Line 135**), consistent with the osteolytic feature of PC-3M-derived metastasis models.

2. The intrasosseous model used in Fig. 1J and Fig. 4 is a model of tumor outgrowth in bone, but it does not represent bone metastasis, since it does not involve any of the processes involved in systemic dissemination and homing to the skeleton. The authors should clarify that this model represents the outgrowth of tumors in bone, rather than referring to it as a model of bone

metastasis. Did the authors attempt to use intracardiac injection for the ASH1L-F3 cells and the syngeneic DX1 model? If not what are the reasons for direct intratibial injection?

Response: We thank this reviewer for pointing this out! In the revised manuscript, we changed the description from “bone metastasis” to “tumor outgrowth in bone” in the main text and figure legends related to intratibial injection-derived models. Previously, we tried intracardiac injection of DX1 cells in B6 mice, but there was no bone metastasis. Considering LNCaP cells have a low capacity for metastasizing to bone, we didn’t use intracardiac injection for the ASH1L-F3 LNCaP cells. Although the intraosseous model does not involve any of the processes involved in systemic dissemination and homing to the skeleton, it remains a standard tool for studying bone metastasis, particularly for investigating the interaction between cancer cells and bone niche.

3. The quantification of the immunofluorescence data in Fig. 5H and Fig. 7E were not clearly explained. Why were there more than 3 data points if the sample size was 3 mice? Are the data points technical replicates of individual views? If so, the mean of those measurements should be used for each mouse.

Response: In Fig. 5h and Fig. 7e, we performed multiplex IHC staining using indicated TAM markers in bone outgrowth tumors from three mice per group. The stained slides were scanned by Vectra Polaris (Akoya Biosciences). We randomly selected 3-5 individual views on each slide, and all views from 3 mice were used for quantification, statistical analysis, and generating the data plot. The description of multiplex IHC staining data analysis in the figure legends and Method section has been revised for clarification (Line 856).

4. The markers for all clusters in Fig. 4E should be included to further support the annotation of immune cells. The MDSC clusters (C7 and C8) appeared to be very similar to what would conventionally be annotated as neutrophils, unless there are specific markers distinguishing them as MDSCs? C6 are actually pre-neutrophils due to expression of early markers e.g Mpo and Elane. Similarly, annotation of the high-resolution clustering of the monocytes and macrophages in Fig. 5 needs further clarification

Response: As suggested, we annotated the C6 cluster as Pre-Neutrophil and C7/C8 clusters as MDSC-1/2 in the revised Fig. 4h and the related main text (Line 337). The bubble plot in Fig. 4i presented the marker gene expression for all clusters, along with a detailed description of markers used for annotating each cluster in the main text (Line 334-339). The differentially expressed genes were also listed in the revised Supplementary Table 6.

For monocytes and macrophages, we annotated seven subclusters (MC1-7) based on markers published previously and our differential gene expression analyses (Fig. 5b). There is a paragraph (Line 354-371) in the revised manuscript to introduce the phenotype and functional markers in each subcluster. The differentially expressed genes were also listed in the revised Supplementary Table 7.

6. The markers used for qPCR in Fig. 5I are not the same as those identified in scRNAseq. For example, for lipid-associated TAMs CD163 was not differentially expressed in MC5_Lipid-associated TAM and IL10 was moderately upregulated. Wouldn't Trem2, Mrc1 and Apoe be better markers to quantify? The same argument applies to CD80 for inflammatory TAMs, which was not differentially expressed in MC6.

Response: As suggested, we determined lipid-associated TAM markers (such as APOE and TREM2) and inflammatory TAM markers (such as IRF1 and JAK2) in human monocytes cultured in conditional medium or THP-1 cells cocultured with control or ASH1L-depleted prostate cancer cells. The results indicated that ASH1L depletion in cancer cells significantly suppressed the expression of lipid-associated TAM markers but induced inflammatory TAM marker expression (Fig. 5i and Supplementary Fig. 7g; Line 392-299). Together with scRNA-seq analysis in Fig. 5, this validation using in vitro co-culture systems strengthens our conclusion that ASH1L in invading cancer cells promotes monocyte differentiation into lipid-associated and angiogenic TAMs and reprogramming them toward pro-tumoral and anti-inflammatory states.

7. The signature genes of oxphos pathway used in Fig. 6B should be disclosed.

Response: 43 genes in the OXPHOS pathway were used to generate the colored UMAP in Fig. 6B. This geneset has been disclosed in the revised **Supplementary Table 10**.

8. More detail should be included in the Methods section to explain how the immune cell composition analysis was performed using CIBERSOFT for Fig. S9J. The data of the 208 metastatic PCa samples should also be more clearly described – which study was this data derived from?

Response: The Method (**Line 1040**) has been rewritten to include more details in immune cell composition analysis and the datasets of 208 metastatic PCa samples. Please also see below:

“CIBERSORT (Cell-type Identification by Estimating Relative Subsets of RNA Transcripts) created by Newman et al., is an analytical tool that allows the abundance of member cell types in a mixed cell population to be estimated using gene expression data⁸⁷. CIBERSORT gene signature matrix (LM22) contains 547 genes and distinguishes 22 human hematopoietic cell phenotypes, including 7 T cell types, naïve and memory B cells, plasma cells, NK cells, and myeloid subsets. The bulk RNA-seq and CNA datasets of 208 metastatic PCa samples³ (SU2C dataset) were downloaded from cbiportal and uploaded to CIBERSORTx (<https://cibersortx.stanford.edu>) to generate the proportions of immune cells in metastatic PCa tumors. For statistical analysis, 22 immune cell populations were combined into nine major immune cell subtypes. Metastatic tumors containing ASH1L gene gain or amplification were classified as the “Amp/Gain” group, and tumors with diploid ASH1L were classified as the “Diploid” group. The immune cell proportion in the “Amp/Gain” versus “Diploid” groups was compared using the unpaired Student’s t-test in GraphPad Prism version 9.2.0.”

9. The discussion section is rather long and could be more concise.

Response: The discussion section has been trimmed from 6 pages to 4 pages. The revised discussion section looks more concise (**Line 529-626**).

Reviewer #2 (Remarks to the Author): expert in epigenetics and ASH1L

The manuscript by Chenling Meng et al. presents different pieces of evidence in support of a role of the methyltransferase ASH1L in the formation of bone metastasis. They principally used a genetically engineered mouse model of prostate cancer, a xenograft metastasis model of prostate cancer in immunosuppressed (nude) mice, as well as a syngeneic mouse model of prostate cancer metastases in bones.

First, the authors show that the ASH1L is frequently amplified and overexpressed in metastatic cancer cells. Second, this ASH1L gain of function results in increased histone methylation. Third, ASH1L interacts with HIF-1 α to induce a gene expression profile that is favorable to metastasis. Forth, cancer cells overexpressing ASH1L induce monocyte differentiation into tumor-associated macrophages, which are known to stimulate tumor progression. This effect is mediated by IGF-2. Fifth, in human cancer, ASH1L overexpression correlates with the presence of tumor-associated macrophages. Finally, in preclinical models, inhibition of IGF-2 or inhibition of ASH1L suppresses the ability of cancer cells to form metastases.

Response: We appreciate this reviewer's nice summary, positive feedback, and constructive comments that helped us improve the manuscript. Please find our point-by-point responses below:

1. The possibility that ASH1L is a driver of metastasis is an appealing observation. However, gene amplifications are very common events during the progression from primary cancer to metastases. Therefore, it is very important to unequivocally show that the described effects are caused by ASH1L itself and not by accompanying genetic changes. It is also possible that ASH1L overexpression generally increases the cell fitness required to occupy metastatic niches through overall stimulation of gene expression without any specific mechanistic pathway. For that reason and considering the potential of off-target effects in gene deletions, I propose to confirm the data of Figure 1 with a true "rescue" experiment, where the endogenous ASH1L gene deletion in PC-3M cells is reversed by expression of the full-length protein (or fragment F3) from an integrated vector construct. The gain-of-function effects observed in LNCaP cells provides additional circumstantial evidence but is not a substitute for rescue experiments.

Response: To address this point, we introduced the ASH1L-F3 fragment (full-length protein is not available) into ASH1L-knockout PC-3M cells and determined its rescuing effects. The results showed that ASH1L-F3 fully rescued H3K4me3 and H3K36me3 (**Supplementary Fig. 3c; Line 149**). Cell migration and 2D/3D invasion assays revealed that the re-expression of ASH1L-F3 partially rescued the migration and invasion capacity of metastatic cancer cells (**Supplementary Fig. 3d-e; Line 150-152**). Besides, the re-introduction of ASH1L-F3 in ASH1L-depleted PC-3M cells rescued the expression of ASH1L target genes involved in the metastatic progression (**Supplementary Fig. 5b; Line 249**). Together with loss-of-function studies in diverse human and mouse prostate cancer cells (two sgRNAs in PC-3M, siRNA in DU145, and two sgRNAs + one shRNA in murine DX1 cells) as well as the gain-of-function studies of ASH1L-F3 in LNCaP cells, the new results of true "rescue" experiment further strengthens our conclusion that ASH1L is an epigenetic driver in cancer metastasis.

- For the same reasons as above, across the manuscript, experiments should be carried out with a catalytically inactive mutant of ASH1L (or of fragment F3) to confirm the hypothesis that the effects of ASH1L occur through protein methylation. This concern also applies to the DX1 cells used in the syngeneic model of bone metastases.

Response: Prior studies reported that F2260A is a catalytically inactive mutant of ASH1L (PMID: 37393406; 21239497). As suggested, we introduced the wild-type and F2260A mutant of ASH1L-F3 fragment into LNCaP cells (**Supplementary Fig. 3m; Line 165**) and compared their effects on histone marks, ASH1L target genes, and cell migration/invasion capacities. We verified that the F2260A mutant failed to catalyze methylations of histone H3 at K4 and K36 in prostate cancer cells (**Supplementary Fig. 3n; Line 167**). Consistent with our old data, overexpression of wildtype ASH1L-F3 promoted cancer cell migration and invasion and induced pro-metastatic gene expression, whereas F2260A mutation abolished these effects (**Supplementary Fig. 3o and Supplementary Fig. 5d; Line 169 and 256**), suggesting methyltransferase activity of ASH1L is

required for its pro-metastatic role. Although we tried many times, unfortunately, we couldn't introduce the F3 or F2260A mutant into the DX1 cells for unclear reasons.

Further comments:

3. How do the profiles of CUT&RUNseq compare with those of canonical histone- ChIPseq? For the interpretation of findings, it would be useful to confirm that the two methods are equivalent. This would be important particularly for the H3K36me2/3 marks that are more broadly distributed across gene bodies. In this context, I would urge to be more cautious as to the causative link between histone methylation changes and transcription. Perhaps, histone methylation effects are a consequence rather than the cause of active transcription.

Response: CUT&RUN sequencing (cleavage under targets and release using nuclease) is an emerging technology to analyze gene regulation and transcription factor-DNA binding, and is an alternative to the current standard of ChIP-seq. Some limitations of ChIP-Seq include the cross-linking step promoting epitope masking and false-positive binding sites, suboptimal signal-to-noise ratios, and poor resolution [PMID 25223782, 26685864]. CUT&RUN-seq has the advantage of being a simpler technique with lower costs due to the high signal-to-noise ratio, requiring less depth in sequencing [PMID: 28079019]. Thus, CUT&RUN-seq has been recently used in epigenetic studies: histone marks H3K4me3 [PMID: 37286593; 32066947; 34647658; 32884577; 32352380] and H3K36me2/3 [PMID: 38076924; 34480866; 36918927; 33990599; 37402365; 34555356]. The data has been published in numerous high-profile peer-reviewed journals. ASH1L plays an important role in catalyzing histone methylations, and prior studies found that ASH1L regulates gene transcription by modifying H3K4me3 or H3K36me2/3 in leukemia [References 16-20]. Here, we used the CUT&RUN-seq to determine the effects of ASH1L depletion on H3K4me3 and H3K36me3 marks in metastatic cancer cells. Combined with transcriptional profiling using RNA-seq, CUT&RUN-seq analyses helped us identify the potent target genes of ASH1L in the context of metastatic cancer cells.

4. How do we know if ASH1L methylates only histones or whether the pro-metastatic effects are triggered by direct methylation of transcription factors or other proteins?

Response: We don't exclude the possibility; however, no report has shown that ASH1L directly methylates proteins beyond histones. Our studies in this manuscript focus on determining the novel biological functions of ASH1L in metastatic progression and tumor-associated macrophages in metastatic niches. It would be interesting to explore if ASH1L has methyltransferase activity on transcription factors or other proteins in the future.

5. For the dislocation of a self-inhibitory loop, ASH1L requires the regulatory co-factor MRG15. What is the stoichiometry of ASH1L and MRG15 in metastatic cancer compared to the primary cancer counterparts? Is the ASH1L overexpression accompanied by an MRG15 overexpression?

Response: To address these comments, we analyzed the mRNA levels of ASH1L and MRG15 (encoded by the gene MORF4L1) in TCGA and SU2C datasets but didn't find any correlations between their expression in primary or metastatic prostate cancers (see below). Besides, we didn't see the MORF4L1 gene amplified in these datasets.

6. Figure 3G shows that HIF-1alpha and ASH1L co-exist in the same protein complex but does not prove that ASH1L “interacts” with HIF-1alpha. Biochemical experiments with recombinant proteins or protein fragments are necessary to demonstrate a direct interaction. Accordingly, the model of Figure 8 is a working model (or hypothesis), not a “summary”.

Response: To address these comments, we overexpressed and purified Flag-tagged HIF1a protein and HA-tagged ASH1L-F3 fragment, and then performed an in vitro pull-down assay. The results indicated that ASH1L directly interacts with HIF-1a protein (**Supplementary Fig. 5e; Line 259**). Besides, we have changed the title of Fig 8 to “Schematic working model” as suggested.

7. The terms co-opt and cooperate are very vague. What does it mean that ASH1L and HIF-1alpha "cooperate"? Do ASH1L and HIF-1alpha co-localize across chromatin in ChIP-seq experiments? Does ASH1L locate preferentially in the promoter of HIF-dependent genes? Is there a methylation of HIF-1alpha by ASH1L?

Response: As noted above, we found the direct interaction between ASH1L and HIF-1a using co-IP and in vitro pulldown assay. Through integrating the RNA-sequencing data and the H3K4me3/H3K36me3 CUT&RUN-sequencing data in control and ASH1L-depleted PC-3M cells, we identified 1180 ASH1L direct target genes in metastatic prostate cancer cells (**Fig. 2f and Supplementary Table 4; Line 209-213**). Previously, we tested several ASH1L antibodies for ChIP or CUT&RUN, but unfortunately, none worked well. Thus, we analyzed a published HIF-1α ChIP-sequencing dataset in PC-3 cells (GSE106305) and found that 83.2% (982 in 1180) of ASH1L direct target genes were bound with HIF-1α protein in their promoter regions, including those are associated with cancer invasiveness and metastasis (**Supplementary Fig. 5a and Supplementary Table 5; Line 242-247**). These results indicated that HIF-1α is an important partner of ASH1L in regulating pro-metastasis genes in metastatic cells. However, ASH1L depletion has little impact on histone methylations of the HIF1A gene, suggesting it is not a target of ASH1L.

Furthermore, we downloaded published ASH1L (ENCSR115BBC) and HIF-1a (GSE123461) ChIP-seq datasets in leukemia K562 cells, and found that ASH1L and HIF-1a co-localize on the promoter region of 1132 genes (**Supplementary Fig. 5k,l; Supplementary Table 5**). However, only 32.5% (1132 in 3482) of ASH1L-bound genes were shared with HIF-1α, and these genes are distinct from those in metastatic prostate cancer. It suggests that ASH1L/HIF-1α interaction and co-localization on target gene promoters is a universal mechanism in regulating gene expression, but their target genes may vary in different contexts (**Line 278-283**).

8. Please clarify if IGF-2 is also induced by HIF and if the pro-metastatic macrophages are also enriched in response to HIF activation.

Response: When looking into HIF-1a ChIP-Sequencing dataset in PC-3 cells (GSE106305), we found that HIF-1a is bound to the promoter region of IGF-2 (**Supplementary Fig. 8a; Line 444**). Then, we determined the IGF2 expression in control and ASH1L-F3 LNCaP cells with or without HIF1A knockdown using two individual siRNAs. The result indicated that ASH1L overexpression induces IGF2, but HIF1A knockdown dampened the IGF2 upregulation (**Supplementary Fig. 8b; Line 446**). The results indicate that IGF-2 is induced by HIF.

To determine if HIF is involved in the pro-metastatic macrophage enrichment driven by ASH1L in vivo, we injected control and ASH1L-depleted DX1 cells into B6 male mice and then treated the mice with vehicle or PX-478 thrice weekly (i.p., 40mg/kg) for three weeks. IVIS imaging and ex vivo fluorescence imaging revealed that ASH1L depletion significantly suppressed the tumor outgrowth in bone, but this effect was abolished by PX-478 treatment (**Fig. 7h and Supplementary Fig. 9g**). Besides, HIF1A inhibition showed anti-tumor effects in the control group but not in the ASH1L depletion group (**Fig. 7h and Supplementary Fig. 9g**). Furthermore, we performed multiplex IHC staining in the above bone tumors and found that pro-metastatic macrophages were significantly reduced in ASH1L-expressing tumors upon HIF1A inhibitor treatment (**Fig. 7i and Supplementary Fig. 9h**). These new results demonstrate that HIF1A plays a vital role in both tumor outgrowth in bone and pro-metastatic macrophage enrichment driven by ASH1L (**Line 494-507**).

Fig. 7.

h, 5×10^5 control and ASH1L-depleted DX1 cells were injected into the tibia of C57BL/6J male mice, followed by treatment of vehicle or HIF-1 α inhibitor PX-478 three times per week (i.p., 40mg/kg) for 3 weeks. Bioluminescence images and BLI intensity quantification of bone tumors over time are shown. **i**, Quantification of pro-tumoral TAMs (F4-80+CD206+) and pro-inflammatory TAMs (F4-80+CD206-) in bone tumors, determined by multiplex IHC staining. Data represent the mean \pm standard deviation of > 10 individual views from 4 mice per group.

9. The quantification of metastatic tumors in Figure 7G shows only 5 control (vehicle) animals out of a total of 10 animals. What happened with the analysis of the missing 5 animals? Why is their tumor cell load not shown?

Response: We thank this reviewer for pointing this out. In the revised manuscript, we included IVIS imaging quantification data of 7 mice (revised **Fig. 7e**) and survival data of 9 mice (revised **Fig. 7f**) in the vehicle control group. Two mice have no IVIS imaging data on Day 34, because they were found dead earlier. Raw data was included in the resubmission package. In the revised manuscript, we excluded one mouse in the survival data of the vehicle control group due to unstable IVIS signals (we observed a strong IVIS signal from 14 to 28 days, but the signal was suddenly lost on Day 34 and after).

Minor comments:

10. In the Western blot of Figure 1D, there is some residual ASH1L in the knockout cells. Please explain the reason. Are these cells able to reamplify the gene during repeated passages?

Response: The ASH1L-depleted PC-3M in Fig 1D were generated using the lentiviral-mediated CRISPR-sgRNA system. Considering the heterogeneity of PC-3M cells, we didn't pick the single-cell clone to generate complete knockout sublines. Instead, we used the pooled ASH1L-depleted cells for functional assays to better reflect the feature of PC-3M in vitro and in vivo. Thus, there is some residual ASH1L in the knockout cells. Because ASH1L knockout has little impact on cell fitness or cell proliferation (**Supplementary Fig. 2a-b**), the depletion of ASH1L was well maintained over passages.

11. Please describe the genotype of the "nude" mouse used in the xenograft metastasis model.

Response: All nude mice used in this study are the strain of CrTac:NCr-Foxn1nu (Sp/Sp) purchased from Taconic Biosciences (Cat # NCRNU-M). This information has been added to Method (**Line 701**).

12. Line 129: should inhibition be replaced by deletion?

Response: The "inhibition" in the original Line 129 has been replaced by "deletion" (**Line 130**).

Reviewer #4 (Remarks to the Author): expert in prostate cancer metastasis, bone metastasis and scRNA-seq

Bone metastasis is associated with high rates of mortality in cancers including prostate, breast, lung, kidney, ovarian, and melanoma, with 80% of prostate cancer patients developing painful osteoblastic bone lesions. Despite a known association between alterations in histone methyltransferases and metastatic disease across cancer types, epigenetic determinants of bone metastases, and the mechanisms governing this, are not well understood. Meng and colleagues identify prostate cancer cell expressed ASH1L as an epigenetic driver of tumor cell invasion and outgrowth. Furthermore, they demonstrate that ASH1L leads to upregulation of HIF1 target genes, including IGF2, and that deletion of ASH1L in tumor cells is associated with a reduction in tumor-associated macrophages (TAMs) within the tumor microenvironment. They go on to conclude that secreted IGF2 signals to macrophages, promoting a pro-tumoral lipid-associated TAM phenotype through OXPPOS. Using small molecule inhibitors, they demonstrate that ASH1L can be targeted to reduce prostate tumor burden in the bone and improve overall outcome in a pre-clinical model. While the authors combine several omics strategies with mouse models and in vitro studies, there are some issues that need to be addressed prior to publication:

Response: We are grateful for this reviewer's positive feedback and for highlighting our study's novelty, significance, and translational potential. We also appreciate his/her constructive comments, which helped us improve the manuscript. Please find our point-by-point responses below:

1. Throughout the text, the authors ascribe their findings that ASH1L is an epigenetic regulator of prostate tumorigenesis to generic "metastatic malignancies" and "cancers". It is recommended that specific text is used instead to refer to their data-supported conclusions within prostate cancer

Response: As suggested, we revised the "Result" section and made major conclusions within prostate cancer.

2. Figure 1H requires quantification and statistics to support claims that "ASH1L- depleted PC-3M cells had impaired capacities to colonize the bone and form osteolytic lesions"

Response: We analyzed the bone metastasis rates in control versus ASH1L-knockout groups and found that control PC-3M cells metastasized to the bones in 67% (8 in 12) of mice; in contrast, no skeletal metastasis was detected in ASH1L knockout groups (Line 132-134). Also, we quantified the osteoclast and osteoblast in metastatic tumors (Fig. 1h and Supplementary Fig. 2j). The results indicated that ASH1L depletion impaired the capacity of PC-3M cells to colonize the bone and significantly suppressed the osteoclast abundance and osteolytic lesion formation; however, it had little impact on osteoblast (Line 135-139).

3. Figures 4C, and 4D require quantification and statistics to support claims that intratibial injected DX1 cells “produce osteolytic/osteoblastic features”

Response: In collaboration with the Bone Histomorphometry Core Laboratory, we performed TRAP and Toluidine blue staining on DX1 bone outgrown tumors in **Figures 4** and **Supplementary Fig. 6** and assessed bone volume, osteoblast, and osteoclast number. As shown in the new figures (**Fig. 4c-g** and **Supplementary Fig. 6a, i-k**; **Line 294-299**), histopathological analysis revealed significant absorption of cortical bone and remarkable new woven bone production in the outgrown tumors, along with enriched osteoclasts and osteoblasts, confirming that DX1 tumors in bone displayed mixed osteolytic/osteoblastic features. Besides, ASH1L depletion in invading cancer cells protected cortical bone from absorption and reduced new woven bone production (**Fig. 4c-e** and **Supplementary Fig. 6d**). Both osteoclasts and osteoblasts were decreased in bone-outgrown tumors upon ASH1L deletion (**Fig. 4c, f, g**). Similar effects were observed when knocking down ASH1L using shRNA (**Supplementary Fig. 6h-k**) (**Line 303-310**). These new results of quantification and statistics further strengthened our conclusions. Please find more technical details about the software and data analyses in our response to Comment #3 from reviewers #1 and #3.

4. The representative images in Figure S9E do not reflect the data in the matched main figure 7I

Response: Infiltration of TAMs in metastatic tumors after treatment was determined by immunofluorescent staining in **Supplementary Fig. 9f** (original S9E). We randomly selected 7-8 individual views on three samples per group with a magnification of 20X. Due to the space limit, the representative images in **Fig. 7g** (original 7I) were cropped (only half of the images were presented) and zoomed in to show the specificity and details of TAM staining. To avoid confusion, we changed the figure legends to “Representative images (enlarged)” in the figure legends.

5. It is recommended to include percentages in Fig 1A either in the panel or within the legend

Response: The percentages of each genetic alteration type were added in the revised **Fig. 1a**.

6. Please include brightfield images for Figs S1G, S2I, S6B, S6E

Response: The brightfield images have been added in **Supplementary Fig. 1g, 2i, 6c** (original S6B), and **6g**(original S6E)

7. Please include information on how the in vitro proliferation assays were performed in the Methods section

Response: A detailed “Cell proliferation assays” method was added to the Methods section (Line 792-795).

8. Intratibial injections do not model the metastatic process of invasion, intravasation, or extravasation but instead model outgrowth in the metastatic target tissue. It is recommended that careful, clear language be used in describing this model, reaching conclusions from the data generated using this model, and in the design of Figure 8. It should also be noted that intratibial injections cause massive inflammation which activates and enhances macrophage populations, a consideration for data interpretation

Response: We thank this reviewer for pointing this out! In the revised manuscript, we changed the description from “bone metastasis” to “tumor outgrowth in bone” in the main text and figure legends related to intratibial injection-derived models. We further clarified the language in describing this new syngeneic model and performed essential histopathological analysis to assess bone volume, osteoblast, and osteoclast number, as suggested (see more details in our response to Comment #3). Also, the inflammation and macrophage enrichment caused by intratibial injections were discussed as a limitation of this model (Line 572-579). The schematic working model in Fig 8 summarizes the conclusions drawn from our in-depth mechanistic studies and diverse mouse models using intracardiac and intratibial injections, which mimic different steps during the metastatic process.

9. If IVIS images were available during tumor growth time course corresponding to Figures 1J, 4B, 7B, a tumor growth curve would be a nice addition

Response: As suggested, we quantified IVIS BLI intensity at different time points and generated the tumor growth curves for Figs 1j, 4b, and 7b.

10. For figure 3d, why not look at cut/run seq for these genes as done in figure 2h

Response: There are 110 HIF-1 α target genes (listed in **Supplementary Table 10**), thus it is impossible to present the CUT&RUN tracks of individual genes in **Fig 3d**. Instead, we used scatter plots to demonstrate the changes in gene expression (upper plots), H3K4me3 (left bottom), and H3K36me3 (right bottom) marks of HIF-1 α target genes upon ASH1L depletion. Each red dot indicates a gene with downregulated signals (FDR \leq 0.05).

11. Figure 3d, only one panel is sg#3 the rest are sg#2 – why? In addition, please report R2 values

Response: We used two individual sgRNAs, #2 and #3, targeting ASH1L when performing the RNA-sequencing in **Fig. 2a-c** and **Supplementary Fig. 4a, b**. The results from these sgRNAs are largely overlapped, thus, we only used sg#2 for H3K4me3 and H3K36me3 CUT&RUN-Sequencing analyses. As noted above, in **Fig 3d**, we used scatter plots to present the changes in gene expression (two upper plots for two individual sg#2 and #3), H3K4me3 (left bottom for sg#2), and H3K36me3 (right bottom for sg#2) marks of HIF-1 α target genes upon ASH1L depletion. Each red dot indicates a gene with downregulated signals (FDR \leq 0.05). Two dotted lines represent FC = 1.5, not the linear regression. In this figure, p-values were calculated using paired t-tests to assess the significance of changes in expression or histone modifications in HIF-1 α target genes following ASH1L depletion. The primary purpose of these scatter plots is not to demonstrate a linear relationship between expression or histone modifications in ASH1L-depleted versus control samples, but rather to illustrate the magnitude of changes. Therefore, we think reporting R² values is not appropriate and may mislead readers regarding the intended focus of these analyses.

Fig. 3d, Scatter plots demonstrating the changes in gene expression, H3K4me3, and H3K36me3 marks of HIF-1 α target genes upon ASH1L depletion. Genes with downregulated signals (FDR \leq 0.05) are highlighted in red. Two dotted lines represent FC = 1.5.

12. Figure S5 – the y axis is cropped

Response: The y-axis was fully presented in the revised **Supplementary Fig. 5h** (original S5A).

13. When discussing the validation efforts in figure 5H with CD206 staining – it was confusing why this marker was chosen. It would be helpful to the reader if introduced on line 317 that the pro-tumoral gene Mrc1 identified in cluster MC5 is transcribed into the protein called CD206

Response: In the revised manuscript, we clarified the CD206 marker is encoded by the gene Mrc1 to avoid confusion. (**Line 388**)

14. Representative images in Figure S7E do not reflect the changes shown in the corresponding bar graphs

Response: Infiltration of TAMs in control and ASH1L-F3 bone tumors was determined by multiplex IHC staining in **Supplementary Fig. 7f** (original S7E). We randomly selected 7-8 individual views on three samples per group with a magnification of 20X. Due to the space limit, the representative images were cropped (only half of the images were presented) and zoomed in to show the specificity and details of TAM staining. To avoid confusion, we changed the figure legends to “Representative images (enlarged)” in the figure legends.

15. Please use clear language when describing how the loss of ASH1L in tumor cells effects the macrophage populations. An example is line 370 when the authors write “..., indicating ASH1L promoting the proliferation and survival of TAMs”. This becomes confusing when conceptualizing the mechanism

Response: Thanks for pointing this out! We have rewritten this sentence (original Line 370) to clarify the phenotype and functional changes of TAMs are attributed to ASH1L in cancer cells. We also revised other sentences to avoid confusion.

[Line 414] *“It indicates that ASH1L in invading cancer cells contributes to the proliferation, survival, and anti-inflammatory phenotype of TAMs”.*

16. On line 387 the authors refer to Figure 6A to conclude that “...TAMs from ASH1L- depleted bone tumors exhibited decreased IGF uptake...”, however Figure6A shows pathway analysis and therefore can only support an inference via IGF downstream signaling

Response: As suggested, we removed the “IGF update” from the original Line 387.

[Line 431-434] “Pathway analysis in Fig. 6a also indicates that TAMs from ASH1L-depleted bone tumors exhibited decreased IGF downstream signaling, i.e. AKT and MAPK pathways, suggesting IGF signaling may be involved in the TAM plasticity and metabolic programming driven by ASH1L.”

17. Throughout the text, the authors refer to their fragment 3 expression of ASH1L (see Fig S3A) as “ASH1L overexpression”. The full length protein is not being expressed in this model, however. Please clarify in text

Response: Like other high-molecular weight histone methyltransferases, ectopic expression of full-length ASH1L protein in mammalian cells is challenging (PMID: 27614073; 28598443). In this study, we constructed three ASH1L fragments that respectively encode 1–882 amino acids (F1), 883–1,890 amino acids (F2), and 1,891–2,969 amino acids (F3) of human ASH1L protein (**Supplementary Fig. 3a; Line 142-147**). Among them, fragment F3 contains all functional domains of the ASH1L protein (**Supplementary Fig. 3a**) and presents histone methyltransferase activity at H3K36 and H3K4 (**Fig. 1i and Supplementary Fig. 3c; Line 148**). To verify the function of the F3 fragment, we re-expressed ASH1L-F3 in ASH1L knockout PC-3M cells and found that ASH1L-F3 can reverse the effects of ASH1L knockout on target gene expression (**Supplementary Fig. 5b; Line 248-250**), cell migration, and 2D/3D invasion (**Supplementary Fig. 3d,e; Line 150-153**). Together with the effects of ASH1L-F3 on promoting cancer cell invasiveness and tumor outgrowth in bone (**Fig. 1j-k and Supplementary Fig. 3f-l**), the new data strengthen our conclusion that ASH1L overexpression promotes prostate cancer cell invasion and tumor outgrowth in bone. To increase the accuracy, we replaced “ASH1L overexpression” with “ASH1L-F3 overexpression” when describing the results throughout the text. Given that ASH1L-F3 mimics the function of the full-length protein, we used the term “ASH1L overexpression” when describing conclusions.

Supplementary Fig.3.

c, ASH1L-F3 was reintroduced in ASH1L-depleted PC-3M cells, followed by western blot analyses of HA and H3 marks. **d**, Representative images and quantification of migrated and invaded control, ASH1L-depleted, and ASH1L-F3 reintroduced PC-3M cells. Scale bar = 1000µm. **e**, 3D sphere invasion assays of control, ASH1L-depleted and ASH1L-F3 reintroduced PC-3M cells. Scale bar = 100 µm.

18. On line 410 the authors write “we also found that IGF-2 expression was strongly associated with lipid-associated TAM markers and pro-tumoral phenotype markers” in reference to Figures S8B,C. However, the Pearson’s correlations range between 0.29 and 0.4. Please revise language to reflect the weaker correlation that is shown

Response: As suggested, the adverb “strongly” has been removed.

[Line 460-462] *“In metastatic PCa patients, we also found that IGF-2 expression was associated with lipid-associated TAM markers and pro-tumoral phenotype markers (Supplementary Fig. 8f and g)”*

19. In reference to the data presented in Figure 7K,L, a correlation plot would be a nice addition to evaluate ASH1L expression and lipid-associated TAMs, agnostic of binning

Response: As suggested, we tried to generate a correlation plot in Fig. 7L. As shown below, ASH1L and lipid-associated TAM abundance showed a certain trend of positive association; however, the small sample size could not support statistical correlation analysis. To address the concern of binning, we analyzed the expression level of ASH1L from the scRNA-seq dataset (GSE210358) of epithelial cells from 13 patients with metastatic prostate cancer (Supplementary Fig. 10d; Line 516). Statistical analysis showed significant differences in ASH1L expression in “ASH1L-high” versus “ASH1L-low” groups, suggesting the binning of 13 samples is reasonable for assessing ASH1L’s correlation with lipid-associated TAM abundance in Fig. 7I.

Supplementary Fig. 10d, 13 patients were classified into two groups (ASH1L_Low vs ASH1L_High) by ASH1L expression levels in epithelial cells.

Fig. 7I. Percentage of LA-TAMs in total non-epithelial cells from metastatic tumors with low versus high ASH1L expression.

20. For all stains where images were used for quantification, please include secondary-only controls in the supplement, split by channels

Response: In the revised supplementary figures, we included secondary-only control images split by channels for all multiplex IHC staining (**Supplementary Fig. 7e**) and IF staining (**Supplementary Fig. 9b**).

Reviewer #5 (Remarks to the Author): expert in hypoxia, HIF, epigenetics and tumor microenvironment

ASH1L is a histone methyltransferase. In this manuscript, the authors report that ASH1L drives the metastatic progression of prostate cancers via co-op with hypoxia/HIF1A to induce pro-metastatic transcriptome and to promote monocyte differentiation to pro-tumor and anti-inflammatory macrophages via IGF2. Although it is known that hypoxia upregulate IGFs, which may induce immune repressive macrophages, this study provides more mechanistic details. Overall, this is an excellent study, and will lead to new understanding in how tumor cells, especially prostate cancer, develop immune repressive metastasis via ASH1L and HIF1A. The strength of this manuscript includes 1) the combinational use of single-cell and bulk sequencing data to develop new insights of ASH1L interaction with hypoxia/HIF1A and the consequences on metastatic niche and monocyte/macrophage differentiations, 2) strong gain/loss-function studies of ASH1L in vitro and in metastatic mouse models (intracardiac and intratibial). A major weakness is that, in the 2nd half of the study, the role of HIF1A in ASH1L-driven macrophage plasticity is less established. It is unclear whether the ASH1L/IGF2/pro-tumor and anti-inflammatory macrophage axis requires HIF1A or hypoxic microenvironment.

Response: We thank this reviewer for his/her positive feedback and for highlighting this study's novelty, significance, and strength. We also appreciate the constructive comments on HIF1A, which helped us improve the manuscript significantly. Please find our point-by-point responses below:

Major concerns:

1. Hypoxia is known to upregulate IGFs. However, the role of hypoxia/HIF1A in facilitating the ASH1L/IGF2/TAM axis is unclear. In the first half of the manuscript, the authors provide strong data in establishing that ASH1L promotes metastatic gene expression in the presence of HIF1A. However, it is less clear whether the communication between tumor cells and macrophage cells still requires hypoxia/HIF1A. The role of HIF1A in ASH1L-driven IGF2 mRNA and TAM differentiation is not specifically tested in Figures 4-7.

Response: We thank this reviewer for pointing this out! When looking into HIF-1a ChIP-Sequencing dataset in PC-3 cells (GSE106305), we found that HIF-1a is bound to the promoter region of IGF-2 (**Supplementary Fig. 8a; Line 444**). Then, we determined the IGF2 expression in control and ASH1L-F3 LNCaP cells with or without HIF1A knockdown with two individual siRNAs. The result indicated that ASH1L overexpression induces IGF2, but HIF1A knockdown dampened the IGF2 upregulation (**Supplementary Fig. 8b; Line 446**). The results indicate that HIF1A plays an important role in ASH1L-driven IGF2 expression.

To determine if HIF1A contributes to the tumor outgrowth in bone and TAM plasticity driven by ASH1L, we injected control and ASH1L-depleted DX1 cells into B6 male mice and then treated the bone tumor-bearing mice with vehicle or HIF1A inhibitor PX-478 (i.p.,40mg/kg) three times per week for 25 days. IVIS imaging and ex vivo fluorescence imaging revealed that ASH1L depletion significantly suppressed the tumor outgrowth in bone, but this effect was abolished by PX-478 treatment (**Fig. 7h and Supplementary Fig. 9g**). Besides, HIF1A inhibition showed anti-tumor effects in the control group but not in the ASH1L depletion group (**Fig. 7h and Supplementary Fig. 9g**). Furthermore, we performed multiplex IHC staining in these tumors using monocyte and TAM markers. The results indicated that HIF1A inhibition, phenocopying ASH1L depletion, significantly reduced the abundance of pro-tumoral/pro-metastatic TAMs

(F4/80+ CD206+) in bone tumors, whereas increased the pro-inflammatory TAMs (F4/80+ CD206-) and monocytes (Ly6c+; F4/80-) (Fig. 7i and Supplementary Fig. 9h). Besides, we found that the effects of ASH1L depletion on pro-tumoral/pro-metastatic macrophages were completely abolished by HIF1A inhibition (Fig. 7i and Supplementary Fig. 9h). These new results establish the vital role of HIF1A in ASH1L-driven tumor outgrowth in bone and TAM plasticity in metastatic bone niches (Line 494-507).

2. The involvement of HIF1A makes sense in PC3M cells, since HIF1A protein is present in non-hypoxic condition in this cell line. However, it may also cast uncertainty of the involvement of this gene/protein in other prostate cancer cells or cell lines in non-hypoxic conditions. Most of the prostate cancer cell lines e.g., LNCaP, have very low or none-detectable levels of HIF1A proteins in non-hypoxic conditions. In vivo, at least in primary prostate cancers, the impact of hypoxia was reported as low based on hypoxia-signature genes via bulk RNA-Seq (Bhandari et al, Nat Genet, 2019). Thus, it will be highly informative to use the single-cell sequencing data from the 13 patients and the mouse models to evaluate the involvements (requirements) of hypoxia and/or HIF1A in subpopulations of the tumor and macrophage cells in Figures 4, 5, and 7.

Response: To address this comment, we first performed IHC staining in control and ASH1L-F3 LNCaP cell-derived bone tumors and found high expression of HIF1a (**Supplementary Fig. 5c**). We agree with this reviewer that HIF1A expression is low in LNCaP cells, but the hypoxia condition in the bone tumors may induce the stabilization of HIF1a protein, which co-operate with ASH1L in regulating pro-metastatic genes and TAM-modulating genes. Then, we analyzed the published bulk RNA-seq and single-cell transcriptomic datasets from human primary and metastatic prostate tumor samples. The results showed that, among HIF family members, HIF-1 α most significantly co-expressed with ASH1L in epithelial components (**Supplementary Fig. 5f-h**). Compared to the ASH1L low-expressing group, PCa tumors expressing high ASH1L levels exhibited elevated HIF-1 α transcriptome in epithelial components (**Fig. 3h and Supplementary Fig. 5i**) (**Line 262-267**).

As this reviewer suggested, we analyzed the HIF1a signature genes in published scRNA-seq datasets of 13 metastatic prostate cancer samples and our murine bone metastasis models. Consistent with our findings in primary tumors above, HIF1a and its transcriptome were found in subpopulations of human metastatic prostate tumors, and their expression was higher in tumors with high levels of ASH1L (**Supplementary Fig. 10d,e; Line 516-520**). In murine bone outgrown tumors, we also found HIF1A and its target genes expressed in a subpopulation of prostate cancer cells, and their expression was down-regulated upon ASH1L knockout (**Supplementary Fig. 6o; Line 327-331**). Compared to invading cancer cells, monocytes and TAMs in the metastatic bone niche expressed lower levels of HIF1A target genes, regardless of ASH1L status (**Right bottom, Below**). Together, these new results provide additional evidence supporting the involvement of HIF1A in subpopulations of primary and metastatic prostate tumors and its positive association with ASH1L expression.

Supplementary Fig. 10.

Single-cell transcriptomic analysis of human PCa metastatic tumors (GSE210358; n = 13 patients). **d**, 13 patients were classified into two groups (ASH1L_Low vs ASH1L_High) by ASH1L expression levels in epithelial cells. **e**, tSNE views of epithelial cells color-coded by the count of HIF-1 α target genes.

Minor concerns:

1. Figure 3 shows the interaction between ASH and HIF1 to regulate gene expression. LNCaP is used for Fig 3F, however, HIF1A protein is expressed at very low level in non-hypoxic condition in this cell line, thus the question is whether ASH1L overexpression bypasses the requirement of HIF1A to upregulate these genes (unlikely based on results in Fig 3I). The level of HIF1A protein in 3I is needed.

Response: To address this comment, we used CoCl₂ treatment to mimic the hypoxia condition and re-did the experiments in the original Fig 3F. As this reviewer predicted, the upregulation of ASH1L target genes was even more significant than the old data, indicating hypoxia/HIF1A protein stabilization is important in mediating the effects of ASH1L overexpression (**Fig. 3f; Line 156**). Moreover, HIF1A knockdown using siRNAs or HIF1A inhibitor treatment can impair the induction of pro-metastatic genes driven by ASH1L overexpression (**Fig. 3i, Line 268-274; Supplementary Fig. 8b, Line 446**). Along the same line, we re-did the cell migration assays of control and ASH1L-F3 overexpressing LNCaP cells in the presence of CoCl₂ and found that ASH1L-F3 promotes the migration capacity of LNCaP cells, whereas HIF1A inhibitors treatment completely abolished this phenotype (**Fig. 3j; Line 273**). Together, these new results suggest that HIF1A is required in pro-metastatic gene induction and invasiveness driven by ASH1L overexpression.

2. Most metastatic prostate cancer cells express androgen receptor (AR). However, PC3 cells, which do not express AR, are used in many cell/xenograft models. In TCGA and SU2C analysis (Fig 1A), what are the association between ASH1L and AR and AR activity? Further, AR-expressing LNCaP has a metastatic origin, however, the ASH1L mRNA level is low, it is unclear how this gene (expression) is regulated.

Response: We agree that most metastatic prostate cancer patients express androgen receptors. Unfortunately, there are a limited number of metastatic prostate cancer cell lines available for preclinical studies, and most of them are AR-negative. As suggested, we assessed the correlations between ASH1L and AR expression in primary (TCGA dataset) and metastatic (SU2C dataset) prostate cancer patients. To our surprise, we found a strong correlation between ASH1L and AR expression in primary prostate tumors but a much weaker correlation in metastatic prostate cancer patients (**Fig. A,B below**). Next, we treated AR-positive LNCaP cells with androgen (Dihydrotestosterone, DHT) or AR inhibitor (Enzalutamide, Enza), but didn't observe a meaningful effect of AR activation or inhibition on ASH1L expression (**Fig. C,D below**). AR target gene KLK3/PSA was used as a positive control. These results indicated that AR signaling doesn't directly regulate ASH1L. Hence, our loss-of-function studies in AR-negative PC-3M cells and gain-of-function studies in LNCaP cells are sufficient to support the major conclusions in this manuscript.

A-B. Scatter plot illustrating the correlation between ASH1L and AR expression in primary prostate cancer (**A, TCGA dataset**) and metastatic prostate cancer patients (**B, SU2C dataset**). The Pearson correlation coefficient (r) and P value are reported. **C.** Expression of ASH1L and AR target gene KLK3 (PSA) in LNCaP cells treated with different concentrations of Dihydrotestosterone (DHT), determined by qPCR. **D.** Expression of ASH1L and AR target gene KLK3 (PSA) in LNCaP cells treated with different concentrations of AR inhibitor Enzalutamide (Enza), determined by qPCR.

3. Some of the single-cell analyses require quantification, e.g., Figures 3H and 6B. Also, the exact identities of genes used for these analyses are required.

Response: The UMAP plots of scRNA-seq analysis in Figs 3H and 6B were color-coded by the count of HIF1A target genes and OXPHOS pathway genes. Technically, this type of data can not be quantified. As suggested, we included the lists of HIF1A target genes ($n=110$) and OXPHOS pathway genes ($n=43$) in the revised **Supplementary Table 10**.

4. IGF2 was identified via bulk sequencing data (Fig 6C). What are the levels of IGF2 in single-cell sequencing of the human and mouse tumors?

Response: The IGF2 expression can be detected using qPCR assays and bulk RNA-seq, both our PC-3M datasets and published datasets of primary (TCGA) and metastatic (SU2C) prostate cancers. However, for unclear reasons, IGF2 mRNA was undetectable in scRNA-seq datasets, either human prostate cancer scRNA-seq datasets or our murine scRNA-seq datasets.

5. For the intratibial models in figure 1J and 4B, it would be interesting to see the bone tumor signals over several time points.

Response: As suggested, we quantified IVIS BLI intensity at different time points and generated the tumor growth curves for **Figs 1j, 4b, and 7b**.

6. The exact molecular mechanism, by which IGF2 promote immune repressive phenotype via OXPHOS remains unclear.

Response: To address this comment, we treated THP1-derived macrophages with different concentrations of IGF2 recombinant proteins and then determined TAM phenotype markers and OXPHOS pathway genes using qPCR. As shown in **Supplementary Fig. 8d,e**, IGF2 promoted macrophage differentiation into lipid-associated TAMs and upregulated OXPHOS pathway genes, but had little impact on markers of inflammatory TAM (**Line 455-457**). Pathway analysis of our murine scRNA-seq datasets revealed that ASH1L depletion in invading cancer cells down-regulates the AKT and MAPK signaling pathways in monocytes and TAMs (**Fig. 6a; Line 410**), consistent with prior studies of IGF2 and its receptors (PMID: 17360667; 21595894). To verify the downstream signaling of IGF2 in TAMs, we assessed the phosphorylation levels of key factors in both signaling pathways using Western blot. The results showed that IGF2 activated the AKT-GSK3b-mTOR signaling, but not the MAPK pathway, in TAMs (**Supplementary Fig. 8e; Line 455-457**). These new results suggest that IGF2 promotes immunosuppressive TAMs and elevated OXPHOS via activating AKT-GSK3b-mTOR signaling.

REVIEWER COMMENTS

Reviewer #1 (Remarks to the Author):

In this revised version of the manuscript, the authors have answered each of our comments. The additional data provided by the authors has addressed the major concerns and has substantially improved the manuscript. However, further clarification is needed regarding the methods used for bone histomorphometric analysis and data presentation:

Response: We appreciate this reviewer's positive feedback and his/her efforts to improve this study. Please find our point-by-point responses below:

1. The Y axes of Suppl. Fig. 5i and Suppl. Fig. 10d should be labelled clearly (ASH1L expression?). We strongly suggest that both graphs should be plotted, and statistics performed, using normalised count/expression level of ASH1L on a per patient basis, instead of per cell basis. The latter can underestimate variability and overstate statistical significance.

Response: As suggested, we regenerated the graphs using the normalized ASH1L mean expression in all epithelial cells from each patient sample and presented the statistical significance of ASH1L expression between ASH1L-high vs ASH1L-low groups. The Y axes of Fig. S5i and Fig. S10d were revised into "ASH1L mean expression (normalized)".

2. The addition of quantitative data is an improvement in confirming the effects of tumor on bone parameters in a ASH1L-dependent manner. However, the methods used for the quantitative analysis are not entirely clear. In Fig. 4 and Suppl.Fig. 6, the regions of interest (ROI) for cortical and woven bone BV/TV analysis are not defined. What were the criteria for selecting these ROIs? Also, the meaning of "# osteoclasts/osteoblasts per view" is unclear. How many "views" were analyzed, and what is the ROI? It may be more appropriate to quantify osteoclasts/osteoblasts per unit of bone surface (number/mm).

Response: We thank this reviewer for pointing this out. We collaborated with the Bone Histomorphometry Core Laboratory at MD Anderson Cancer Center when performing bone parameter quantification. Typically, BioQuant Osteo software is used for analyzing BV/TV and the numbers of osteoblasts (N.Ob/BPm, mm^{-1}) and osteoclasts (N.Oc/BPm, mm^{-1}) within selected ROIs that begin 150 μm away from both the cortical bone and the growth plate. However, the intratibial injection and the aggressive nature of this model caused damage to the growth plate, and cancer cells fully occupied the bone marrow cavities, limiting our ability to obtain bone image data using BioQuant. Besides, we observed a significant formation of new woven bone adjacent to the outside of cortical bone in response to tumor cell invasion. These areas provide a good alternative for bone marrow cavities for evaluating osteoblast and osteoclast activity. Thus, we employed Qupath, an open-source AI-based image analysis software that has been validated in many studies for both brightfield and fluorescent imaging. It allowed us to classify and quantify bone staining images.

To assess cortical bone volume fraction, we used Qupath software version 0.5.1-x64 on H&E-stained images to define an ROI encompassing the entire tibia, excluding regions of new woven bone formation. The measured ROI area was considered the tissue volume. We then trained the software to specifically classify the cortical bone area as cortical bone volume. For new woven bone volume fraction analysis, we used Qupath software on H&E-stained images to define a tissue area including the new woven bone and tumor region, and trained the software to specifically classify the new woven bone area as woven bone volume. For clarity and consistency in our analytic approach, we report the results as bone volume/tissue volume (BV/TV), and we batch processed all images using a saved script to ensure uniformity. For TRAP staining analysis, the osteoclast number per area (mm^2) in the new woven bone area was assessed as multinucleated TRAP+ as well as the adjacent bone surface cells. Toluidine blue staining was administered to quantify the number of cuboid osteoblasts per area (mm^2) in the new woven bone area. This information was included in the revised Method section- “Histomorphometric analysis of the bone samples” [Line 727-743].

In the revised manuscript, Fig S6a was added to visualize representative regions used for cortical BV/TV and Woven BV/TV analyses. Also, we revised the legends of **Fig 4d-e** and **S6i** to clarify the definition of the cortical BV/TV and Woven BV/TV (see below). As suggested, we reanalyzed the TRAP and Toluidine blue staining data and quantified the “# of osteoclasts/osteoblasts per mm^2 ” in Fig **4f-g** and **S6j-k** (see below).

3. The histological images in Fig. 4C are shown at different magnifications. It would be helpful to define which areas were magnified, to guide readers in focusing on the relevant areas.

Response: Due to the space limit in the main figures, the magnification areas were disclosed in the revised source data files.

4. In Figs. 5h and 7e the authors clarified that data points and statistical analyses were based on technical replicates that were pooled from all 3 mice (therefore >10 data points instead of just 3). It is debateable whether the use of pooled data is appropriate. While the authors stated that this method of analysis was included in both the methods section and figure legend, it was not found in the latter. Please include this information in the figure legend to make it clear that the data shown are technical replicates pooled from 3 mice.

Response: In Figs. 5h and 7e, we randomly captured the images from 3 tumor samples (3 mice), and 3-4 views were captured in each sample. Due to the variations among samples as well as the spatial distribution of TAMs, we eventually pooled all individual views for statistical analysis. In the revised figure legends and Method section, we clarified that “Data represent the mean \pm standard deviation of >10 individual views pooled from three biological replicates (3 mice; 3-4 views per mouse).”

Reviewer #2 (Remarks to the Author):

The reviewer acknowledges that major issues are now eliminated by additional experimental data that strenghten the manuscript.

Response: We thank this reviewer for his/her positive feedback and efforts in improving this study!

Reviewer #3 (Remarks to the Author):

Response: We appreciate this reviewer’s positive feedback and his/her efforts to improve this study. Please find our point-by-point responses above.

Reviewer #4 (Remarks to the Author):

The revised manuscript contains new rescue experiments and loss/gain of function experiments, which strengthen the conclusions that ASH1L directly contributes to tumor cell invasion and migration. Furthermore, the addition of experiments using the HIF1a inhibitor PX-478 demonstrate the effects of ASH1L-driven tumor outgrowth and pro-tumoral TAM accumulation are dependent on HIF1a activity. This supports the mechanism described in the second half of the manuscript. There are a few minor points to address however.

Response: We thank this reviewer for his/her positive feedback and efforts in improving our study! Please find our point-by-point responses below:

1. Line 157 – the authors write “Functionally, overexpression of ASH1L-F3, not F1 or F2, can significantly augment cell migration and invasion (Supplementary Fig. 3f-h)”. However, Supplementary Figure 3 does not show invasion data from cells expressing F1 or F2 (as for migration in supp fig 3g). It is recommended to change the language in text or include the data indicating no changes to cell invasion in cells with F1/F2.

Response: This sentence has been revised to “Although overexpression of ASH1L-F3 did not affect cell growth, it significantly augmented cell migration and invasion, while F1 or F2 had little effect on cell migration (**Supplementary Fig. 3f-i**).” [Line 156-158]

2. The changes on the western blot in supplementary figure 8e are extremely slight. Please provide appropriate densitometric & statistical analysis supporting the claim in lines 455-456 that AKT-GSK3b-mTOR signaling is activated.

Response: As suggested, we analyzed the signal intensity ratio of phosphorylated protein versus the total protein of GSK3b, mTOR, AKT, and ERK. The ratios were normalized to the control (IGF2=0 ng/ml) sample for each marker.

3. We thank the authors for including the consideration that intratibial injection-induced inflammation may influence macrophage behavior into lines 572-579 in the discussion section. However, rather than copying/pasting our suggestion into the discussion, the authors should instead consider how inflammation resulting from their model may activate/recruit macrophages with appropriate citations.

Response: Previously, other reviewers suggested we trim down the Discussion section and make it concise. As this reviewer suggested here, we expanded a bit more to discuss the impact of intratibial injection-induced inflammation and macrophage and cited literature to support this. Please see Lines 576-582 and attached below.

"Prior studies reported the invasive bone drilling procedure of intratibial injections caused local inflammation^{63, 64}, which is known to enhance monocyte recruitment and alter the phenotypes and function of tissue-resident macrophages^{65, 66}. Thus, caution should be taken when investigating the bone niche and TAMs using intratibial injection models. Like what we did throughout this study, appropriate control groups and functional validation are required for immunophenotyping and data interpretation."

Reviewer #5 (Remarks to the Author):

The revision has adequately addressed all my concerns.

Response: We are grateful to this reviewer for his/her positive comments and efforts in improving this study!